# Multi-parametric thrombus profiling microfluidics detects intensified biomechanical thrombogenesis associated with hypertension and aging

Misbahud Din[1,2], Souvik Paul[1,2], Sana Ullah[1,2], Haoyi Yang[3], Rong-Guang Xu [1,2,4], Nurul Aisha Zainal Abidin[5], Allan Sun[5,6,7,8], Yiyao Catherine Chen[5], Rui Gao[5,8], Bari Chowdhury[1,2], Fangyuan Zhou [9], Stephenie Rogers[1], Mariel Miller[1,2], Atreyee Biswas[1,2], Liang Hu[10], Zhichao Fan [11], Christopher Zahner[2], Jing Fan[12], Zi Chen [4], Megan Berman[13], Lingzhou Xue [3], Lining Arnold Ju [5,6,7,8] & Yunfeng Chen [1,2] ✉

Arterial thrombosis is a leading cause of death and disability worldwide with no effective bioassay for clinical prediction. As a symbolic feature of arterial thrombosis, severe stenosis in the blood vessel creates a high-shear, high-gradient flow environment that facilitates platelet aggregation towards vessel occlusion. Here, we present a thrombus profiling assay that monitors the multi-dimensional attributes of thrombi forming in such biomechanical conditions. Using this assay, we demonstrate that different receptor–ligand interactions contribute distinctively to the composition and activation status of the thrombus. Our investigation into hypertensive and older individuals reveals intensified biomechanical thrombogenesis and multi-dimensional thrombus profile abnormalities, endorsing the diagnostic potential of the assay. Furthermore, we identify the hyperactivity of GPIbα-integrin $\alpha_{IIb}\beta_3$ mechanosensing axis as a molecular mechanism that contributes to hypertension-associated arterial thrombosis. By studying drug-disease interactions and inter-individual variability, our work reveals a need for personalized anti-thrombotic drug selection that accommodates each patient's pathological profile.

Arterial thrombosis, which describes the formation of pathological blood clots in the artery, is one of the leading causes of mortality and morbidity worldwide[1,2]. Pathological conditions such as hypertension, diabetes, metabolic syndrome, and aging not only increase thrombotic risks but also foster resistance to conventional antiplatelets that target soluble agonists (e,g., ADP, thrombin)-induced platelet activation and aggregation[3–6], contributing to high incidence and recurrence rates of cardiovascular diseases (CVD)[7]. However, the associated mechanisms are not fully elucidated. The current

clinical paradigm is further challenged by the lack of a standard bioassay for evaluating thrombotic risks: while conventional coagulation assays and aggregometry assays were indicated to be unreliable in predicting thrombosis or major adverse cardiovascular events, the new generation of hematological function assays (e.g., global coagulation assays and seer sonorheometry) also have limited evidence supporting their performance, and contain major drawbacks such as high cost, low sensitivity, and lack of standardization[8,9].

As an understudied but symbolic mechanism of arterial thrombosis, discoid platelets can be mechanically driven by the elevated shear stress and shear gradient caused by vessel stenosis to form large aggregates[5,10]—a phenomenon we termed "biomechanical platelet aggregation"[11,12]. The biomechanical platelet aggregation process is composed of two steps, mainly involving three molecular interactions. Firstly, shear-induced von Willebrand factor (VWF) activation[13] and glycoprotein (GP) Ibα (GPIbα)–VWF catch bond[14] together facilitate GPIbα–VWF binding under force, which initiates the aggregation of platelets in an activation-independent manner. Then, the GPIbα–VWF binding under force triggers GPIbα mechanosignaling that activates integrin $\alpha_{IIb}\beta_3$ to reach an intermediate affinity and an extended-close (E$^+$Act.$^-$) conformation[12], which subsequentially binds to fibrinogen (Fg) and VWF to allow more stable thrombus development. Notably, E$^+$Act.$^-$ integrin $\alpha_{IIb}\beta_3$ is only achievable via GPIbα mechanosignaling but not soluble agonist-induced platelet activation[12]. Biomechanical platelet aggregation cannot be effectively inhibited by conventional antiplatelets or amplification loop blockers (ALBs), but is strongly impeded by shear rate decrease and reversible upon the release of vessel stenosis[10,12]. Unfortunately, platelet mechanobiology was barely investigated in pathological contexts[15,16]. It remains unclear whether biomechanical platelet aggregation is intensified by any thrombotic risk factor and contributes to a higher incidence of CVD in certain human populations. Also, existing methods for observing biomechanical platelet aggregation[10,12,17] cannot provide all-around information and quantitative analysis regarding the composition of the thrombus and the activation status of platelets within, which hinders our understanding of arterial thrombosis and improvement of anti-thrombotic treatment. For example, it remains elusive how the three molecular interactions mediating platelet crosslinking in biomechanical platelet aggregation, i.e., GPIbα–VWF, integrin $\alpha_{IIb}\beta_3$–VWF and integrin $\alpha_{IIb}\beta_3$–Fg[10,12,18], respectively mediate the VWF and Fg levels and platelet activation in the biomechanical thrombus, and whether they are dysregulated by thrombotic risk factors to cause abnormal biomechanical thrombogenesis.

To address these outstanding clinical and scientific needs, we develop a thrombus profiling assay that combines a standard stenosis microfluidics setup with multi-color thrombus staining. It enables quick and all-around thrombus characterization under conditions that mimic the biorheological settings of arterial thrombosis. Using this assay, we delineated the differential roles of complex platelet cross-linking mechanisms in biomechanical thrombogenesis. We identified exacerbated biomechanical thrombogenesis and multi-dimensional thrombus abnormality associated with hypertension and aging, unraveling a clinical linkage between mechanobiology and arterial thrombosis. With complementary data from other experimental approaches, we further demonstrated that GPIbα and integrin $\alpha_{IIb}\beta_3$ receptors on hypertension patients' platelets have endogenous hyperactivity. By using the thrombus profiling assay to study drug–disease interactions and acquire personal thrombus profiles, we identified a gap in standard approaches of anti-thrombotics evaluation, which urges a re-evaluation of the efficacy and safety of anti-thrombotics using the "thrombus profile" and in the context of different pathology models. All the above results also showcase the potential of our thrombus profiling assay for anti-thrombotic drug screening, diagnosis of thrombotic risks, and personalized anti-thrombotic regimen selection.

## Results

### Thrombus profiling assay: development and validation

Our microfluidic chip is composed of ten rectangular (width × height: 200 μm × 50 μm) channels with respective inlets and outlets for tubing connection (Fig. 1a, b). A pump drives a syringe to perfuse heparinized blood (0.5 mL) through the channel pre-coated with VWF. An 80% stenosis site stimulates biomechanical thrombogenesis (Fig. 1c). A perfusion rate of 18 μL/min was selected, which creates a wall shear

stress (WSS) of 857 dyn/cm² at the stenosis site according to fluid dynamics simulation (Fig. 1d and Supp. Fig. 1a, f). The same stenosis site WSS is achieved by an inlet wall shear rate of 1485 s⁻¹ in a circular vessel with the same cross-sectional area (Supp. Fig. 1b, d, f), mimicking human arterioles[19], human arteries during systole, and mouse arteries[20]. The calculated Reynolds numbers in channels of different shapes are within the same scale (Supp. Fig. 1f). With the above settings, platelet thrombi can be consistently observed within the channel, which is primarily driven by shear force because no external agonist is added to the blood and the high-speed perfusion prevents the localized accumulation of agonists released from attached platelets and red blood cells. Due to the high-shear force, most thrombi have a tendency to grow toward the downstream side of the stenosis. Nonetheless, most thrombi (>85%) cover the whole stenosis apex, and most (>85%) thrombi have the point in their contour most close to the opposing channel wall positioned above the stenosis apex (Supp. Fig. 2), making the stenosis apex still the most likely position for occlusion. Replacing VWF with collagen for channel coating did not significantly affect thrombus formation. However, with collagen coating, thrombus formation was basically eliminated by antibody RU5 which blocks plasma VWF binding to collagen, reflecting an indispensable role of VWF on the hump for platelet attachment (Supp. Fig. 3a). Replacing heparin with citrate or ethylenediaminetetraacetic acid (EDTA) for anticoagulation attenuated thrombus formation (Supp. Fig. 3a), because the latter two chelate calcium from the blood and inhibit platelet activation, while EDTA also eliminates integrin $\alpha_{IIb}\beta_3$ activity. These results validate the use of VWF and heparin for channel coating and blood anticoagulation, respectively.

To comprehensively characterize the thrombus, 7 biomarkers with their respective molecular sensors were selected (Fig. 1e). Platelets are reported by SZ22, a monoclonal antibody (mAb) against integrin $\alpha_{IIb}\beta_3$[12]. Fg is reported by purified Fg that spikes the blood at 2% of plasma concentration. VWF is reported by non-inhibitory mAb 2.2.9[21]. P-selectin is reported by AK4 to indicate platelet α-granule release[22]. Phosphatidylserine (PS) exposure in the membrane is reported by Annexin V to signify platelet procoagulant function[23]. Conformationally extended (E$^+$) and fully activated (Act.) integrin $\alpha_{IIb}\beta_3$ are detected by mAbs MBC 370.2 and PAC-1, respectively, which together report integrin $\alpha_{IIb}\beta_3$ activation status[12]. The above sensors were grouped into two sets for fluorophore conjugation (Fig. 1e), where SZ22 appears in both sets for reference. All sensors have negligible influence on thrombogenesis (Supp. Fig. 3b–d).

Fluorescent signals were observed from all seven biomarkers (Fig. 1f). Agreeing with previous observations, real-time tracking showed rapid thrombogenesis in the first 300–400 s followed by a quasi-steady phase, in which the thrombus reaches a relative equilibrium between platelet aggregation and disaggregation[10,24] (Fig. 1g). Thus, we selected 450 s after the onset as the time point for quantitating fluorescent signals so as to assess the thrombus in the fully developed status while avoiding unnecessary waiting (Fig. 1h). Signal intensities of Fg, VWF, P-selectin, PS, and E$^+$ and Act. $\alpha_{IIb}\beta_3$ were normalized by platelet signal to assess their enrichment (Fig. 1i), where the high E$^+$ $\alpha_{IIb}\beta_3$ signal and low Act. $\alpha_{IIb}\beta_3$ signal agrees with our previous discovery that biomechanical platelet aggregation is mainly mediated by an intermediate activation state of $\alpha_{IIb}\beta_3$ integrins[12]. P-selectin expression and low-level PS exposure observed here (Fig. 1f, i; further confirmed using different microscope setup, staining agents, and microfluidic channel design (Supp. Fig. 4)) should be induced by GPIbα and/or integrin $\alpha_{IIb}\beta_3$ mechanosignaling[25–29]. The total signal intensity of platelets (first dimension, indicating thrombus size) and the normalized signal intensities of Fg, VWF, P-selectin, PS, E$^+$, and Act. $\alpha_{IIb}\beta_3$ (2nd–7th dimensions) are summarized into a seven-dimension thrombus profile (Fig. 1f, i).

Blood stored for >6 h or refrigerated overnight failed to generate visible thrombi (Fig. 1h), likely due to a loss of platelet activity

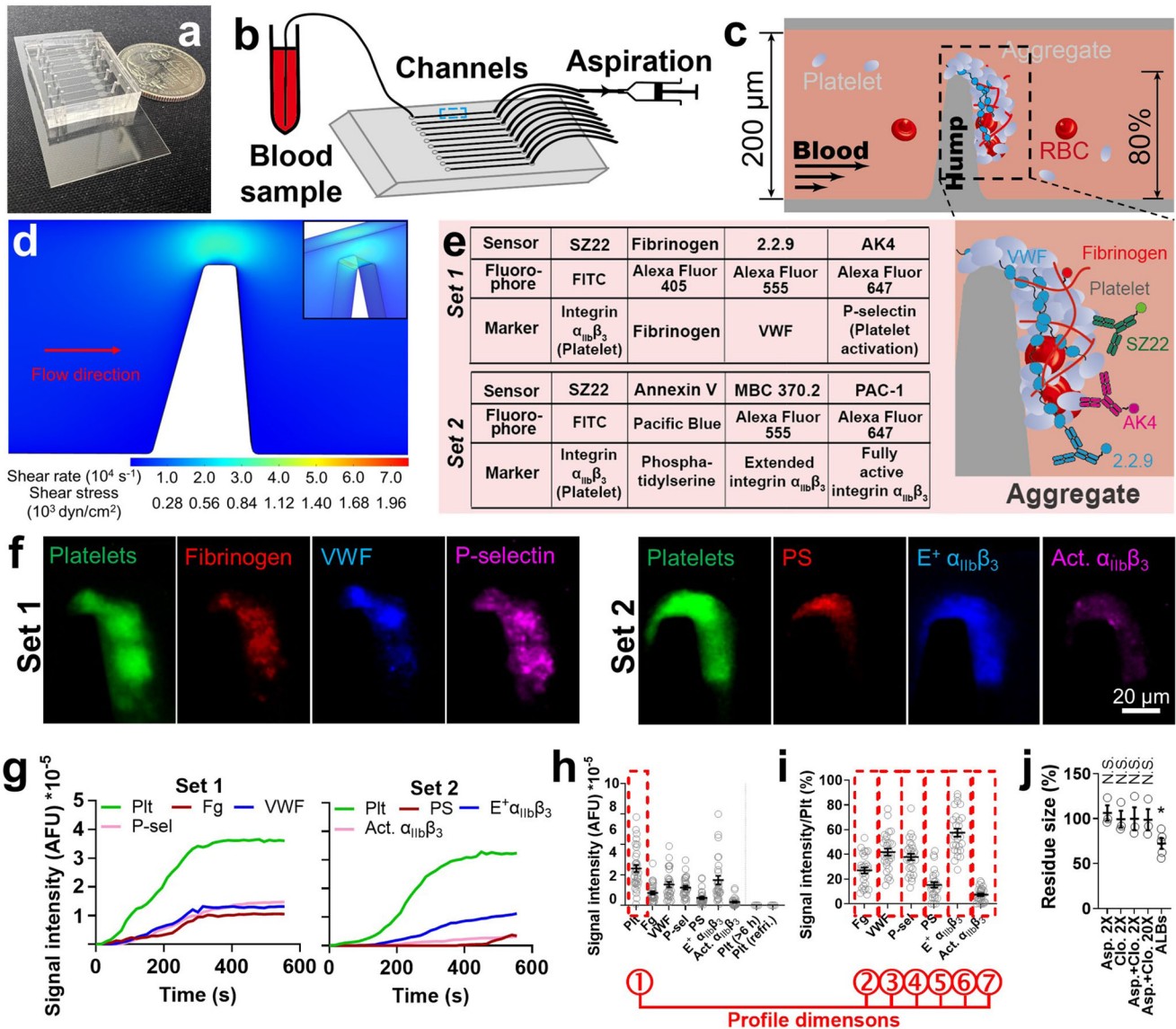

**Fig. 1 | Combining microfluidic stenosis assay with multi-fluorescence imaging to comprehensively characterize biomechanical platelet aggregation. a** A microfluidic chip with a quarter coin placed adjacently. **b** Illustration of the experimental setup. **c** Zoom-in of the dashed box in (**b**). A hump inside the channel creates 80% stenosis. When blood is perfused over, platelets spontaneously aggregate around the hump. **d** Shear rate and shear stress at the stenosis area estimated by fluid dynamics simulation. **e** Left: a layout of the two sets of fluorescently tagged sensors for thrombus profiling. Right: zoom-in of the thrombus shown in (**c**), illustrating the staining of Sensor Set 1. **f** Representative fluorescent images of thrombi stained with Sensor Set 1 (left) and 2 (right). The experiment was repeated independently on 28 healthy young subjects' blood samples. **g** Representative time courses of signal intensity of biomarkers in Sensor Set 1 (left, detecting platelets (Plt), fibrinogen (Fg), VWF, and P-selectin) and 2 (right, detecting platelets, phosphatidylserine (PS), extended integrin $\alpha_{IIb}\beta_3$ ($E^+ \alpha_{IIb}\beta_3$) and fully activated $\alpha_{IIb}\beta_3$ (Act. $\alpha_{IIb}\beta_3$)). AFU: arbitrary fluorescence unit. **h, i** Scatter plots with mean ± s.e.m. ($n = 28$) of the signal intensity of all biomarkers (**h**; expired and refrigerated blood samples were tested as controls, $n = 4$) and the normalized signal intensity of Fg, VWF, P-selectin, PS, $E^+ \alpha_{IIb}\beta_3$ and Act. $\alpha_{IIb}\beta_3$ (**i**) 7.5 min after the onset of thrombus formation. The definition of each dimension of the 7-dimension thrombus profile is indicated below the graphs. **j** Scatter plots with mean±s.e.m. ($n = 3$ subjects for the first 4 groups and =5 subjects for the last group) of the thrombus residue size in the presence of aspirin (2×) or clopidogrel (2×) or both (2× or 20×), or ALB cocktail. N.S., not significant, compared with no drug treatment, assessed by one-way ANOVA ($F$-value = 2.87, degrees of freedom = 21) and multiple comparison ($p = 0.9848$, >0.9999, >0.9999, >0.9999, =0.0447, respectively, from left to right).

during room temperature storage and GPIbα shedding during cold storage[30], respectively. Activated platelets can release and/or help produce soluble agonists such as thromboxane A2 and ADP to further activate themselves and recruit surrounding platelets to the growing thrombus, wherein the activation signaling processes are called amplification loops. However, conventional antiplatelet aspirin (targeting thromboxane A2 (TXA2)) and clopidogrel (targeting P2Y$_{12}$−ADP interaction) rendered negligible inhibition to the biomechanical thrombogenesis both separately and combined at twice or 20 times of human plasmatic concentrations[31] (Fig. 1j). In

contrast, both drugs can significantly inhibit ADP or collagen-induced platelet aggregation at much lower concentrations[31]. Also, a platelet ALB cocktail (including apyrase, MRS2179 and 2-MeSAMP to block ADP, indomethacin to block TXA2, and hirudin to block thrombin, all at saturating concentrations) only reduced the thrombus size by ~20% (Fig. 1j). These results corroborate the previous observations that inhibiting platelet amplification loops is ineffective in suppressing biomechanical platelet aggregation[10,12,32], endorsing a secondary role of soluble agonists in biomechanical thrombogenesis.

## Delineating the contribution of different receptor–ligand interactions

Biomechanical platelet aggregation is mainly mediated by a mechanosensing axis on the platelet surface composed of two mechanoreceptors: GPIbα and integrin $\alpha_{IIb}\beta_3$. GPIbα first binds to VWF to initiate platelet crosslinking, during which GPIbα mechanosignaling induces integrin $\alpha_{IIb}\beta_3$ intermediate activation (E⁺Act.⁻). The activated integrin $\alpha_{IIb}\beta_3$ binds to its ligands VWF and Fg to reinforce the platelet crosslinking process and also trigger its further activation towards the fully activated state (E⁺Act.⁺)[12]. To investigate how the above platelet-crosslinking mechanisms, namely, GPIbα–VWF, integrin $\alpha_{IIb}\beta_3$–VWF, and integrin $\alpha_{IIb}\beta_3$–Fg interactions, respectively mediate the growth, composition, and activation status of biomechanical thrombi, blood was treated with a panel of highly specific inhibitory mAbs to inhibit the interactions one at a time (Fig. 2a). AK2 and NMC4 both inhibit GPIbα–VWF interaction, with AK2 targeting GPIbα, and NMC4, previously shown to have anti-thrombotic effects[33], targeting VWFA1 domain (VWFA1) which binds to GPIbα[33,34]. LJ-P5 and 152B6 both inhibit integrin $\alpha_{IIb}\beta_3$–VWF interaction, with LJ-P5 blocking integrin $\alpha_{IIb}\beta_3$ binding to VWF but not Fg[35], and 152B6 blocking VWF binding to integrin $\alpha_{IIb}\beta_3$ but not GPIbα[36]. 7E9, LJ-155B39, and LJ-134B29 inhibit integrin $\alpha_{IIb}\beta_3$–Fg interaction by respectively blocking one of the three integrin-binding sites in Fg: γ408-411 (AGDV), Aα95-98 (RGDF), and Aα572-575 (RGDS)[37,38].

Single fluorescence imaging was first used to measure the dose-dependency of the above mAbs in inhibiting thrombogenesis (Fig. 2b–e). Only AK2 and NMC4, but not the other mAbs, eliminated thrombogenesis (Fig. 2c–e), which agrees with previous findings that GPIbα–VWF interaction serves as the initiator of biomechanical platelet aggregation[10,12]. At high concentrations, both LJ-P5 and 152B6 reduced the thrombus size to <20% (Fig. 2d), and the cocktail of 7E9, LJ-155B39, and LJ-134B29 also reduced the thrombus size to ~5% (Fig. 2e), indicating comparable importance of integrin $\alpha_{IIb}\beta_3$–VWF and $\alpha_{IIb}\beta_3$–Fg interactions. 7E9 alone achieved a strong inhibitory effect comparable to the cocktail, which corroborates the primary role of AGDV in Fg for integrin $\alpha_{IIb}\beta_3$ binding[39,40]. However, LJ-155B39 and LJ-134B29 also manifested considerable inhibition (Fig. 2e).

Half-maximal inhibitory concentrations (IC50) were acquired for these mAbs via model fitting (Supp. Table 1), which were then used in thrombus profiling. Both AK2 and NMC4 significantly decreased VWF, P-selectin, and E⁺ and Act. $\alpha_{IIb}\beta_3$ levels in the thrombus (Fig. 2f, g). In comparison, LJ-P5 and 152B6 only reduced VWF enrichment, while 7E9, LJ-155B39, and LJ-134B29 only reduced Fg enrichment; neither set of mAbs inhibited PS exposure, P-selectin expression, or integrin $\alpha_{IIb}\beta_3$ activation (Fig. 2h–l). None of the above mAbs affected the average signal intensity of SZ22-FITC, ruling out the possibility that the reduced VWF and Fg signals were due to increased platelet density (Supp. Fig. 5a). Altogether, our results indicate that different platelet-crosslinking mechanisms cooperatively mediate biomechanical thrombogenesis, with each having a distinct focus in their contribution to the thrombus composition and activation status.

To succinctly express the effects of different factors on biomechanical platelet aggregation, we created an "effect barcode" system with seven columns, each corresponding to one dimension of the thrombus profile. A positive, neutral, or negative effect of a factor on a dimension is respectively represented by a bar at the top, middle, or bottom of the column, also numerically expressed as "+", "0", or "−". Using this system, the effects of AK2 and NMC4 on the thrombus profile are both summarized as [- 0 - - 0 - -], those of LJ-P5 and 152B6 as [- 0 - 0 0 0 0], and those of 7E9, LJ-155B39, and LJ-134B29 as [- - 0 0 0 0 0] (Fig. 2m).

## Identifying an "addition rule" in the effect barcode system

Intrigued by how different receptor–ligand interactions synergize in mediating biomechanical thrombogenesis, we tested inhibitors with combinational effects. 7E3 (prototype of the antiplatelet abciximab) and 10E5 are mAbs that block integrin $\alpha_{IIb}\beta_3$ binding to both Fg and VWF[12,41,42]. Unlike specific inhibitors of integrin $\alpha_{IIb}\beta_3$–VWF or $\alpha_{IIb}\beta_3$–Fg, both 7E3 and 10E5 eliminated thrombogenesis at high concentrations (Fig. 3a). At IC50, both mAbs reduced Fg and VWF levels in the thrombus without affecting platelet activation markers, rendering an effect barcode of [- - - 0 0 0 0] (Fig. 3c, d). Interestingly, this barcode equals the add-up of those of integrin $\alpha_{IIb}\beta_3$–VWF ([- 0 - 0 0 0 0]) and $\alpha_{IIb}\beta_3$–Fg ([- - 0 0 0 0 0]) inhibitors (Fig. 3f).

Negatively charged nanoparticles inhibit platelet aggregation at high-shear rates due to their inhibition of VWF extension and, therefore, VWF–platelet interactions[43]. We tested two sizes of polystyrene negatively charged nanoparticles (PS-CNP) (50 and 510 nm), both showing biphasic dose-dependency in thrombus inhibition (Fig. 3b), consistent with the original report[43]. A concentration that decreases the thrombus size by ~50% was estimated for the 510-nm PS-CNP to perform thrombus profiling (Fig. 3e), which derived an effect barcode of [- 0 - - 0 - -]. Again, this barcode equals the add-up of those of GPIbα–VWF ([- 0 - - 0 - -]) and integrin $\alpha_{IIb}\beta_3$–VWF ([- 0 - 0 0 0 0]) inhibitors (Fig. 3f). The above results demonstrate that the mathematical addition rule applies to the effect barcode system. This addition rule will be further validated below in drug–disease interactions.

## Multi-dimensional thrombus profile abnormality in hypertension and aging

Aging and hypertension are strong risk factors for thrombosis[44,45]. To test the performance of our assay in identifying risks of arterial thrombosis, we first compared the thrombus size of healthy adults at different ages and identified that older ages (≥50) significantly increase the thrombus size (Fig. 4a). Furthermore, we tested blood samples from primary hypertension patients, which formed much larger biomechanical thrombi than healthy young subjects (Fig. 4b). By fitting the "thrombus size *versus* time" curves with the sigmoidal model, it was observed that unlike the growth of healthy young subjects' thrombi which approached a plateau at ~400 s, hypertension patients' thrombi remained in the rapid development phase until ~500 s, again indicating a prothrombotic tendency (Fig. 4b). Characterizing the thrombus profile revealed that aging and hypertension, either alone or together, significantly increased the thrombus size, Fg level as well as integrin $\alpha_{IIb}\beta_3$ activation in the thrombi, rendering the same effect barcode of [+ + 0 0 0 + +] (Fig. 4c). Two-way ANOVA with variance heterogeneity identified a bi-directional cooperation between hypertension and aging in increasing the thrombus size and E⁺ $\alpha_{IIb}\beta_3$ level, indicating synergy between these two risk factors (Fig. 4d). None of the above abnormalities was contributed by platelet density changes in the thrombus or hematocrit changes or platelet count increase in the blood (Supp. Fig. 5b–e).

We then evaluated the inter-correlation of the different biomarkers and their performance in distinguishing different cohorts. To address the scattering patterns of the signal intensities (Fig. 4c), which is likely due to inter-individual variability, multiple statistical analyses were performed for cross-checking. Firstly, by using the linear regression model, Spearman rank correlation coefficient[46] and Kendall's tau correlation coefficient[47], a positive correlation was consistently identified between thrombus size and Fg, E⁺ $\alpha_{IIb}\beta_3$, and Act. $\alpha_{IIb}\beta_3$ levels but not the other factors (Fig. 4e and Supp. Fig. 6), with E⁺ $\alpha_{IIb}\beta_3$ being the strongest correlating factor (Supp. Table 2). Secondly, among all markers, E⁺ $\alpha_{IIb}\beta_3$ has the best performance in separating healthy young from hypertensive and/or older age groups (Fig. 4e and Supp. Fig. 6a–e), with specificity and sensitivity respectively reaching 86 and 85%, comparable to the performance of thrombus size (Supp. Table 2). The consistency of E⁺ $\alpha_{IIb}\beta_3$ level with thrombus size in group separation also reached 81%. Altogether, these results unraveled intensified biomechanical thrombogenesis and multi-dimensional thrombus profile abnormality associated with hypertension and

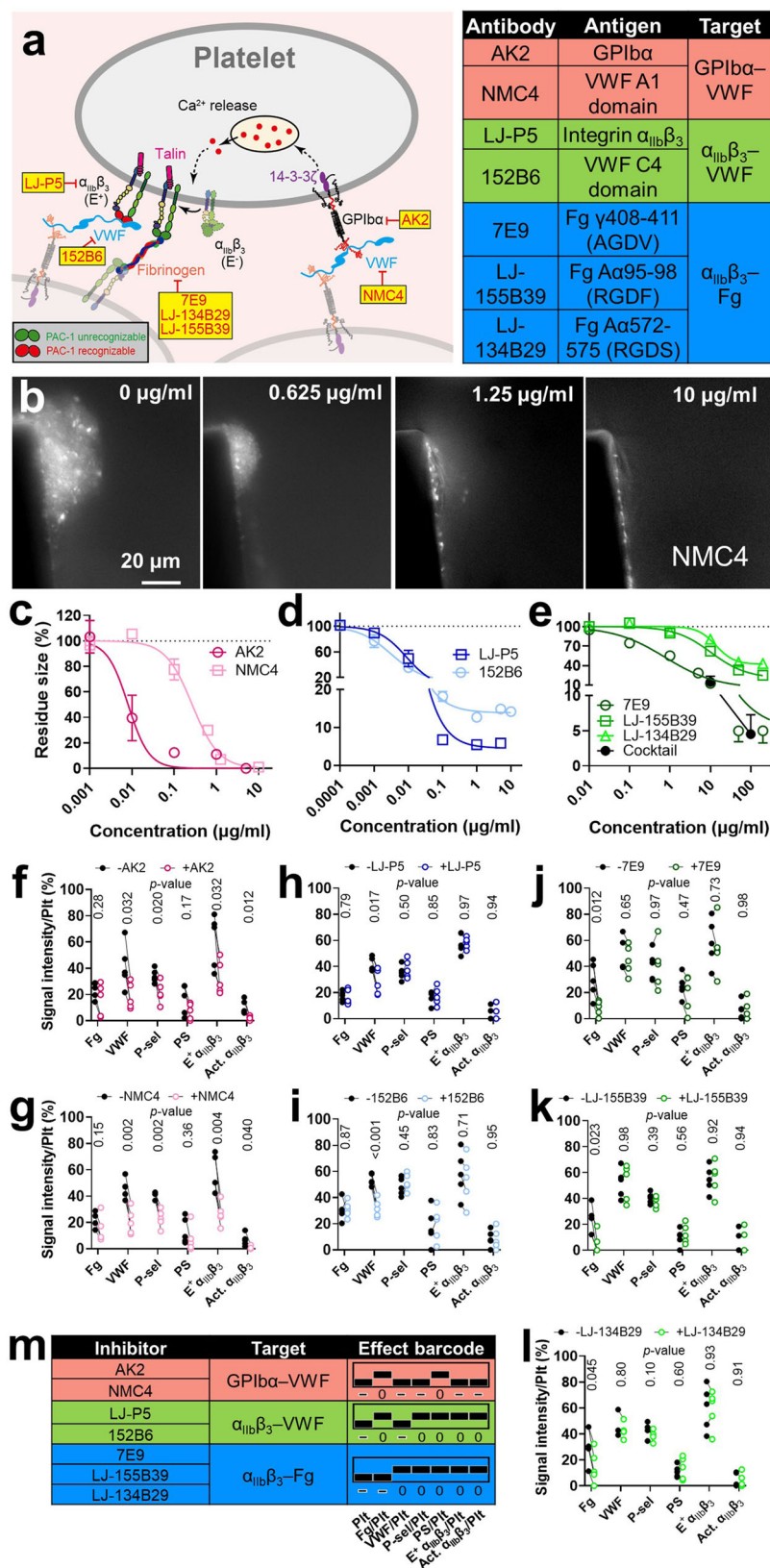

aging, and suggest $E^+$ $\alpha_{IIb}\beta_3$ as a potential biomarker for intensified biomechanical thrombogenesis.

Most hypertension patients enrolled in this study had their blood pressure well controlled by medication (systolic/diastolic <140/90 mmHg, respectively) and had hemoglobin A1C (HbA1C), body mass index (BMI) and cholesterol levels within the healthy range (Fig. 4f and Supp. Table 3). Furthermore, neither the size nor the $E^+$ $\alpha_{IIb}\beta_3$ level of these patients' thrombi has a significant correlation with the disease duration, systolic or diastolic blood pressure, or the sum of the two, or the patients' HbA1C level, BMI, total cholesterol, low-density lipoprotein cholesterol (LDL-C), high-density lipoprotein cholesterol (HDL-C), or triglyceride levels (Fig. 4f and Supp. Fig. 7a, b). Also, the thrombi of

**Fig. 2 | Delineating the respective contribution of GPIbα–VWF, α$_{IIb}$β$_3$–VWF, and α$_{IIb}$β$_3$–fibrinogen interactions to biomechanical platelet aggregation. a** (Left) Illustration of key receptor–ligand interactions in a biomechanical thrombus, highlighting the GPIbα-integrin α$_{IIb}$β$_3$ mechanosensing axis. A panel of monoclonal antibodies and their respective targets are indicated, which were used to inhibit one receptor or ligand at a time. The head of the integrin α$_{IIb}$β$_3$ colored green or red, respectively denotes the integrin being unrecognizable or recognizable by PAC-1. (Right) Table layout of tested antibodies and their respective antigens and targeting receptor–ligand interactions. **b** Representative images of DiOC$_6$(3)-labeled thrombi formed in the presence of different concentrations of NMC4. The experiment was repeated independently on 3 healthy young subjects' blood samples. **c–e** Dose-dependency curves of antibodies against GPIbα–VWF (**c**), α$_{IIb}$β$_3$–VWF (**d**), and α$_{IIb}$β$_3$–fibrinogen (**e**) interactions in reducing the size of

biomechanical thrombi (mean ± s.e.m.; data acquired from three subjects; $n \geq 2$ for each data point). **f–l** Comparing the normalized signal intensities of Fg, VWF, P-selectin, PS, E$^+$ α$_{IIb}$β$_3$, and Act. α$_{IIb}$β$_3$ in the biomechanical thrombi, in the absence and presence of AK2 (**f**), NMC4 (**g**), LJ-P5 (**h**), 152B6 (**i**), 7E9 (**j**), LJ-155B39 (**k**), and LJ-134B29 (**l**), respectively (mean ± s.e.m.; $n = 5$ subjects). $P$ values are results of multiple $t$-test with points without and with drug treatment paired. **m** Summarizing the effects of AK2, NMC4, LJ-P5, 152B6, 7E9, LJ-155B39, and LJ-134B29 on the thrombus profile into seven-digit barcodes. A positive, neutral, or negative effect is denoted by a bar being at the top, middle, and bottom of the column, respectively; it is also numerically denoted by "+", "0", or "−", respectively. The antibodies are categorized by their target receptor–ligand interaction, which is indicated by different background colors.

hypertensive subjects who have systolic and diastolic blood pressures and HbA1C, BMI, and cholesterol levels all in the normal ranges still have larger sizes and higher E$^+$ α$_{IIb}$β$_3$ levels than healthy young subjects, regardless of aging (Supp. Fig. 7c). These results indicate that hypertension can independently cause intensified biomechanical thrombogenesis and thrombus profile abnormality even with relatively short disease duration and effective antihypertensive medication. Nonetheless, we cannot exclude the likelihood that poorly controlled blood pressure, diabetes (high HbA1C level), obesity (high BMI) or dyslipidemia (abnormal cholesterol levels) can have extra contributions to the thrombus profile abnormality, especially considering that the latter three diseases are known risk factors of CVD.

Next, we inspected whether demographics other than age affect the thrombus profile. Within healthy young as well as hypertensive and/or older subjects, no significant difference in the thrombus size or E$^+$ α$_{IIb}$β$_3$ level was found between males and females or among different races/ethnicities (Fig. 4g–l). Seemingly in discrepancy with previous reports of a higher prevalence of CVD in males than in females and slight prevalence differences in different ancestries, these results corroborate more careful cohort studies demonstrating that the correlation of gender and ancestry with thrombotic risks is mainly due to the differential prevalence of social determinants of health and cardiovascular risk factors[45,48,49].

Due to size variations, different human arteries and arterioles have distinct Reynolds numbers (affecting flow patterns such as laminar versus turbulent) and shear rates in the blood flow[50,51], together resulting in a certain extent of diversification in the shear stress. However, changing the perfusion rate in our assay from 18 to 13.5, 27, and 36 μl/min (respectively changing the shear stresses to 0.75, 1.5, and 2 times of the original) did not significantly affect the thrombus profiling outcome (Supp. Fig. 8), wherein significantly larger thrombus size (Supp. Fig. 8a) and higher E$^+$ and Act. α$_{IIb}$β$_3$ levels (Supp. Fig. 8f, g), marginally higher Fg level (Supp. Fig. 8b) but comparable VWF, P-selectin, and PS levels (Supp. Fig. 8c–e) were consistently observed in the thrombi of hypertensive young subjects than healthy young subjects. These results validated that our assay could assess the general shear-driven platelet "aggregatability" of blood samples.

### Hypertension causes hyperactivity of the GPIbα-integrin α$_{IIb}$β$_3$ mechanosensing axis

We previously identified that the intermediate activation state of α$_{IIb}$β$_3$ integrin with an extended-close conformation (E$^+$Act.$^-$) plays a crucial role in biomechanical platelet aggregation[12]. Thus, the over-expressed E$^+$ α$_{IIb}$β$_3$, predominantly E$^+$Act.$^-$ α$_{IIb}$β$_3$ in the thrombi of hypertensive patients (Fig. 4c and Supp. Fig. 6f) should directly contribute to their intensified biomechanical thrombogenesis. We hypothesize that the E$^+$ α$_{IIb}$β$_3$ over-expression is possibly due to (1) hyperactivity in GPIbα, with triggers stronger mechanosignaling for integrin activation[12] and/or (2) integrin α$_{IIb}$β$_3$ pre-activation in the patients. To test these two hypotheses, we used four complementary approaches to investigate the activities of GPIbα and integrin α$_{IIb}$β$_3$ in hypertension patients.

Firstly, a conventional laminar flow chamber assay was used to assess the overall ligand binding activity of the two receptors. Unlike the stenosis assay, here, the channels adopt a plain surface pre-coated with VWFA1 or Fg to engage GPIbα and integrin α$_{IIb}$β$_3$, respectively. Plasma in the blood was depleted and replaced with buffer to remove endogenous VWF and Fg and prevent platelet aggregation. By perfusing blood through the channels under varied shear rates, it was found that platelets from hypertensive young, hypertensive older, and healthy older groups all achieved much higher surface coverage and slower rolling on VWFA1 than the healthy young group (Fig. 5a–c and Supp. Fig. 9a, b). On the other hand, only hypertensive young and hypertensive older groups achieved high surface coverage on Fg (Fig. 5a, d). These results indicate that both hypertension and aging cause GPIbα hyperactivity, but only hypertension induces hyperactivity in integrin α$_{IIb}$β$_3$ at the same time. Considering that the activities of GPIbα and integrin α$_{IIb}$β$_3$ in hypertensive young and hypertensive older subjects were comparable (Fig. 5b–d), mechanistic studies below combined young and older hypertensive subjects into a single cohort to compare with healthy young subjects. However, this does not exclude the possibility that aging can influence hypertensive patients' GPIbα and integrin α$_{IIb}$β$_3$ as a secondary factor, which shall be inspected in future studies.

Secondly, a single-molecule force spectroscopy technique, bio-membrane force probe (BFP)[52], was used to measure the ligand binding of single platelets. A micropipette-aspirated biotinylated human red blood cell (RBC) was used as an ultrasensitive force transducer, and a probe bead co-functionalized with streptavidin and VWFA1 or Fg was glued to the RBC apex. A platelet was aspirated by an opposing micropipette and driven to repeatedly contact the bead, which induced adhesion events to measure the receptor–ligand binding kinetics (Fig. 5e). Adhesion frequency assay was first deployed to enumerate the absence or presence of adhesion events after long contacts to calculate the steady-state adhesion frequency, $P_a$[53]. The $P_a$ of hypertensive subjects' platelets adhering to the same batch of VWFA1 and Fg beads were significantly higher than healthy young (Fig. 5f, i), reflecting a significantly higher effective avidity (ligand-binding capability of each unit of platelet surface area) of both GPIbα and integrin α$_{IIb}$β$_3$ (Fig. 5g, j, left), consistent with the platelets' enhanced capability of engaging VWFA1 and Fg in the flow chamber (Fig. 5a–d). Dividing effective avidities by the receptors' surface densities showed that the average effective affinities of GPIbα and integrin α$_{IIb}$β$_3$ on the hypertensive subjects' platelets were also significantly enhanced (Fig. 5g, j, right). Then, the BFP force-clamp assay was used to measure the stability of single GPIbα–VWFA1 and single integrin α$_{IIb}$β$_3$–Fg bonds under force. This was achieved by adjusting the contact time between the bead and the platelet to achieve $P_a \approx 20\%$, thereby realizing a ~90% probability of single bonds[53]. The GPIbα–VWFA1 bond lifetime of hypertensive subjects' platelets manifested a 'slip bond' instead of a triphasic 'slip-catch-slip' trend seen on healthy young subjects' platelets[54], resulting in a substantial prolongation of bond lifetimes under forces <20 pN (Fig. 5h). On the other

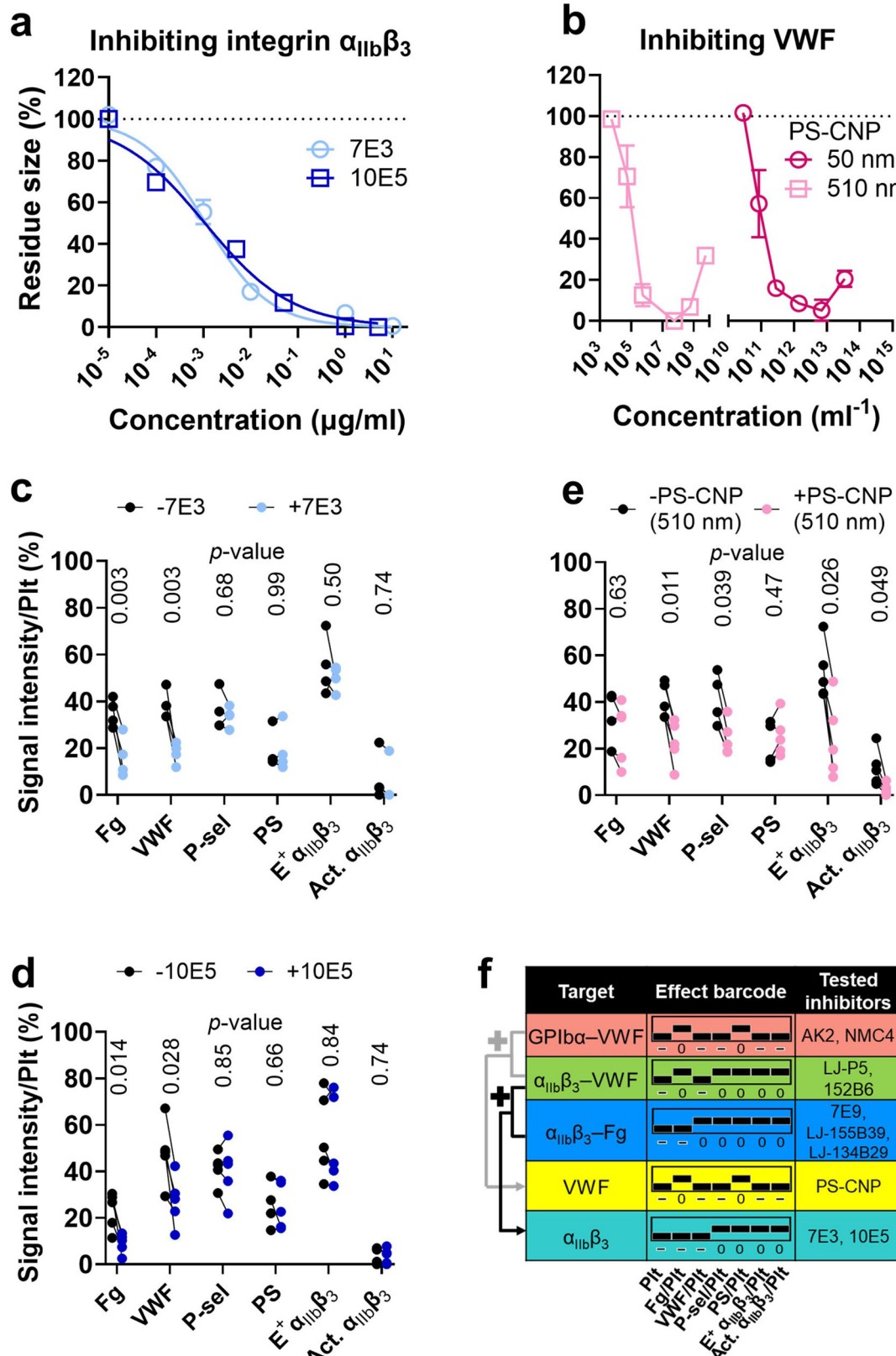

hand, hypertension caused a substantial rightward and upward shift of the integrin $\alpha_{IIb}\beta_3$–Fg catch bond[12], so that the peak force increased from ~15 to ~35 pN, the peak lifetime increased from ~5 to ~10 s, and the force range where lifetime events were observable was widened from 0-40 to 0-65 pN (Fig. 5k). Notably, this lifetime curve from hypertensive subjects' platelets also resembles healthy young subjects' E⁺Act.⁻

integrin $\alpha_{IIb}\beta_3$–Fg lifetime curve characterized before[12]. Altogether, our BFP results indicate that hypertension increases not only the avidity, but also the affinity and force-regulated ligand binding strength of platelet GPIbα and integrin $\alpha_{IIb}\beta_3$.

Thirdly, we combined fluorescence imaging with BFP (fBFP) to study whether the increased affinity and ligand binding strength of

**Fig. 3 | Testing the effects of inhibitors against integrin $\alpha_{IIb}\beta_3$ and VWF on biomechanical platelet aggregation. a** Dose dependency of 7E3 and 10E5, in reducing the size of the biomechanical thrombi (mean ± s.e.m.; data acquired from three subjects; $n \geq 2$ for each data point). **b** Dose dependency of two sizes (50 and 510 nm) of polystyrene negatively charged nanoparticles (PS-CNP), in reducing the size of the biomechanical thrombi (mean±s.e.m.; data acquired from three subjects; $n \geq 2$ for each data point). **c–e** Comparing the normalized signal intensities of Fg, VWF, P-selectin, PS, E$^+$ $\alpha_{IIb}\beta_3$, and Act. $\alpha_{IIb}\beta_3$ in the biomechanical thrombi, in the absence and presence of 7E3 (**c**), 10E5 (**d**), and PS-CNP (diameter: 510 nm) (**e**), respectively (mean ± s.e.m.) ($n$ = 4 or 5 subjects). $P$ values are the results of multiple $t$-test with points without and with drug treatment paired. **f** Summarizing the seven-digit effect barcodes of 7E3, 10E5, and PS-CNP. A rule of addition is indicated, demonstrating that the add-up of the barcodes of GPIbα−VWF inhibition and $\alpha_{IIb}\beta_3$−VWF inhibition equals that of VWF inhibition, and that the add-up of the barcodes of $\alpha_{IIb}\beta_3$−VWF inhibition and $\alpha_{IIb}\beta_3$−Fg inhibition equals that of integrin $\alpha_{IIb}\beta_3$ inhibition.

GPIbα in hypertension patients can result in stronger mechanosignaling to better induce integrin $\alpha_{IIb}\beta_3$ activation. Platelets pre-loaded with a $Ca^{2+}$ dye (Fura-2) were repeatedly stimulated by a VWFA1-coated bead in force-clamp cycles at a fixed 2-s contact time for 5 min (Fig. 5l), while the normalized intraplatelet $Ca^{2+}$ level was monitored (Fig. 5m). Agreeing with our hypothesis, hypertension patients' platelets fluxed stronger $Ca^{2+}$ signals−reflected by higher $Ca^{2+}$ peak increase−than healthy young subjects' platelets under a wide force range (Fig. 5n). Unlike healthy young subjects' platelets where the $Ca^{2+}$ signal intensity first increases and then decreases as clamping force increases, mirroring their lifetime's 'catch-slip' trend, the $Ca^{2+}$ signal intensity of hypertension patients' platelets manifested a gradual decline, also consistent with the shape of their GPIbα−VWFA1 lifetime slip bond (Fig. 5n). This corroborates our previous finding that the GPIbα mechanosignaling intensity, manifested by both $Ca^{2+}$ flux and integrin $\alpha_{IIb}\beta_3$ activation, heavily relies on the duration of force pulling on GPIbα[55].

Fourthly, flow cytometry was used to investigate whether the $\alpha_{IIb}\beta_3$ integrins on hypertension patients' platelets are pre-activated. While similar high expression of integrin $\alpha_{IIb}\beta_3$ and baseline expression of Act. $\alpha_{IIb}\beta_3$ and P-selectin were detected on the platelets of healthy young and hypertensive subjects, the expression of E$^+$ $\alpha_{IIb}\beta_3$ in the hypertensive group was much higher than in the healthy young group (Fig. 5o−s). Although hypertension patients' platelets are slightly larger than healthy young subjects', a positive correlation between E$^+$ $\alpha_{IIb}\beta_3$ signal and platelet volume was found only in the hypertensive group but not the healthy young group (Supp. Fig. 9c, d). These results indicate that hypertension patients' platelets are pre-activated, with integrin $\alpha_{IIb}\beta_3$ up-regulated to the intermediate activation state (E$^+$Act.$^-$) and minimal P-selectin expression.

Altogether, our results indicate that two mechanisms work in parallel to induce E$^+$ $\alpha_{IIb}\beta_3$ over-expression in the biomechanical thrombi of hypertensive patients (Fig. 5t): (1) some $\alpha_{IIb}\beta_3$ integrins already adopt a native E$^+$ status rather than remaining inactive as on healthy platelets; and (2) hyperactive GPIbα triggers stronger mechanosignaling upon VWF binding, inducing more $\alpha_{IIb}\beta_3$ integrins to undergo E$^+$ activation than on healthy platelets.

### Expanding the addition rule to drug−disease interactions

Using the thrombus profiling assay, we tested how anti-thrombotic inhibitors affect the thrombus profile of hypertension patients. Consistent with our results on healthy subjects, the combination of aspirin and clopidogrel at twice their human plasmatic concentrations in clinical practice[31] showed no effect on hypertension patients' thrombi (Fig. 6a). In contrast, at IC50, NMC4 reduced the thrombus size and E$^+$ $\alpha_{IIb}\beta_3$ and Act. $\alpha_{IIb}\beta_3$ expressions to healthy levels, but also lowered VWF and P-selectin levels that were unaffected by hypertension (Fig. 6a). The resulting effect barcode, [0 + - - 0 0 0], equals the add-up of those of NMC4 and hypertension (Fig. 6b). Similarly, adding 7E3 to hypertension patients' blood resulted in an effect barcode of [0 0 - 0 0 + +], equaling the add-up of the effect barcodes of 7E3 and hypertension (Fig. 6a, b). These results indicate that the addition rule of the barcode system can also be applied to predict drug−disease interactions. Neither NMC4 nor 7E3 completely corrected the effect barcode of hypertension, with 7E3 even incapable of suppressing the integrin $\alpha_{IIb}\beta_3$ over-activation, implying a treatment mismatch between the inhibitors and the patients.

### Inter-individual variability in personal thrombus barcodes

Lastly, to evaluate the normality and abnormalities of individuals' thrombus profiles, we created the concept of "personal thrombus barcodes". From the thrombus profiles of healthy young subjects, values of each dimension were fitted to a Gaussian distribution, of which the mean ± 2 s.d. (~95% confidence interval) was defined as the reference range ("0") (Supp. Fig. 10), and values lower or higher were defined as abnormally low ("−") and high ("+"), respectively (Fig. 6c).

Applying this system to healthy young subjects rendered all dimensions of thrombus profiles being dominated by normal values, with only very small fractions being abnormally low or high, which is consistent with the definition of the reference ranges (Fig. 6d). In contrast, much larger fractions of healthy older, hypertensive young, and hypertensive older subjects had abnormally large thrombi and high E$^+$ and Act. $\alpha_{IIb}\beta_3$ levels, with moderately higher fractions also having abnormally high Fg levels (Fig. 6d). Most of these subjects (26/36) have abnormally high values in thrombus size and E$^+$ $\alpha_{IIb}\beta_3$ level, yet 3 subjects with abnormally large thrombi have a normal E$^+$ $\alpha_{IIb}\beta_3$ level. An abnormally high Act. $\alpha_{IIb}\beta_3$ level was observed in half of the subjects with large thrombi (14/29), but also in two subjects with normal-sized thrombi (Supp. Fig. 11). Most subjects in these three groups (23/36) have abnormal VWF, P-selectin, and PS levels, which may or may not co-exist with abnormally high values of thrombus size and E$^+$ $\alpha_{IIb}\beta_3$ level. Of all the 69 subjects, a total of 30 different personal thrombus barcodes were identified (Supp. Fig. 11). Overall, the above results indicate strong inter-individual variability in the personal thrombus barcode that cannot be ascribed to only disease and aging, and demonstrate obvious decoupling of the different dimensions in the thrombus profile. Notably, we repeated our test on 14 randomly picked subjects after different time intervals (from 2 weeks to 9 months). Among a total of 21 re-tests, only two showed changes in the personal thrombus barcodes, which were associated with the longest time intervals (7 and 9 months, respectively) (Supp. Fig. 11). This reflects the high reliability of our assay and indicates that the personal thrombus profiles of individuals are relatively stable but can still vary over time.

Inter-individual variability was also observed in the subjects' responses to anti-thrombotic inhibitors. While NMC4 effectively corrected the size and E$^+$ $\alpha_{IIb}\beta_3$ level of most hypertension patients' thrombi (Fig. 6d), it did not uniformly modify their personal thrombus barcodes, but instead produced three different barcodes in five patients' blood samples (Fig. 6e). Similar diversification was found in 7E3, despite its consistent negative effect on the thrombus size and neutral effect on E$^+$ $\alpha_{IIb}\beta_3$ level (Fig. 6d, e). These diversifications cannot be completely ascribed to differences in the patients' original personal thrombus barcodes (Fig. 6e).

## Discussion

The methodology framework developed in this study includes not only the experimental setup itself, but also the 'thrombus profile', the barcode systems, and the "addition rule" as conceptual elements. Unlike conventional laboratory and point-of-care assays, our thrombus profiling assay mainly assesses biomechanical platelet aggregation. Because the high-shear, high-gradient blood flow associated with arterial thrombosis reinforces biomechanical platelet aggregation[10,32] and, at the same time, impedes soluble agonist-induced platelet aggregation and coagulation by limiting the local accumulation of

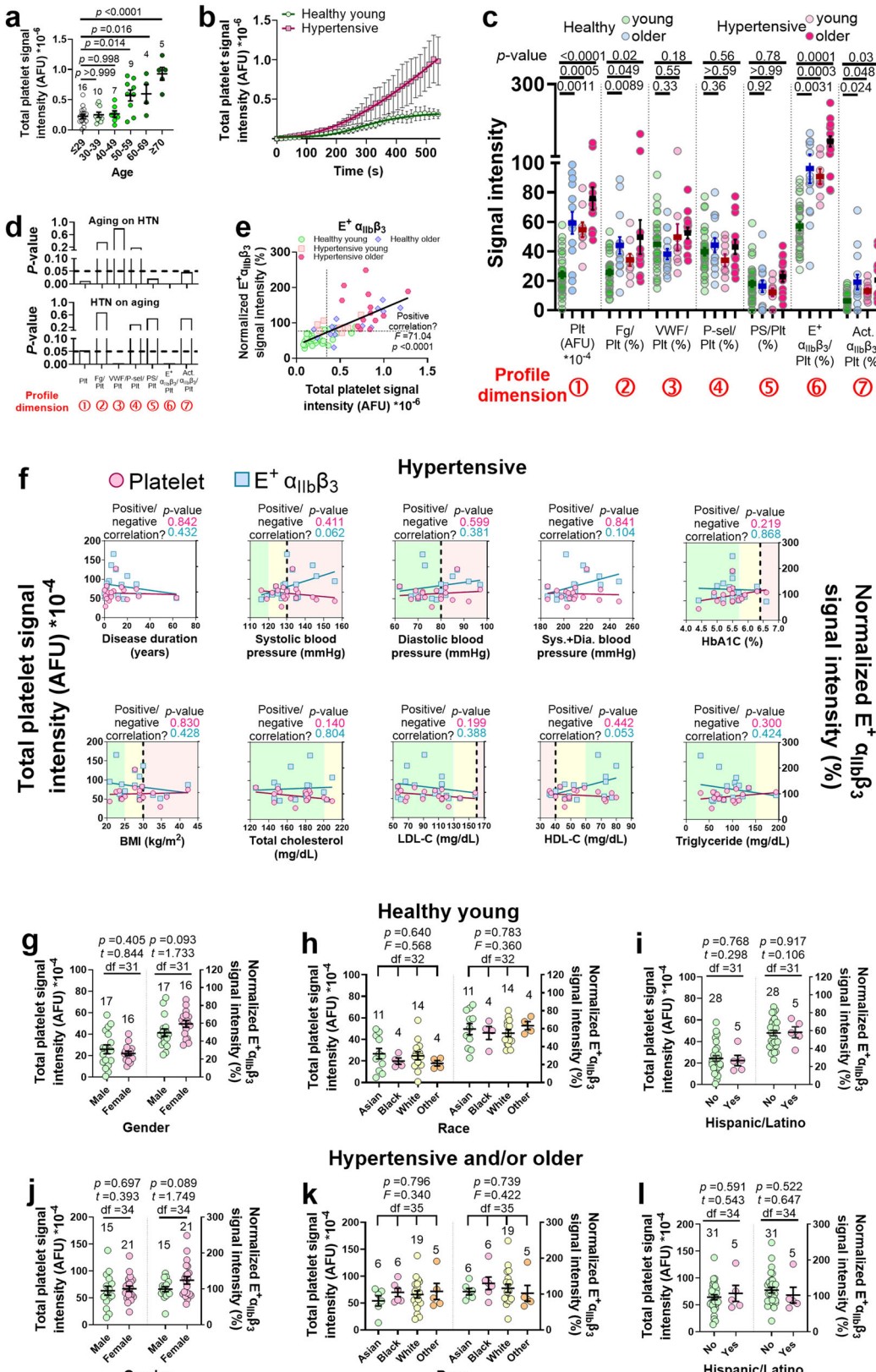

soluble substances[56], biomechanical platelet aggregation should be one, and possibly the most, essential mechanism of arterial thrombosis. This rationalizes the outstanding performance of our thrombus profiling assay in testing clinical subjects associated with higher risks of arterial thrombosis, demonstrating its potential to clinically assess thrombotic risks in general populations. In this context, the detection

of integrin $\alpha_{IIb}\beta_3$ over-activation and the identification of "treatment mismatch" further showcase the assay's ability in identifying the mechanisms of prothrombotic tendency and in evaluating prevention/treatment strategies. Meanwhile, cost-effectiveness and low sample volume represent additional advantages. Hardware upgrades, e.g., using a multi-channel syringe pump and a motorized stage or a multi-

**Fig. 4 | Characterizing abnormalities in the thrombus profiles associated with hypertension and aging. a** Scatter plots with mean ± s.e.m. (*n* indicated above each column) of the size of thrombi generated by healthy subjects grouped by age. *P* values are the results of one-way ANOVA (*F* value = 14.11, degrees of freedom = 50) and multiple comparison. **b** Comparing the time course of thrombus growth (mean ± s.e.m. with fitting lines of a sigmoidal model) between healthy young and hypertensive groups (*n* = 8). **c** Scatter plots with mean ± s.e.m. of the thrombus profiles of healthy young, healthy older, hypertensive young, and hypertensive older adult subjects (*n* = 33, 14, 9, and 13, respectively). *P* values are the results of two-way ANOVA (*F* values = 7.85, 106.3, 43.66; degrees of freedom = 18, 6, 3, for interaction, row factor and column factor, respectively) and multiple comparison. **d** *P* values of the significance of the impact of aging on the thrombus profile of hypertensive subjects (top), and that of hypertension on the thrombus profile of older subjects (bottom). **e** Scatter plot of normalized E$^+$ $\alpha_{IIb}\beta_3$ signal intensity vs. thrombus size, with blood from healthy young, healthy older, hypertensive young, and

hypertensive older subjects (*n* = 33, 14, 9, 13, respectively). Solid line: linear fitting of all data points, with a two-sided regression slope test performed to show a significant positive correlation. Dash lines: threshold values that best separate healthy young and other groups. **f** Scatter plots and linear fits (*P* values: results of two-sided regression slope test) of thrombus size and normalized E$^+$ $\alpha_{IIb}\beta_3$ signal intensity vs. hypertension duration, systolic and diastolic blood pressures and their sum, HbA1C, BMI, total cholesterol, LDL-C, HDL-C, and triglyceride in hypertension patients. Green, yellow, and red background colors indicate normal, borderline abnormal, and pathologically abnormal ranges, respectively. **g–l** Scatter plots with mean ± s.e.m. (*n* indicated above each column) of the thrombus size (left) and normalized E$^+$ $\alpha_{IIb}\beta_3$ signal intensity (right) of healthy young (**g–i**) or hypertensive and/or older (**j–l**) subjects, grouped by gender (**g, j**), race (**h, k**), and ethnicity (**i, l**). One-way ANOVA and multiple comparison or Student's *t*-test was performed for data comparison, with *p* values, *F*- or *t*-values, and degrees of freedom (df) annotated on the figures.

camera array to reach relatively high throughput, and/or system automation, will enable the current setup to become more suitable for clinical practice. To provide more accurate diagnosis and treatment suggestions, the assay can benefit from more detailed segmentation (e.g., borderline, stage-I, and stage-II abnormal) in judging normal *versus* abnormal thrombus barcodes, and can be combined with other existing diagnostic approaches, e.g., risk score assessment[57,58]. Notably, the assay also has the potential of evaluating bleeding tendency in humans and the bleeding side effect of anti-thrombotic agents[24], which warrants future investigation. As a limitation, our assay cannot recapitulate the biomechanical scenarios of thrombosis in all different arteries and arterioles, especially in large stenotic arteries where the Reynolds number can reach sufficiently high to trigger turbulence[50]. Nonetheless, by replicating critical aspects of thrombosis, the assay allows the evaluation of the general prothrombotic tendency of blood samples.

Using a panel of mAbs with highly specific targets, we showed that GPIbα–VWF, integrin $\alpha_{IIb}\beta_3$–VWF, and integrin $\alpha_{IIb}\beta_3$–Fg interactions all contribute to the size growth of biomechanical thrombi. However, suggesting a central role of GPIbα–VWF interaction in biomechanical thrombogenesis, only its blockage but not the blockage of integrin $\alpha_{IIb}\beta_3$–VWF or integrin $\alpha_{IIb}\beta_3$–Fg interaction can eliminate thrombus formation. Also, only blocking GPIbα–VWF interaction inhibits integrin $\alpha_{IIb}\beta_3$ activation while blocking integrin $\alpha_{IIb}\beta_3$–ligand interactions failed so, reflecting a primary role of GPIbα for integrin $\alpha_{IIb}\beta_3$ activation in the GPIbα-integrin $\alpha_{IIb}\beta_3$ mechanosensing axis[12]. We showed that GPIbα–VWF and integrin $\alpha_{IIb}\beta_3$–VWF interactions both modulate the deposition of VWF into the thrombus, while integrin $\alpha_{IIb}\beta_3$–Fg only modulates that of Fg, which seems intuitive because both VWF and Fg need to be bound to their respective platelet receptors to maintain their presence in the thrombus. However, the fact that inhibiting either VWF or Fg binding to integrin $\alpha_{IIb}\beta_3$ does not enrich the other ligand, but both reduce the thrombus size, suggests that VWF and Fg cooperate, rather than mutually compensate, in integrin $\alpha_{IIb}\beta_3$ crosslinking for biomechanical platelet aggregation. Our observation that inhibiting the two RGD sequences in Fg effectively reduces the thrombus size contrasts with the previous report that mutating either of these two sequences did not impair Fg function in mediating ADP-induced platelet aggregation[59]. This is likely because ADP activates integrin $\alpha_{IIb}\beta_3$ to the fully active state, while biomechanical platelet aggregation is mainly driven by intermediate state integrin $\alpha_{IIb}\beta_3$[12], so that the RGD sequences in Fg are redundant in the former scenario for platelet crosslinking but become a useful supplement to the Fg AGDV sequences in the latter. This suggests mechanistic distinctions when Fg mediates biomechanical *versus* biochemical platelet aggregation and unravels an underestimated contribution of the Fg RGD sequences to arterial thrombosis. On the other hand, previous works showed that when the shear rate increases, the dependency of shear-induced platelet aggregation on GPIbα and integrin $\alpha_{IIb}\beta_3$ becomes progressively

stronger and weaker, respectively[60]. It will be interesting to test whether changing the shear rate in our assay affects how the three receptor–ligand interactions contribute to the thrombus profile. Lastly, our results appear to indicate that the effect barcode of each anti-thrombotic agent is dictated by its target rather than its pharmacological design. Moreover, the observed "addition rule" suggests a lack of synergy or discord when multiple targets are concurrently inhibited, indicating that different molecular interactions and signaling pathways function in relatively independent and parallel ways. These principles are potentially useful for drug screening, enabling us to quickly narrow down the possible target(s) of uncharacterized anti-thrombotic agents using their effect barcode. To serve the above purpose, inhibitors of all other contributing factors of biomechanical platelet aggregation, e.g., mechanosignaling of GPIbα, integrin $\alpha_{IIb}\beta_3$, and Piezo1[16,61,62], need to be tested to acquire their effect barcodes.

Hypertension is the leading cause of CVD and is also closely associated with antiplatelet (e.g., aspirin and clopidogrel) resistance[3,4]. Among multiple postulated mechanisms, abnormal platelet activation has been identified as a central contributor to the prothrombotic status of hypertension patients, where changes in platelet morphology and biochemical activities (e.g., elevated sensitivity to soluble agonists, reduced sensitivity to exogenous nitric oxide) were reported[63]. In comparison, we discover that GPIbα in hypertension patients are hyperactive and can induce stronger mechanosignaling, while a substantial amount of integrin $\alpha_{IIb}\beta_3$ molecules are already in the E$^+$ status, which together results in an over-expression of E$^+$ integrin $\alpha_{IIb}\beta_3$ in the patients' biomechanical thrombi. Considering the central roles of GPIbα and E$^+$ integrin $\alpha_{IIb}\beta_3$ in biomechanical platelet aggregation[12], these results explain the intensified biomechanical thrombogenesis observed in hypertension patients' blood, and suggest that GPIbα-integrin $\alpha_{IIb}\beta_3$ mechanosensing axis hyperactivity directly contributes to the high incidence rate of CVD in hypertension patients. On the other hand, antiplatelet resistance is conventionally believed to be due to patients' lack of sensitivity to antiplatelets in inhibiting platelet amplification loops[64]. However, we found that biomechanical thrombogenesis is essentially "immune" to aspirin and clopidogrel in both healthy young subjects and hypertension patients (Supp. Fig. 12). These, together with similar observations by other works[10,12,32], indicate a new mechanism of antiplatelet resistance: biomechanical platelet aggregation can mediate arterial thrombosis independent of platelet amplification mechanisms, and therefore the sole inhibition of platelet amplification loops allows thrombotic risks to persist by leaving biomechanical platelet aggregation active. Altogether, our results strongly advocate the development of GPIbα and/or integrin $\alpha_{IIb}\beta_3$ targeting anti-thrombotic "mechanomedicines" that can work complementarily with conventional antiplatelets for enhanced treatment efficacy. The results also underscore the pathophysiological relevance of E$^+$-closed integrin $\alpha_{IIb}\beta_3$, which should inspire future investigations on the importance of the E$^+$-closed conformation in other integrins and

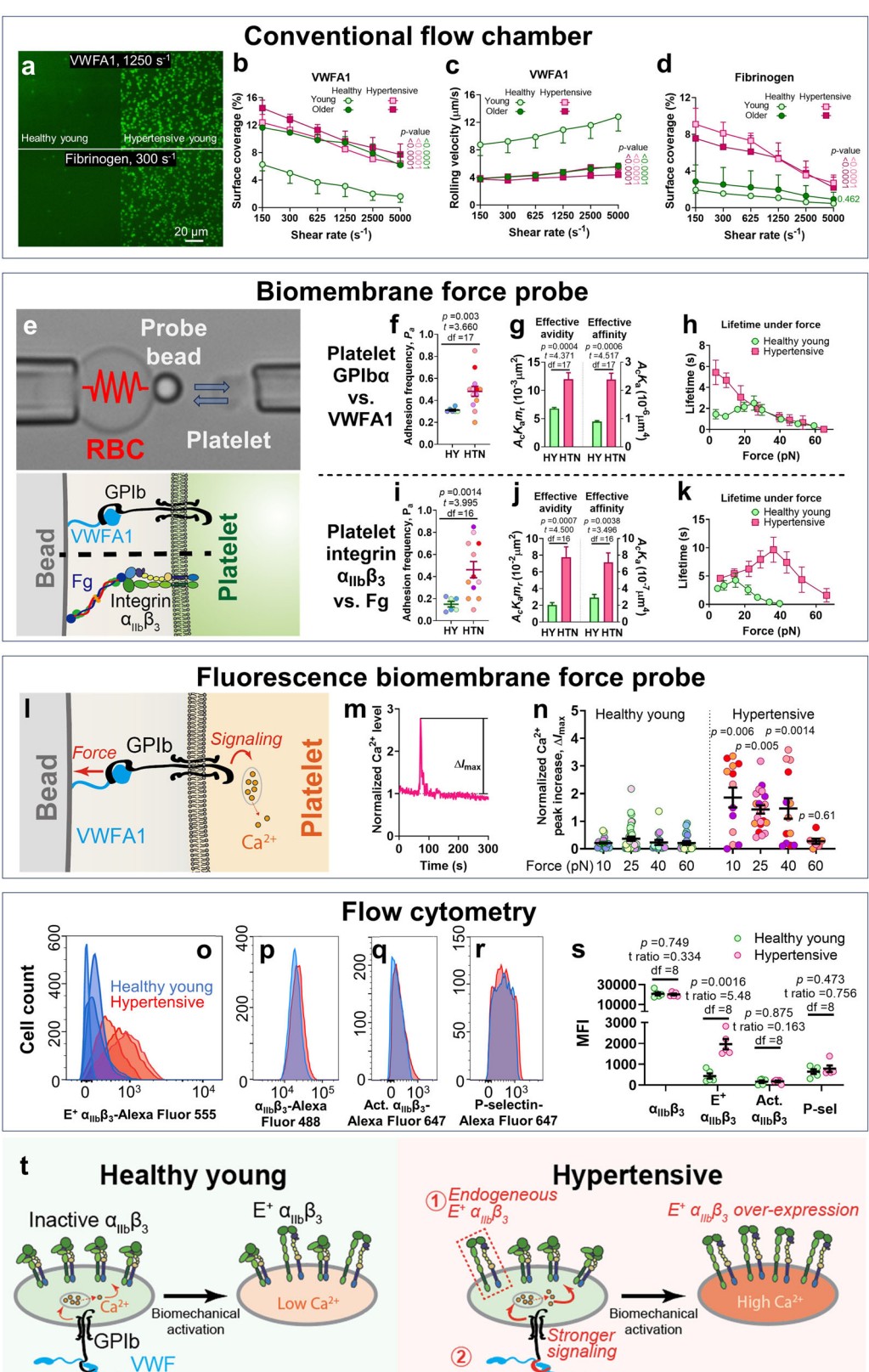

in the context of other diseases. The causes of GPIbα and integrin αIIbβ3 hyperactivity in hypertension patients as well as the similar trend of platelet hyperreactivity in older people warrant further investigation, which are possibly relevant to hypertension/aging-associated oxidative stress and inflammation that cause platelet pre-activation[65–67], dysregulated glycosylation of GPIbα and integrin αIIbβ3 by metabolic disorders[68,69], and/or the activation of mechanosensitive ion channel Piezo1 that causes platelet hyper-sensitivity to shear force[16]. On the other hand, the slightly higher Fg level in hypertensive and older subjects' thrombi is likely due to the elevated Fg plasma concentration in these populations[70,71] as well as integrin αIIbβ3 hyperactivity that more efficiently recruits Fg.

**Fig. 5 | Hyperactivity of GPIbα and integrin $\alpha_{IIb}\beta_3$ associated with hypertension. a** Snapshots of healthy young and hypertensive young subjects' platelets adhering to VWFA1 (upper) or Fg (lower). Experiments were repeated for four subjects. **b–d** Mean ± s.e.m. ($n = 4$ subjects) of platelet surface coverage (**b**, **d**) and rolling velocity (**c**) vs. shear rate on a surface pre-coated with 25 µg/mL VWFA1 (**b**, **c**) or 100 µg/mL Fg (**d**). $P$ values are the results of two-way ANOVA ($F$ values = 0.183, 13.16, 42.39 (**b**), 0.352, 3.345, 69.22 (**c**), 0.768, 6.518, 27.65 (**d**); degrees of freedom = 15, 5, 3 (**b**), 15, 5, 3 (**c**), 15, 5, 3 (**d**) for interaction, row factor and column factor, respectively) and multiple comparison. **e** BFP setup (*top*) and molecular binding illustration (*bottom*). **f–k** Adhesion frequency (Scatter plots with mean ± s.e.m.; $n ≥ 6$) (**f**, **i**), effective avidity and affinity (mean ± standard error; $n ≥ 6$) (**g**, **j**) and bond lifetime vs. force (mean ± s.e.m., $n ≥ 50$ for each data point) (**h**, **k**) of VWFA1- (**f–h**) or Fg- (**i–k**) coated beads binding to healthy young (HY; three subjects) or hypertensive (HTN; four subjects) subjects' platelets. Student's *t*-test results are annotated. **l** fBFP setup. **m** Representative time course of a hypertensive subject's platelet's Ca²⁺ level during repeated VWFA1 pulling at 40-pN clamping force. Peak increase $\Delta I_{max}$ is marked. **n** Intraplatelet $\Delta I_{max}$ (scatter plot with mean ± s.e.m.; $n ≥ 7$) of healthy young (left) and hypertensive (right) subjects' platelets during VWFA1 pulling at different clamping forces. $P$ values are results of two-way ANOVA ($F$ values = 5.601, 6.146, 63.34; degrees of freedom = 3, 3, 1, for interaction, row factor and column factor, respectively) and multiple comparison. **f, i, n** different colors indicate different subjects. **o–r** Representative flow cytometry histograms of E⁺ $\alpha_{IIb}\beta_3$ (**o**), total $\alpha_{IIb}\beta_3$ (**p**), Act. $\alpha_{IIb}\beta_3$ (**q**), and P-selectin (**r**) signals on healthy young (blue) and hypertensive (red) subjects' platelets. **s** Scatter plot with mean ± s.e.m. ($n = 5$ subjects) of flow cytometry MFI of total $\alpha_{IIb}\beta_3$, E⁺ $\alpha_{IIb}\beta_3$, Act. $\alpha_{IIb}\beta_3$, and P-selectin signals on healthy young (green) and hypertensive (red) subject's platelets. $P$ values, t ratios, and degrees of freedom (df) of Multiple *t*-test are annotated. **t** Proposed mechanisms of E⁺ $\alpha_{IIb}\beta_3$ over-expression in the biomechanical thrombi of hypertension patients.

P-selectin and Act. $\alpha_{IIb}\beta_3$ are widely used markers of platelet activation, but their performance in diagnosing thrombosis is unsatisfactory due to low sensitivity[72]. We show that E⁺ $\alpha_{IIb}\beta_3$ has a much better performance than P-selectin and Act. $\alpha_{IIb}\beta_3$ in correlating with the biomechanical thrombus size and in separating healthy young subjects and subjects carrying thrombotic risk factors. Furthermore, only E⁺ $\alpha_{IIb}\beta_3$, but not P-selectin or Act. $\alpha_{IIb}\beta_3$, was detected on platelets freshly isolated from hypertension patients. These results underscore the accuracy and sensitivity of E⁺ $\alpha_{IIb}\beta_3$ in detecting platelet hyperreactivity, suggesting its use as an independent biomarker for predicting arterial thrombosis in certain populations. To validate this application requires an investigation on the correlation between native E⁺ $\alpha_{IIb}\beta_3$ expression (assessed by flow cytometry) and thrombus size (assessed by thrombus profiling assay) in different patient cohorts.

Over the past decades, a routine has formed to evaluate the efficacy of new anti-thrombotic strategies solely based on thrombus size reduction and without considering inter-individual variability[11,73]. Our work demonstrates that thrombi possess multi-dimensional characteristics that can be orthogonal, which should be summarized as a "profile" or a "barcode". Because different individuals have differential personal thrombus barcodes, and different anti-thrombotics have differential effect barcodes, a treatment mismatch can easily occur. Conceptually distinguished from antiplatelet resistance, a drug with treatment mismatch is still effective in reducing the thrombus size, but has limited or undesired effects on changing the thrombus composition and/or activation status (Supp. Fig. 12). The life-threatening danger of treatment mismatch has been documented in multiple phase III trials where conventional integrin $\alpha_{IIb}\beta_3$ antagonists (e.g., orbofiban), despite high potency in inhibiting soluble agonist-induced platelet aggregation (and also biomechanical platelet aggregation as demonstrated in this work), paradoxically increased patient mortality by enhancing the risk of myocardial infarction[74,75]. It was later realized that the failure of these drugs was associated with their effect of stimulating integrin $\alpha_{IIb}\beta_3$ activation[76]. Addressing this issue, a chemical principle was recently discovered to develop anti-thrombotic candidates that lock integrin $\alpha_{IIb}\beta_3$ in the inactive state[77]. Developing diversified anti-thrombotics and determining their effect barcodes and their interactions with thrombosis-exacerbating factors can help avoid treatment mismatch, in which the "addition rule" could be helpful for prediction. The inter-individual variability in drug efficacy further urges the personalized selection of anti-thrombotics for treatment optimization.

## Methods

### Reagents

SZ22-FITC and P2-Alexa Fluor 488 (Beckman Coulter), Type I collagen, AK4-Alexa Fluor 647 and PAC-1-Alexa Fluor 647 (BioLegend), AK2, HIP-8-Alexa Fluor 488, Annexin V-Pacific Blue, Annexin V-Alexa Fluor 488, heparin, DiOC₆(3), and Alexa Fluor 405, 555, and 647 conjugation kits (Thermo Fisher Scientific), MBC 370.2 (Kerafast), fibrinogen (Innovative Research), NMC4, 2.2.9, LJ-P5, 152B6, LJ-155B39, LJ-134B29 and VWFA1[78] (MERU VasImmune), VWF monomer (Sino Biological), RU5 (Creative Biolabs), and PS-CNP beads (Bangs Laboratories) were purchased. 7E9, 7E3, and 10E5 were gifts from Barry S. Coller (Rockefeller University).

### Human subjects

All procedures involving human subjects were approved by the Institutional Review Board of the University of Texas Medical Branch (protocol number: 22-0015) and the University of Sydney (ethics reference number: 2023/582). Informed consent was obtained from all subjects to allow the publishing of data acquired from their blood samples and their demographic information that is relevant to research while protecting their privacy. All subjects were compensated for their participation.

Number ($n$) and age (mean ± s.d.) of subjects who participated in the thrombus profiling assay: healthy young: $n = 33$, age = 34.0 ± 6.3; healthy older: $n = 14$, age = 62.1 ± 8.9; hypertensive young: $n = 9$, age = 36.1 ± 8.7; hypertensive older: $n = 13$, age = 60.2 ± 9.4. All groups contained both male and female subjects with multiple races and both non-Hispanic/Latino and Hispanic/Latino ethnicities.

All hypertension patients were taking prescribed hypertension medications (e.g., prazosin, amlodipine, and enalapril). Patients taking other medications or under treatment for other diseases within 2 weeks before the blood draw were excluded from this study.

### Blood collection, reconstitution, and platelet isolation

For whole blood stenosis assay, blood was slowly drawn from the vein of a volunteer into a syringe pre-loaded with heparin (20 U/mL). In some control experiments, sodium citrate (4%) or EDTA (1.5 mg/ml) was used as the anticoagulant instead.

For laminar flow chamber assay, BFP assays, and flow cytometry, blood was drawn into a syringe pre-loaded with ACD buffer. Then blood reconstitution[78] was performed for laminar flow chamber assay to deplete plasma and reach a hematocrit of 45% and platelet count of 20,000 µL⁻¹. Or, platelet isolation was performed for BFP assays and flow cytometry[12], with platelets finally resuspended in modified Tyrode's buffer (135 mM NaCl, 11.9 mM NaHCO₃, 2.9 mM KCl, 0.42 mM NaH₂PO₄, 10 mM Hepes, 5.5 mM dextrose, and pH 7.4).

### Microfluidic device preparation

Polydimethylsiloxane (PDMS) was applied on a silicon mold (1-µm resolution), which was heated at 75 °C for 1 h for curing, peeled off, and cut into single pieces. Holes were drilled to create outlets and inlets. The devices then underwent plasma treatment and were bonded to glass coverslips.

### Microfluidic stenosis assay

Microfluidic channels were coated with VWF monomer (2 µg/mL) for 1 h. In some control experiments, the coating was done with 100 µg/ml

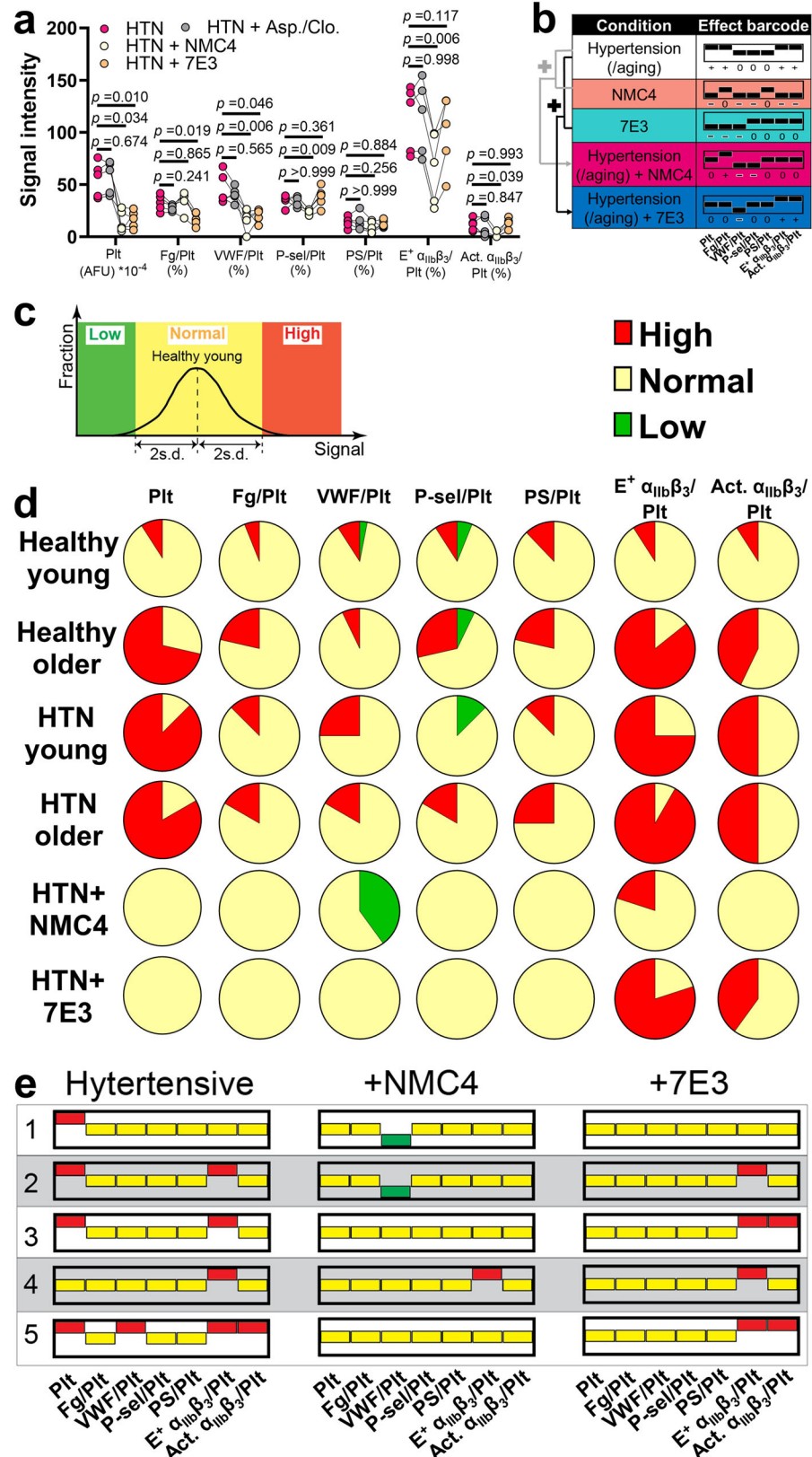

collagen instead. Blood was incubated with $DiOC_6(3)$ (5 µM) for 1 min, or with Sensor Set 1 (SZ22-FITC (0.5 µg/mL), Fg-Alexa Fluor 405 (60 µg/mL), 2.2.9-Alexa Fluor 555 (1 µg/mL) and AK4-Alexa Fluor 647 (1 µg/mL)), or Set 2 (SZ22-FITC (0.5 µg/mL), Annexin V-Pacific Blue (1 µg/mL), MBC 370.2-Alexa Fluor 555 (1 µg/mL), and PAC-1-Alexa Fluor 647 (1 µg/mL)) for 10 min, and perfused through the channel. Thrombus formation was observed using a Leica DM IL LED microscope (camera: Leica DFC360 FX; objective lens: air, 20×; acquisition software: LAS X). No bleed-through between fluorescence channels was observed. Platelet autofluorescence was detected in 391-nm channel[79], which was subtracted when calculating signals. Data analysis was performed using ImageJ 1.53 (Fiji, National Institutes of Health).

**Fig. 6 | Drug–disease interactions and personal thrombus barcodes. a** Individual point plot ($n = 5$; lines connecting points of the same subjects) of the thrombus profiles of hypertension patients without and with aspirin/clopidogrel (2×), NMC4 (IC50), and 7E3 (IC50) treatment. *P* values are the results of two-way ANOVA (*F* values = 12.59, 23.13, 34.27; degrees of freedom = 18, 6, 3, for interaction, row factor and column factor, respectively). **b** The seven-digit effect barcodes of hypertension without and with NMC4 or 7E3 treatment. A rule of addition is indicated. **c** Illustration of how values in the personal thrombus profiles being low, normal or high are defined. Thrombus profiles of healthy young subjects were used as the reference, the values of which are fitted to a Gaussian distribution. The mean ± 2 s.d. range is defined as normal, and values lower and higher are defined as abnormally low and high, respectively. **d** Fractions of abnormally high, normal, and abnormally low values in each dimension of the personal thrombus barcodes from healthy young, healthy older, hypertensive young, hypertensive older, hypertensive+NMC4, and hypertensive+7E3 groups. **e** Comparing the personal thrombus barcodes of hypertensive subjects without and with NMC4 or 7E3 inhibition. Blood samples from a total of five subjects was tested. For easier visualization, bars indicating "high", "normal", and "low" are respectively marked by red, yellow, and green.

In some experiments, different concentrations of aspirin (with 15 μg/mL defined as 2×) and/or clopidogrel (with 6 μg/mL defined as 2×) or ALB cocktail (1 U/mL apyrase, 100 mM MRS2179, 10 mM 2-MeSAMP, 10 μM indomethacin, 800 U mL⁻¹ hirudin) were added into blood to inhibit platelet amplification loops.

Hill equation was used to derive IC50 of inhibitors:

$$\text{Residue size} = R + (100 - R)/\left(1 + (\text{IC50}/C)^{\text{HillSlope}}\right) \quad (1)$$

wherein *C* is the inhibitor concentration, *R* is the residue size when the effect of the inhibitor saturates, and HillSlope is a constant.

For subjects who were tested multiple times, average values of these test results were used for data presentation and statistical analyses.

### Microfluidic laminar flow chamber assay
Reconstituted blood added with DiOC$_6$(3) (10 μM) was perfused at different shear rates over straight channels pre-coated with VWFA1 or fibrinogen. After 5 min, fluorescent signals from platelets were recorded at 40 frame s⁻¹. Data analysis was performed using ImageJ 1.53 (Fiji, National Institutes of Health).

### Biomembrane force probe (BFP) and fluorescence BFP (fBFP)
In a chamber filled with modified Tyrode's buffer + 0.5% BSA (plus 1 mM Ca²⁺/Mg²⁺ when interrogating platelets with Fg beads), a streptavidin-coated glass probe bead was glued to the apex of a biotinylated RBC, which is aspirated by a micropipette to form an ultra-sensitive force probe[52]. The probe bead was also coated with VWFA1 or Fg. On the opposing target side, a freshly isolated platelet was aspirated by a second micropipette, which was driven by a piezoelectric translator (Physical Instrument) to repeatedly bring the platelet in and out of contact with the bead to form adhesion events. The bead was monitored under an inverted microscope (IX83, Olympus) by a high-speed camera. A custom image analysis LabView (National Instrument) program tracks the bead position with 3 nm precision in real time. The BFP spring constant *k* was determined by the suction pressure inside the probe pipette and the geometric parameters of the force transducer assembly[80].

For adhesion frequency assay, the platelet was repeatedly brought into contact with the probe bead for 2 seconds and retracted. Adhesion events were signified by the elongation of the RBC upon platelet retraction, which yielded a tensile force signal on the bead. Adhesion and non-adhesion events in 30 cycles were enumerated to calculate adhesion frequency, $P_a$. The effective avidity ($A_cK_am_r$) and affinity ($A_cK_a$) were derived by the following equation[53],

$$P_a = 1 - \exp\{-m_r m_l A_c K_a\} \quad (2)$$

where $m_r$ and $m_l$ are the receptor and ligand surface densities derived from flow cytometry.

For the force-clamp assay, contact time was shortened until achieving infrequent (~20%) adhesion, which ensures that most (~90%) of the adhesion events are mediated by single receptor–ligand bonds. Once an adhesion event was observed, the platelet would be held at a desired clamping force to wait for the bond to dissociate[52]. Lifetime was determined as the time from the instant when the force reached the desired level to the instant of bond dissociation. The collected lifetimes were categorized into bins that cover successive force ranges. The average lifetime in each force bin was calculated to plot the "lifetime vs. clamping force" curve.

For fBFP, platelets were pre-loaded with Fura-2-AM and interrogated by VWFA1 beads with the force-clamp assay mode, but the contact time was kept at 2 s. Ratiometric imaging with a light source that alternates between 340 nm (to excite Ca²⁺-engaged Fura-2) and 380 nm (to excite Ca²⁺-free Fura-2) was used to measure the Ca²⁺ level in the aspirated platelet[55]. Matlab R2020b was used to analyze fluorescence images from fBFP experiments. Signal intensity from the 340-nm channel was divided by that from the 380-nm channel and then normalized by the average value of the first 10 frames to derive the normalized Ca²⁺ level.

### Flow cytometry assay
Platelet suspension was incubated with 2 μg/mL of HIP-8-Alexa Fluor 488, MBC 370.2-Alexa Fluor 555, PAC-1-Alexa Fluor 647, or AK4-Alexa Fluor 647 for 10 min, diluted with Hepes-Tyrode buffer by ten times, and immediately analyzed by flow cytometry.

### Statistical analysis
GraphPad Prism 10 was used for data plotting and statistical analysis. The statistical significance of the differences between the two groups was determined by a two-sided Student's *t*-test or multiple *t*-test. For the test of drug effects, multiple *t*-test assuming paired experimental design was used. For multi-group analysis, one-way or two-way ANOVA was used. When significant differences were shown, data was subjected to the Tukey test for multiple comparisons. A regression slope test was used to assess whether the slope of a linear fitting is significantly non-zero. Spearman rank correlation coefficient[46] and Kendall's tau correlation coefficient[47] were also used to test whether a positive correlation exists between different readouts of the thrombus profile. *P* values <0.05 were considered significant.

### Reporting summary
Further information on research design is available in the Nature Portfolio Reporting Summary linked to this article.

## Data availability
All data supporting the findings of this study are available within the article and its supplementary files. Source data of fluid dynamics simulation generated in this study have been deposited in the Harvard Dataverse repository under accession code (https://doi.org/10.7910/DVN/D4SJIP). Source data of human blood sample experiments are protected and unavailable for public deposition or upon request in accordance with the signed consent of study subjects. Any additional requests for information can be directed to, and will be fulfilled by, the corresponding authors. Source data are provided with this paper.

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

## Acknowledgements

We thank B. S. Coller (Rockefeller University) for sharing precious reagents, F. Ola-Daniel, Y. Wang (The University of Texas Medical Branch), and A. Dupuy and Y. C. Zhao (The University of Sydney) for technical support, and Z. M. Ruggeri (The Scripps Research Institute) for providing valuable suggestions. This work was supported by the following funding sources: National Heart, Lung, and Blood Institute grant R00HL153678 (Y.C.). National Institute on Aging the Claude D. Pepper Older Americans Independence Center Award #P30-AG024832 (Y.C.). UT System Rising STARs award (Y.C.). American Heart Association Postdoctoral Fellowship 20POST35080023 (Y.C.). National Institute of General Medical Sciences grant 1R01GM152812 (L.X.). National Science Foundation grants DMS-1953189, CCF-2007823, and DMS-2210775 (L.X.). MRFF Cardiovascular Health Mission Grants APP2016165 and APP2023977 (L.A.J.). National Heart Foundation Future Leader Fellowship Level 2 (105863) (L.A.J.). Snow Medical Research Foundation Fellowship 2022SF176 (L.A.J.).

## Author contributions

Methodology: M.D., S.P., R.-G.X., F.Z., Z.C., and Y.C. Experiments: M.D., S.P., S.U., B.C., N.A.Z.A., A.S., Y.C.C., R.G., S.R., A.B., and Y.C. Data analysis: M.D., S.P., S.U., B.C., N.A.Z.A., A.S., Y.C.C., S.R., A.B., H.Y., L.X., R.-G.X., F.Z., J.F., and Y.C. Donor recruitment and blood collection: M.D., S.P., S.U., A.B., and M.B. Funding acquisition: L.X., L.A.J., and Y.C. Supervision: L.A.J. and Y.C. Writing: M.D., H.Y., R.-G.X., F.Z., N.A.Z.A., M.M., L.H., Z.F., C.Z., M.B., L.X., L.A.J., and Y.C.

## Competing interests

The authors declare no competing interests.

## Additional information

Yunfeng Chen.

**Peer review information** *Nature Communications* thanks Tzung Hsiai,
and the other, anonymous, reviewer(s) for their contribution to the peer
review of this work. A peer review file is available.

[1]Department of Biochemistry and Molecular Biology, The University of Texas Medical Branch, Galveston, TX 77555, USA. [2]Department of Pathology, The University of Texas Medical Branch, Galveston, TX 77555, USA. [3]Department of Statistics, The Pennsylvania State University, University Park, Pennsylvania, PA 16802, USA. [4]Division of Thoracic Surgery, Brigham and Women's Hospital, Harvard Medical School, Boston, MA 02115, USA. [5]School of Biomedical Engineering, The University of Sydney, Darlington, NSW 2008, Australia. [6]Charles Perkins Centre, The University of Sydney, Camperdown, NSW 2006, Australia. [7]Heart Research Institute, Newtown, NSW 2042, Australia. [8]The University of Sydney Nano Institute (Sydney Nano), The University of Sydney, Camperdown, NSW 2006, Australia. [9]Coulter Department of Biomedical Engineering, Georgia Institute of Technology, Atlanta, GA 30332, USA. [10]School of Integrative Medicine, Shanghai University of Traditional Chinese Medicine, Shanghai 201203, China. [11]Department of Immunology, School of Medicine, UConn Health, Farmington, CT 06030, USA. [12]Department of Mechanical Engineering, The City University of New York - City College, New York, NY 10031, USA. [13]Department of Internal Medicine, The University of Texas Medical Branch, Galveston, TX 77555, USA. ✉e-mail: yunfchen@utmb.edu

