## [Peer Review File · Nature Communications]

REVIEWER COMMENTS

Reviewer #1 (Remarks to the Author):

The authors presented a microfluidic study to characterize the molecular participants and regulators in thrombosis under pathological stenotic conditions (high shear stress) using fluorescence microscopy. The biomembrane force probe (BFP) and fluorescence BFP were also used to measure the molecular kinetics to correlate the specific changes in platelet aggregation with the molecular attributes under health and abnormal (hypertension and aging) conditions. Furthermore, the authors group a set of thrombotic fluorescence signals of specific molecular “sensors” into a “barcode” to explore its clinical implications in assessing the patient risks and screening anti-thrombotic drugs. Although the study is very rich in data, there are some fundamental flaws regarding the design of the micro-device and its utility in addressing real human pathophysiology. Also, the novelty of this paper seems incremental as it mostly confirms existing findings or generates combined observations previously reported by different studies. Moreover, the mechanistic studies and the insights generated by this study are very specific to the thrombosis field, but they lack deeper mechanisms that can be generalizable to other fields.

Major comments:

1. Pathophysiology relevance: The microfluidic assay used in this study forms platelet aggregates at the downstream side of the stenosis where the shear stress is much lower than that at the top. In fact, the location of the platelet aggregation under pathological in vivo conditions most occurs at the apex of the stenosis where the shear is the highest. Since all the signal intensities and analyses were based on this downstream ROI instead of the top, it is unclear whether the platelet aggregation characterized by the authors truly reflects the platelet aggregation that causes human arterial occlusion or not.
2. Hemodynamics in humans: Separately, the Reynolds number in microfluidics is much lower than that in human arterial stenosis due to the differences in lumen size. As a result, the flow in the microfluidic always remains in the Stokes regime, which is again different from the human hemodynamic condition in arterial stenosis, where flow separation and even turbulence can occur. How would these misrepresentations justify the biorheological relevance of the device.
3. Barcode: All biochemical reactions have been shown before. What makes it new here by combining them into a barcode. It is unclear what new information the so-called “barcode” gives and how this can impact clinical decisions. Moreover, what makes knowing all these molecular reactions useful while platelet aggregation is all that is needed to block a vessel.
4. Bleeding risks: Inhibiting platelet aggregation has not been a problem, as the biochemical axes

that control aggregation have been known for decades, and many potential drug candidates have been identified as being able to do so. How to inhibit platelet aggregation while preserving low bleeding risks is, in fact, the real challenge, while this paper and the device developed do not seem to address this issue.

5. Overstatements: The paper contains many overstatements that are not aligned with the status quo of the field. To name a few:

The authors claim that “As a symbolic feature of arterial thrombosis, severe stenosis in the blood vessel creates a high-shear, high-gradient flow However, no approach is currently available to comprehensively characterize thrombogenesis under this biorheological condition.” It has been well documented by many groups such as Brass, Diamond, Ku, Neelamegham, Neeves, and Lam, etc., with many microfluidic designs, including ones with multicolor staining that stenosis creates a high-shear condition and subsequently leads to platelet aggregation that can eventually block the lumen. A comprehensive understanding of the molecular participants in causing platelet aggregation has been largely understood.

Also, the authors claim, “However, whether abnormal biomechanical platelet aggregation is intercorrelated with higher risks of arterial thrombosis in humans was never studied.” which is completely incorrect, as this has been known for decades. Moreover, hypertension and aging have been known for decades to involve higher risks for arterial thrombosis with exacerbated platelet aggregation.

Another one: “For example, platelets in biomechanical platelet aggregation are crosslinked via three molecular interactions: glycoprotein (GP) Iba–von Willebrand factor (VWF), integrin $\alpha\text{IIb}\beta\text{3}$ –VWF and integrin $\alpha\text{IIb}\beta\text{3}$ –fibrinogen (Fg)^{10,12,14}. Their respective contribution to biomechanical thrombogenesis and how they are affected by pathological conditions remains unclear. Jackson (2007) documented this almost 20 years ago, not to mention the recent advances in this area.

6. Broader impact: the study focuses heavily on the biomolecular aspects of platelet aggregation and thrombosis. Although it is good to know how each molecular axis can be separately inhibited and alter platelet behavior, there is a lack of novel and exciting findings that can be translated to other areas, lacking excitement to the general audience.

Additional comments:

- This device is similar to many microfluidic devices that feature (10) individual channels, where each channel needs to be perfused separately. How would such a typical design make this device a high-throughput one? Can you explain what high throughput is vs. the opposite?
- Are all the reagents commercially available? How expensive and how much time does it cost to obtain one useful barcode? Would this be clinically acceptable compared to other platelet functional assays available in the market, such as PFA-100, T-TAS, etc.?

- Physiologically, the thrombotic surface is rich in collagen. Why was the VWF-A1 monomer used for the channel coating instead? How is this justifiable pathophysiologically? Why not use VWF multimer or collagen? How would the results change if VWF multimers or collagen were used?
- The blood was anticoagulated by heparin, while many studies use citrate or EDTA. Can you justify why heparin was used? Can you rule out the anticoagulant dependences?
- It seems that the barcode was obtained based on the changing trend of the fluorescence signal intensity (FSI). It is, however, known that the FSI can be highly noisy and is more of a qualitative metric. Also, some of the FSI features very small changes in magnitude, although being statistically significant (e.g., fig 3E-Act, 3D-fg). How did the authors decide on the barcode specifically? Manually? Automatically based on some threshold? Does a small change make a difference in deciding therapy while statistically being a significant difference?

Reviewer #2 (Remarks to the Author):

The authors present interesting and significant findings on the interactions between anti-thrombotic inhibitors and hypertension, as well as the inter-individual variability in personal thrombus profiles. However, the research could benefit from a more defined focus and clearer conclusions. The manuscript can be strengthened by improving clarity and organization.

1. Consider redefining the title to articulate the novelty and focus of the experimental data. If microfluidics are central to your methods, explicitly include "via microfluidics" in the title. The use of terms like "uncover" and "mechanobiology" might be too broad and non-specific for this investigation. In addition, please find a better way to define your cohort of hypertension and age-related thrombosis.
2. Could you succinctly describe how the study quantifies the impact of biomechanical forces on thrombus formation? Additionally, please provide a brief overview of the specific interactions studied and their distinct influences on the thrombus.
3. Please include citations to substantiate the claim that the biomechanical aspect of platelet aggregation (driven by mechanical forces) has not been thoroughly studied in its correlation with increased risks of arterial thrombosis. Additionally, please elaborate on how interactions involving GPIIb/IIIa–von Willebrand factor (VWF), integrin α IIb β 3–VWF, and integrin α IIb β 3–fibrinogen distinctly contribute to thrombogenesis.
4. In Figure 1, the results indicate initial rapid thrombogenesis followed by slower progression, with signal quantification at 450 seconds. Please explain the rationale behind selecting this specific time point for signal quantification. What do the observed changes in the rate of thrombogenesis imply about the underlying mechanisms of thrombus formation under biomechanical conditions?

5. In Figure 2, the findings indicate that only AK2 and NMC4, which inhibit the GPIIb/IIIa-VWF interaction, completely halted thrombogenesis. Could you discuss the underlying mechanisms that render this interaction critical for initiating biomechanical platelet aggregation?
6. The inhibition of integrin α IIb β 3 interactions with VWF and Fg significantly, albeit variably, affected thrombus size. Particularly with the potent effects of 7E9, could you discuss how these interactions contribute to thrombus structure and growth? What does the variable impact on thrombus size and composition suggest about their roles in thrombogenesis?
7. Your study introduces an “effect barcode” system to depict the influence of different factors on the thrombus profile. How does this system aid in interpreting data and informing clinical or therapeutic decisions?
8. In Figure 3, the results support an “addition rule” where the combined effects of inhibitors can be predicted by summing the effect barcodes of individual inhibitors. Could you elaborate on the mechanistic or biochemical foundations supporting this rule?
9. In Figure 4, the study notes that both aging and hypertension are associated with increased thrombus size, fibrinogen (Fg) levels, and activation of integrin α IIb β 3, without corresponding changes in hematocrit or platelet count. Could you elaborate on the potential physiological or cellular mechanisms that account for these specific increases in older and hypertensive individuals? What implications might these findings have for our understanding of the underlying processes of thrombogenesis in these populations?
10. In Figure 5, despite observing significant impacts of hypertension on thrombus profiles, the study found no significant correlations between thrombus size or E+ α IIb β 3 levels and other cardiovascular risk factors such as HbA1C, BMI, or lipid levels. How do you interpret these findings within the broader context of cardiovascular health?
11. The results suggest that hyperactivity of GPIIb/IIIa and integrin α IIb β 3 leads to increased platelet activation in hypertension due to enhanced mechano-signaling. How might these changes impact the overall hemostatic balance and risk of thrombus formation in hypertensive patients?
12. The discussion emphasizes the unique capabilities of the thrombus profiling assay in evaluating biomechanical platelet aggregation. Could you discuss how this assay compares with conventional methods regarding sensitivity, specificity, and predictive accuracy for clinical outcomes? What are some potential limitations or challenges that could be encountered when integrating this assay into routine clinical practice?
13. The discussion notes significant contributions from the hyperactivity of GPIIb/IIIa and integrin α IIb β 3 to thrombosis in hypertension patients. How does this hyperactivity compare to other recognized mechanisms of thrombosis in hypertension?
14. E+ α IIb β 3 has been identified as a potential biomarker for arterial thrombosis due to its strong

correlation with thrombus size and prevalence in hypertension patients. What steps are necessary to validate E+ α IIb β 3 for clinical application? Additionally, how does this marker compare with others such as P-selectin and Act. α IIb β 3 in terms of diagnostic accuracy and prognostic value?

Reviewer #3 (Remarks to the Author):

The manuscript by Din et al. proposes a new methodology to analyze a growing thrombus in a stenotic microfluidic model, specifically by looking at several of the key players in a growing clot, such as platelets, fibrinogen, integrin α IIb β 3 conformation, and VWF. In addition, this work further investigates an innovative concept related to biomechanical platelet aggregation, a unique phenomenon in which shear and GPIIb α signaling leads to thrombus growth primarily through an intermediate affinity of α IIb β 3. Uniquely, thrombus growth is not affected by conventional antiplatelets or amplification loop blockers. Here, the phenomenon is further explored with a series of inhibitor studies, and then further shown to have potential clinical relevance via a series of studies on clinical samples from hypertensive patients, both with their novel device showing differences in stenotic clot formation, then with follow-up mechanistic studies using a biomembrane force probe, which is a technically challenging approach.

This is an important contribution for several reasons. First, the assay itself represents a more holistic approach to examining clot formation, showing interesting differences in the relative ratio of key components and platelet integrin states. Second, this assay provides new insights into the platelet reactivity of hypertensive patients, and highlights a potential need to explore new antiplatelet strategies. Finally, the tool developed here may find use in helping to screen new antiplatelet approaches, as well as serve as a tool to tailor individual anticoagulant strategies to patients, although a significant development effort would be needed for clinical use.

Owing to the rigor and well-designed studies, I have minor comments aimed at improving the accessibility of the manuscript:

1. If there is space in Figure 2, I wonder if panel B could be moved below panel A and a small table with more specific details on what is targeted by the inhibitors could be added. My only reason for asking is that I was initially confused as to why NMC4 targets VWF on the GPIIb α axis, and 152B6 targets VWF on the α IIb β 3 axis. This was subsequently cleared up when reading lines 154-160, but I found myself creating a table to keep track of the various inhibitors.

2. Lines 205-207: I also agree that the barcode summation idea is interesting. In this second example of addition, one of the addends (GPIIb α -VWF) equals the polystyrene barcode, so it feels akin to saying $A + 0 = A$ (although that is a simplification). Noting that this concept is revisited in Figure 6, I recommend providing a hint to readers that this concept will be further tested later in the paper. (If a reader linearly examines the manuscript, this concept may come across as unconvincing at this point)

3. In Figure 4, I found the labeling related to “Slope non-zero?” a bit confusing at first. I now understand that the null hypothesis is that the slope was zero, but perhaps other wording may help with this.

4. In Figure 4H, I expected the number of healthy young donors to match 4A, were these separate studies?

5. In Figure 5, I understand the logic that in the flow chambers, there is not a large difference between hypertensive young/old platelet behaviors. However, Figure 4 shows that there are likely differences between these groups. I would suggest qualifying that hypertensive patients were not tested separately, but that an effect may exist that could be identified with later studies.

6. In the BFP and fBFP experiments, I had a hard time telling if each dot represented a single platelet from a single donor, or a single platelet from the same donor. Could you clarify?

7. Figure 6C – the idea of personal thrombus barcodes is very interesting, and I appreciate the rigor in showing the proportion of patients that fit the proposed schema of “normal” values. With that said, panel c was a bit difficult to understand at first glance. My suggestions would be to consider a prior panel visually illustrating how high, normal, low are defined, and also including the number of patients represented by these tests (I believe it’s 32, but don’t see it in the figure or caption). Other possibilities include a title for panel c to orient the reader (akin to those in Figure 5), and using the labels healthy, healthy older, instead of HY, HO etc.

8. Methods: I may have missed this, but what are the dimensions of the microfluidic structure, I was specifically looking for the height of the channels, although also providing more details on the geometry tested would help others seeking to replicate these results.

We thank the reviewers for their very valuable and helpful comments. Below please find our point-to-point responses and revisions of the manuscript. Actions taken to address the comments are summarized in purple. Modifications made in the revised manuscript are highlighted in red.

In addition, we also made the following improvements to the manuscript.

- 1) According to the policy of *Nature Communications*, we have revised the format of several figures from bar graphs and boxes and whiskers to scattered points. We have also indicated the p -values and other results of all statistical analyses either as annotations in the graph and/or in the figure legend.
- 2) To make our work more solid, we recruited four more healthy older subjects and performed the thrombus profiling assay for their blood samples. The new data were added to Figure 4a,c,e,g-l, Figure 6d, Supplemental Figure 5b,c, Supplemental Figure 6a-f and Supplemental Figure 11. Statistical analysis results were updated accordingly in the main text, Figure 4d and Supplemental Table 2. No conclusion was affected.
- 3) We improved the data presentation in Figure 6d by showing how aging impacts the thrombus profiling of hypertensive subjects and *vice versa*, how hypertension impacts that of older subjects. The new data presentation demonstrates a bi-directional cooperation between hypertension and aging in enhancing the thrombus size and $E^+ \alpha_{IIb}\beta_3$ level, which better explains the synergy effects between these two risk factors.
- 4) We improved the statistical analyses used in investigating drug effects (Figures 2 and 3) by allowing the pairing of data before and after drug treatment. This did not change any “significance vs. insignificance” conclusion, but only resulted in minor updates in the level of significance (i.e., the number of ‘*’) of certain results in Fig. 2f (P-sel.), 2g (VWF, P-sel., $E^+ \alpha_{IIb}\beta_3$), 2h (VWF), 2i (VWF) and 2l (Fg).

Reviewer #1 (Remarks to the Author):

The authors presented a microfluidic study to characterize the molecular participants and regulators in thrombosis under pathological stenotic conditions (high shear stress) using fluorescence microscopy. The biomembrane force probe (BFP) and fluorescence BFP were also used to measure the molecular kinetics to correlate the specific changes in platelet aggregation with the molecular attributes under health and abnormal (hypertension and aging) conditions. Furthermore, the authors group a set of thrombotic fluorescence signals of specific molecular “sensors” into a “barcode” to explore its clinical implications in assessing the patient risks and screening anti-thrombotic drugs. Although the study is very rich in data, there are some fundamental flaws regarding the design of the micro-device and its utility in addressing real human pathophysiology.

Also, the novelty of this paper seems incremental as it mostly confirms existing findings or generates combined observations previously reported by different studies. Moreover, the mechanistic studies and the insights generated by this study are very specific to the thrombosis field, but they lack deeper mechanisms that can be generalizable to other fields.

Response to overall comments: thank you for recognizing the rich data in our work. The novelty of our work can be summarized as follows:

Firstly, no bioassay is currently available in the standard clinical settings to evaluate the risk of arterial thrombosis in general populations. Although much effort was made to fill this gap, the mechanisms of action of recently developed lab and clinical assays still focus on coagulation and soluble agonist-induced platelet aggregation, but not the biomechanical factors of arterial thrombosis. On the other hand, previous works have used microfluidic stenosis assays to examine shear-induced platelet aggregation. However, while a couple of works checked the potential of these assays in assessing bleeding risks in patients with von Willebrand disease and Hermansky–Pudlak Syndrome^{1,2}, no work ever looked at the contribution of shear-induced platelet aggregation to higher thrombotic risks in human. Our successful detection of intensified biomechanical thrombogenesis in hypertensive and/or aged individuals endorses the potential of our thrombus profiling assay in clinical practice for evaluating thrombotic risks, and also emphasizes ‘shear force’ as an indispensable element in evaluating risks of arterial thrombosis.

Secondly, although it has been identified for a long time that platelets in patients with hypertension, diabetes, aging, cancer, COVID-19, etc., manifest hyperreactivity, the vast majority of previous works only demonstrated abnormal platelet response to soluble agonist (e.g., collagen, thrombin, ADP) and/or baseline platelet activation reflected by P-selectin expression and integrin activation³⁻⁶. To our knowledge, only two papers reported platelet hyperreactivity associated with mechanobiology. The first work reported abnormal activity of Piezo1, a mechanosensitive ion channel, associated with hypertension⁷. However, its exclusive use of mouse blood and mice in the whole paper compromises the human relevance of its conclusion. Moreover, as in many other papers, the mechanistic investigations in this work only stimulated platelets with soluble agonists and observed platelet signalling and aggregation under a normal pressure environment. Thus, we cannot exclude the possibility that the intensified contribution of Piezo1 to platelet hyperactivity observed in this work is because of the dysfunction of biochemical pathways that involves Piezo1, instead of the dysfunction of any biomechanical mechanism. The second work, which was done by some of the authors of the current manuscript, discovered intensified shear-driven thrombus formation in diabetic mice associated with a higher sensitivity of platelets to compressive forces⁸. The paper stated that the compressive forces are likely introduced by frequent collisions

of RBCs in the blood flow during platelet translocation, which is in the phase of platelet recruitment prior to platelet aggregation. Also, the work did not identify the receptor of the compressive forces on platelet surface and thus did not provide the molecular mechanism underlying the discussed mechanosensing. Distinct from all previous studies, our results on hypertension patients and older people for the first time uncovered intensified biomechanical thrombogenesis as an essential contributor to both the high risk of CVD and antiplatelet resistance, and we also unravelled the hyperactivity of GPIIb/IIIa-integrin $\alpha_{IIb}\beta_3$ mechanosensing axis as the underlying molecular mechanism in hypertension.

Thirdly, our observations of the 'treatment mismatch' and inter-individual variability point out the possibility that the conceptualized anti-thrombotic "holy grail" that adequately treats arterial thrombosis in all patients, which has been pursued by the field for decades, may not exist. Instead, the best-matching drug/regimen likely varies from patient to patient. This new perspective challenges the current mainstream strategy of anti-thrombotic drug development which uses only healthy young animals for drug testing and only measures reduction of the thrombus size for drug evaluation, and advocate for the regular use of diseased animal models as well as the application of 'thrombus profile' for evaluating existing and new anti-thrombotic drugs.

Major comments:

1. Pathophysiology relevance: The microfluidic assay used in this study forms platelet aggregates at the downstream side of the stenosis where the shear stress is much lower than that at the top. In fact, the location of the platelet aggregation under pathological in vivo conditions most occurs at the apex of the stenosis where the shear is the highest. Since all the signal intensities and analyses were based on this downstream ROI instead of the top, it is unclear whether the platelet aggregation characterized by the authors truly reflects the platelet aggregation that causes human arterial occlusion or not.

Thank you for pointing this out. What we meant to state was that the main body of the platelet aggregates have the tendency to grow on the downstream side of the stenosis, which is a natural consequence of the high shear flow and it also agrees with the depiction of arterial thrombosis in previous papers (Fig. R1)^{9,10}. However, the aggregates did not exclusively exist at the downstream side of the stenosis without covering the apex of the stenosis. Instead, most thrombi covered the whole apex, extending all the way from the upstream side to the downstream side. When we analyzed the data, the ROI did include the whole thrombus, covering the top, upstream and downstream parts.

Figure R1. Depiction of arterial thrombosis in previous papers: (a) Ref⁹; (b) Ref¹¹; (c) Ref¹⁰.

To address your concern and confirm our observations, we randomly selected 30 experimental runs and inspected the contours of the resulting thrombi, which are summarized in Figure R2. While most thrombi tend to grow towards the downstream side, this tendency is not dominant in many cases. This is reflected by the facts that most thrombi (>85%) cover the whole stenosis apex, and that most (>85%) thrombi have the point in their contour most close to the opposing channel wall positioned above the stenosis apex, suggesting that the stenosis apex is still the place where occlusion first occurs as the thrombi develop larger.

Figure R2. Representative pseudo-colored images of 30 healthy young subjects' thrombi formed in the stenotic channels, with the margins of the channel marked by white lines and curves. Among the thrombi, 26 out of 30 (87%) thrombi have a tendency of growing towards the

downstream side of the stenosis (surrounded by red and blue dashed lines), which include 22 (73%) that cover the whole stenosis apex and 4 (13%) that do not, while the remaining 4 (13%) thrombi have the main body right on the apex or at the upstream side of the stenosis (surrounded by green dashed lines). A total of 26 (87%) thrombi have the point in their contour most close to the opposing channel wall (marked by yellow arrows) positioned above the apex of the stenosis (surrounded by red and green dashed lines), while 4 (13.3%) thrombi have their part most close to the opposing channel wall at the downstream side of the stenosis apex (surrounded by blue dashed lines). A total of 26 out of 30 (87%) thrombi cover the whole stenosis apex.

Action taken:

1) We have added Fig. R2 into the revised manuscript as the new Supp. Fig. 2, and added the following statements in Results:

“Due to the high shear force, most thrombi have a tendency of growing towards the downstream side of the stenosis. Nonetheless, most thrombi (>85%) cover the whole stenosis apex, and most (>85%) thrombi have the point in their contour most close to the opposing channel wall positioned above the stenosis apex (Supp. Fig. 2), making the stenosis apex still the most vulnerable position for occlusion.”

2) We updated the drawings in Fig. 1c,e to reflect the most common shape of the biomechanical thrombi. The following sentence in the legend of Figure 1:

“When blood is perfused over, platelets spontaneously aggregate at the downstream side of the hump.”

was revised to:

“When blood is perfused over, platelets spontaneously aggregate **around the hump.**”

2. Hemodynamics in humans: Separately, the Reynolds number in microfluidics is much lower than that in human arterial stenosis due to the differences in lumen size. As a result, the flow in the microfluidic always remains in the Stokes regime, which is again different from the human hemodynamic condition in arterial stenosis, where flow separation and even turbulence can occur. How would these misrepresentations justify the biorheological relevance of the device.

Indeed, it was shown by simulation that coronary arteries can reach a high Reynolds number and could develop turbulence with 70% or above stenosis¹², and *in vivo* experiments demonstrated that turbulence can have a positive effect on thrombus formation¹³. It was postulated that turbulence contributes to arterial thrombosis by providing extra mechanical forces¹⁴. On the other hand, how flow separate contributes

to thrombosis is unclear. Following your suggestion, we calculated the Reynolds number in our channel at the site of stenosis to be 3.73, which is certainly lower than that in large human arteries, e.g., aorta.

However, it is important to note that the size of blood vessels also varies tremendously between different animal species: the diameter of carotid artery is ~0.7 mm in mice and ~1 mm in rats but can reach ~8 mm in human¹⁵. Thus, the Reynolds numbers in the arterial blood flow of mice and rats *versus* human are also greatly different. Yet, this does not impede the wide application of mouse and rat models for providing valuable information regarding vascular health and diseases with close human relevance and for evaluating anti-thrombotic strategies and drugs.

Furthermore, due to the great size diversification of different human arteries (diameter: roughly 1-20 mm) and arterioles (diameter: 15-240 μm), the Reynolds number of the blood flow in different arteries and arterioles in fact varies drastically, ranging from ~1 to approximately 4,000. Even within different carotid arteries, the Reynolds number can range from tens to 4,000^{12,16}. Considering that a Reynolds number of 2,000 is generally considered the threshold for determining laminar *versus* turbulent flow, and a Reynolds number in the scale of tens would start to allow flow separation, the high variability of Reynolds number in human arteries and arterioles indicates high variability in the flow pattern of blood in different arteries and arterioles. Yet, thrombosis can technically occur in all arteries and arterioles. Therefore, it is impossible to use a single flow pattern to represent all the hemodynamic conditions in arterial thrombosis, or use a single setup to recapitulate hemodynamic conditions in all arteries and arterioles.

While it is true that the Reynolds number in microfluidic systems is lower compared to large human arteries due to differences in lumen size, and we sincerely appreciate you for pointing this out, our device is mainly designed to focus on the biorheological conditions most critical for platelet aggregation and thrombus formation, namely, high shear flow. As stated in our original manuscript, our device recapitulates the hemodynamic environment of typical arteries and arterioles in the sense that, the wall shear stress (WSS) it creates at the stenosis site is comparable to that in human arterioles and mouse arteries, as well as in large human arteries during systole^{15,17}. Although this wall shear rate is higher than the average wall shear rate of large human arteries ($250\text{-}1,000\text{ s}^{-1}$)¹⁵, its resulted extra high shear rate at the site of stenosis would create extra shear stress that to some extent compensates the missing shear stress that ought to be provided by turbulence in large arteries. Of course, this point would require more investigations because it remains unclear exactly how much force is generated by the turbulence. On the other hand, we also compared the Reynolds numbers in channels with the same cross-sectional area but different shapes, which are within the same scale. At the inlet site with no stenosis, the Reynolds number is 0.73 for a rectangular channel (adopted by our device) and 0.71 for a circular (shape of arteries

and arterioles) channel. At the site of stenosis, the Reynolds number is 2.01 for a rectangular channel and 1.15 for a circular channel. These comparisons confirm that the shape of the channel does not incur a great change in the Reynolds number.

Creating turbulent flow in microfluidic channels either requires extremely high perfusion rates which are non-physiological, or requires the channel geometry to contain highly non-smooth components¹⁸, which is difficult to be incorporated into our current design, and also lacks the reference of how much turbulence is considered physiological. Considering that turbulence contributes to arterial thrombosis by providing extra mechanical forces¹⁴, we reason that if we can confirm that an increase or decrease of shear stress within a certain range does not compromise the capability of our setup in assessing the shear-driven platelet “aggregatability” of blood samples, then our assay should have general biorheological relevance. Therefore, we performed new experiments with varied perfusion rates to change the shear stress inside the channel. Our results showed that lowering the blood perfusion rate from 18 to 13.5 $\mu\text{l}/\text{min}$ or increasing it to 27 and 36 $\mu\text{l}/\text{min}$ (which respectively changed the shear stresses to 0.75, 1.5 and 2 times of the original) did not significantly affect the thrombus profiling outcome (Fig. R3). Furthermore, despite variations in the perfusion rate, we could consistently observe significantly larger thrombus size (Fig. R3a) and higher E^+ and $\text{Act. } \alpha_{\text{IIb}}\beta_3$ levels (Fig. R3f,g), marginally higher Fg level (Fig. R3b) but comparable VWF, P-selectin and PS levels (Fig. R3c-e) in the thrombi of hypertensive young subjects than healthy young subjects. These results validated our hypothesis, and confirmed the robustness of our assay.

Figure R3. Testing the effect of changing flow perfusion rate on the thrombus profiling results. The perfusion rate was changed from the original 18 $\mu\text{l}/\text{min}$ (data acquired from Fig. 4c; $n=33$ for healthy young and $n=9$ for hypertensive (HTN) young) to 13.5, 27 and 36 $\mu\text{l}/\text{min}$,

respectively. Then total platelet signal intensity (a) and the normalized signal intensity of Fg (b), VWF (c), P-selectin (d), PS (e), E⁺ $\alpha_{IIb}\beta_3$ (f) Act. $\alpha_{IIb}\beta_3$ (g) were acquired from n =5 healthy young (*left, green points*) and n =5 HTN young (*right, magenta points*) subjects and presented as scatter plots with mean \pm s.e.m.. *P*-values annotated on the graphs are results of two-sided *t*-tests comparing the result of each HTN young group with that of the healthy young group with the identical perfusion rate. One-way ANOVA was performed to compare the results acquired under different perfusion rates, with the outcome annotated on the graphs (green for healthy young, magenta for HTN young).

Action taken:

1) We updated Supp. Fig. 1f to include the calculated Reynolds numbers, described the calculation method in the Supplementary Methods section, and added the following statement to the first paragraph of Results:

“The calculated Reynolds numbers in channels of different shapes are also within the same scale (Supp. Fig. 1f).”

2) The new experimental results (Fig. R3) were added to the revised manuscript as new Supplemental Figure 8. The following description was added to the Results:

“Due to size variations, different human arteries and arterioles have distinct Reynolds numbers (affecting flow patterns such as laminar *versus* turbulent) and shear rates in the blood flow^{12,16}, together resulting in a certain extent of diversification in the shear stress. However, changing the perfusion rate in our assay from 18 to 13.5, 27 and 36 μ l/min (respectively changing the shear stresses to 0.75, 1.5 and 2 times of the original) did not significantly affect the thrombus profiling outcome (Supp. Fig. 8). Furthermore, despite variations in the perfusion rate, we could consistently observe significantly larger thrombus size (Supp. Fig. 8a) and higher E⁺ and Act. $\alpha_{IIb}\beta_3$ levels (Supp. Fig. 8f,g), marginally higher Fg level (Supp. Fig. 8b) but comparable VWF, P-selectin and PS levels (Supp. Fig. 8c-e) in the thrombi of hypertensive young subjects than healthy young subjects. These results validated that our assay can assess the general shear-driven platelet “aggregatability” of blood samples instead of being limited to representing only specific vessels.”

3) We sincerely appreciate your valuable comments and acknowledged the limitations inherent in microfluidics in the Discussion:

“As a limitation, our assay cannot recapitulate the biomechanical scenarios of thrombosis in all different arteries and arterioles, especially in large arteries where the Reynolds number can be sufficiently high to trigger turbulence¹². Nonetheless, by replicating critical aspects of thrombus formation under high shear conditions, the assay

was validated to allow the evaluation of the general prothrombotic tendency of blood samples.”

4) To ensure the rigor of our work, we updated our software version of COMSOL Multiphysics from 5.6 to 6.0 and refined our fluid dynamics simulation model, and re-ran all the fluid dynamics simulations. The newly derived values have trivial differences than the original values (e.g, WSS at stenosis site changes from 860 to 857 dyn cm⁻²), which do not affect any conclusion in the paper. We updated Supp. Fig. 1f with the new values and updated the version number of COMSOL Multiphysics in Supplementary Methods.

3. Barcode: All biochemical reactions have been shown before. What makes it new here by combining them into a barcode. It is unclear what new information the so-called “barcode” gives and how this can impact clinical decisions. Moreover, what makes knowing all these molecular reactions useful while platelet aggregation is all that is needed to block a vessel.

Indeed, that “platelet aggregation is all that is needed to block a vessel”, or in other words, “size is the only important parameter in thrombi”, represents the mainstream viewpoint in the current thrombosis field. However, we would like to respectfully argue that this viewpoint may not be necessarily correct. Around the year 2000, a series of oral antagonists of integrin $\alpha_{IIb}\beta_3$ were developed. which were expected to represent “the dawn of a new era in anti-thrombotic therapy, the era of $\alpha_{IIb}\beta_3$ antagonism”¹⁹. However, as documented by multiple phase III trials, these integrin $\alpha_{IIb}\beta_3$ antagonists (e.g., orbofiban) significantly increased patient mortality than placebo by enhancing the risk of myocardial infarction, despite their high potency in inhibiting platelet aggregation^{20,21}. This led to the discontinuation of these integrin $\alpha_{IIb}\beta_3$ antagonists for anti-thrombotic usage. It was till years later that researchers started to find that the ‘toxicity’ associated with these agents²¹ was due to their effect of stabilizing integrin $\alpha_{IIb}\beta_3$ in the active open conformation²², which corresponds to the E⁺Act.⁺ status defined in our manuscript. The activation of integrin $\alpha_{IIb}\beta_3$ would lead to platelet activation²³, which is featured by granule release and the expression of more, active integrin $\alpha_{IIb}\beta_3$ that may cause relapse of thrombosis once the dose of the $\alpha_{IIb}\beta_3$ antagonist declines in the plasma. In other words, although these $\alpha_{IIb}\beta_3$ antagonists temporarily silence the platelets in patients, they also convert these platelets into prothrombotic “time bombs”. The $\alpha_{IIb}\beta_3$ antagonists also causes thrombocytopenia as another side-effect due to the production of autoantibodies against the epitopes only exposed in E⁺Act.⁺ integrins^{22,24}. Learning from the failure of these $\alpha_{IIb}\beta_3$ antagonists, a chemical principle was recently developed to develop better anti-thrombotic candidates that lock integrin $\alpha_{IIb}\beta_3$ in the inactive state²⁵. Altogether, the above works and findings strongly indicate that the size is not the only important parameter of a platelet aggregate or a thrombus. Instead, the

multi-dimensional characteristics of thrombi, summarized as a 'profile' or a 'barcode' in our work, should be viewed carefully and comprehensively.

Following the above idea, the barcode system is useful in three aspects:

1) Anti-thrombotic drug evaluation The above integrin $\alpha_{IIb}\beta_3$ antagonists are hardly accessible, considering that they were discontinued more than 20 years ago. However, we postulate that they correspond to an effect barcode of [- - - 0 0 + +], based on the barcode of 7E3 and 10E5 ([- - - 0 0 0]) and the fact that they can induce integrin $\alpha_{IIb}\beta_3$ activation. Thus, if we find drug candidates in future that also show a '+' sign in the last two dimensions of their effect barcodes, we would be alarmed that they will likely cause similar life-threatening effects. Although no previous research has reported any anti-thrombotic agents that would increase Fg, VWF, P-selectin or PS level in the thrombus, it is reasonable to suspect that such effects would also bring extra risks to patients' health.

2) Patient prothrombotic status characterization As demonstrated in Figure 4c, people carrying risk factors of arterial thrombosis can have abnormalities in their thrombus barcode. Detecting these abnormalities would provide extra valuable information regarding the patient's prothrombotic status than only checking the size of the thrombus. Taking the study of hypertension in our work as an example, it was the identification of hypertension patients' thrombus barcode, [+ + 0 0 0 + +], with a high $E^+ \alpha_{IIb}\beta_3$ expression, that inspired us to eventually uncover the hyperactivity of GPIIb α -integrin $\alpha_{IIb}\beta_3$ mechanosensing axis as a new mechanism underlying the high risk of CVD in hypertension patients. Only measuring the thrombus size would not provide us with any specific mechanistic cues. Similarly, if another thrombosis-associated disease or unhealthy habit (e.g., smoking) triggers an abnormally high level of Fg, VWF, P-selectin or PS level in the thrombus, then we would be advised that some mechanism(s) around this biomarker have malfunctions, contributing to the higher thrombotic risks. In fact, understanding how different pathological conditions contribute to arterial thrombosis is what we aim to do next. In addition to the hypertension and aging tested in the current work, we have started to test patients with type II diabetes (T2DM), obesity or hyperlipidemia (HLM) and patients with two or more of these conditions (Fig. R4). However, the data we have acquired so far is still insufficient to draw any conclusion. Considering the demographic and health variations between patients, the complicated interactions between different risk factors, and the fact that many other risk factors have not even been included yet, we envision that using the thrombus profiling assay to evaluate risks of arterial thrombosis will eventually become a big program. This current work will just be the first of many to come. On the other hand, we are collaborating with UTMB hospital to investigate whether the thrombus barcode can be used to characterize the prothrombotic status of stroke patients and predict their next

stroke onset, wherein the Fg, VWF, P-selectin and PS levels and integrin $\alpha_{IIb}\beta_3$ activation levels will provide useful information allowing more accurate prediction.

(Redacted)

3) Personalized drug selection Acquiring the thrombus barcode of a patient may also help clinicians decide what drug or combination of drugs can reach higher efficacy and less side-effect in this patient. For example, as demonstrated in the Figure 6a,b and Supplementary Figure 10 of the original manuscript (current Supplementary Figure 12), NMC4 and 7E3 both inhibited the intensified biomechanical thrombogenesis of hypertension patients effectively. However, only NMC4 but not 7E3 can correct the over-activation of integrin $\alpha_{IIb}\beta_3$, indicating that abciximab (a derivative of 7E3 approved for clinical use) may not be the optimal drug for reducing the thrombotic risks in hypertension patients. We suspect that the best-matching drug/regimen likely varies from patient to patient, and we need to develop a library of anti-thrombotic agents to accommodate the need of different patients. As stated in the Discussion of our original manuscript, “developing anti-thrombotics with diversified effect barcodes and determining their interactions with thrombosis-exacerbating factors can help avoid treatment mismatch”.

Besides the above advantages, the barcode system is useful in one more aspect which is related to drug screening and mechanism discovery. As we stated in the original manuscript, an interesting attribute regarding the barcode system is that, “the effect barcode of each anti-thrombotic agent is dictated by its target rather than its pharmacological design”. This point is supported by many lines of evidence in the paper, which are summarized in Figure 3f. We further explained in the manuscript that, “this principle enables our assay to act as a one-step platform to screen anti-thrombotic agents against biomechanical platelet aggregation and use their effect barcodes to reason backward their targets. By the same token, effect barcodes can be used as identifiers of different contributing factors of biomechanical platelet aggregation, which allows the assay to pinpoint the functional context of new mediators and mechanisms, tackling the limitation of existing similar assays in providing mechanistic implications of arterial thrombosis”. As a testimony of this point, we would like to share some unpublished data from three ongoing projects in our lab. For the consideration of confidentiality and IP protection, we present them in their alias: Drug 1, Drug 2, and Drug 3. Drug 1 is a small peptide known to inhibit VWFA1 binding to GPIIb α . Indeed, as we characterized, Drug 1 manifests an effect barcode of [- 0 - - 0 - -] (Fig. R5a), identical to AK2 (GPIIb α inhibitor) and NMC4 (VWFA1 inhibitor) studied in our current work. Drug 2 is a small-molecule compound. In collaboration with MIT, we used our thrombus profiling assay for drug screening and identified ~15 hits from a compound library, with

Drug 2 being one of the most potent. Drug 2 manifests an effect barcode of [- 0 - - 0 - -] (Fig. R5b), suggesting its target to be GPIIb α -VWFA1 interaction. Indeed, our nano differential scanning fluorimetry and ELISA results confirmed that Drug 2 can specifically bind to VWFA1 and inhibit GPIIb α -VWFA1 interaction (data not shown). Drug 3 is a molecule naturally existing in the blood. Our preliminary data on mouse experiments showed that it can effectively inhibit arterial thrombosis (data not shown). We are planning to investigate its mechanism of action in the next step. Considering that the effect barcode of Drug 3 is identical to AK2, NMC4, Drug 1 and Drug 2 (Fig. R5c), we are confident that its target should also be GPIIb α -VWFA1 interaction. In summary, we hope you could appreciate that, using the thrombus profiling assay and the barcode system can help us quickly identify the mechanism of action of unknown agents that manifest anti-thrombotic effects. In contrast, by only assessing the size of the platelet aggregate, we would merely know that this agent is anti-thrombotic but have no clue of why, and will have to look for its target from scratch.

(Redacted)

Action taken: The story about the failure of oral integrin $\alpha_{IIb}\beta_3$ antagonists was already mentioned in the last paragraph of Discussion in the original manuscript, together with our advocacy of using the barcode system to avoid treatment mismatch. To make our message clearer, we also made the following changes to the text:

1) The following sentence in Discussion:

“In this context, our work further showcases the ability of the assay to identify the sources/mechanisms of prothrombotic tendency and suggest counteractive prevention/treatment strategies.”

was revised to:

“In this context, the detection of integrin $\alpha_{IIb}\beta_3$ over-activation and the identification of “treatment mismatch” using the barcode system further showcase the assay’s ability in identifying the mechanisms of prothrombotic tendency and in evaluating counteractive prevention/treatment strategies.”

2) To avoid overstatement as you suggested, the following sentence in Discussion:

“This principle enables our assay to act as a one-step platform to screen anti-thrombotic agents against biomechanical platelet aggregation and use their effect barcodes to reason backward their targets. By the same token, effect barcodes can be used as

identifiers of different contributing factors of biomechanical platelet aggregation, which allows the assay to pinpoint the functional context of new mediators and mechanisms, tackling the limitation of existing similar assays in providing mechanistic implications of arterial thrombosis.”

was revised to:

“These principles may enable us to quickly narrow down the possible target(s) of uncharacterized anti-thrombotic agents using their effect barcode.”

3) The following sentence in Discussion:

“Developing anti-thrombotics with diversified effect barcodes and determining their interactions with thrombosis-exacerbating factors can help avoid treatment mismatch, in which the ‘addition rule’ could be greatly helpful for prediction.”

was revised to:

“Developing diversified anti-thrombotics and determining their effect barcodes and their interactions with thrombosis-exacerbating factors can help avoid treatment mismatch, in which the ‘addition rule’ could be greatly helpful for prediction.”

4. Bleeding risks: Inhibiting platelet aggregation has not been a problem, as the biochemical axes that control aggregation have been known for decades, and many potential drug candidates have been identified as being able to do so. How to inhibit platelet aggregation while preserving low bleeding risks is, in fact, the real challenge, while this paper and the device developed do not seem to address this issue.

1) We respectfully disagree that inhibiting platelet aggregation has been well resolved. As mentioned in our original manuscript, a major challenge faced by the current anti-thrombotic treatments is antiplatelet resistance, meaning that the anti-thrombotic efficacy of the drug is compromised in certain patients²⁶. Pathological conditions such as hypertension, diabetes, metabolic syndrome and aging not only exacerbate thrombotic risks but also foster antiplatelet resistance to conventional antiplatelets. Importantly, the antiplatelet drugs that are being widely used and constantly encountering resistance, such as aspirin and clopidogrel, all target “biochemical axes that control platelet aggregation”, such as thromboxane A₂ (TXA₂)–TXA₂ receptor signaling axis, ADP–P₂Y signaling axis, and thrombin–PAR signaling axis (where platelet activation are all triggered by soluble agonists)^{10,27-29}. This indicates that effectively inhibiting platelet aggregation remains an unachieved goal, and solely targeting biochemical axes may not be able to resolve this problem. In such a context, our work for the first time identified a hyperactive biomechanical axis that intensifies platelet aggregation but cannot be inhibited by conventional antiplatelets, which is both

novel and clinically important, because it uncovers a new mechanism of antiplatelet resistance. We believe that effective inhibition of arterial thrombosis probably requires the inhibition of both biochemical and biomechanical mechanisms of platelet aggregation. This point has been elaborated in the Discussion of our original manuscript as follows:

“On the other hand, antiplatelet resistance is conventionally believed to be due to patients’ lack of sensitivity to antiplatelets in inhibiting platelet amplification loops³⁰. However, we found that biomechanical thrombogenesis is essentially “immune” to aspirin and clopidogrel in both healthy young subjects and hypertension patients (Supp. Fig. 12). These, together with similar observations by other works³¹⁻³³, indicate a new mechanism of antiplatelet resistance: biomechanical platelet aggregation can mediate arterial thrombosis independent of platelet amplification mechanisms, and therefore the sole inhibition of platelet amplification loops allows thrombotic risks to persist by leaving biomechanical platelet aggregation active.”

2) Although not being the focus of this study, our platform does have the potential of evaluating bleeding risks. In a previous work, a microfluidic assay using a similar stenosis design was used to evaluate bleeding risks in patients with von Willebrand disease, and a negative correlation was identified between the aggregate size and bleeding score of these patients². This work suggests the feasibility of using our thrombus profiling assay to test bleeding risks of subjects and to test the bleeding side-effect of anti-thrombotic drugs. In fact, one ongoing project in our lab, which aims to target the GPIIb/IIIa–VWF A1 interaction to safely inhibit arterial thrombosis, is taking advantage of this point.

(Redacted)

(Redacted)

However, we should acknowledge that, due to apparent differences in the biorheological conditions, the stenosis assay is unlikely to perfectly recapitulate the scenario of hemostasis, for which better designs already exist³⁴.

Action taken: In the first paragraph of Discussion, we made the following changes:

1) The thrombus profiling assay’s ability “in evaluating the function of anti-thrombotic drugs and candidates” was added.

2) The following statement was added:

“Notably, the assay also has the potential of evaluating bleeding tendency in patients and the bleeding side effect in antithrombotic agents², which warrants future investigation.”

5. Overstatements: The paper contains many overstatements that are not aligned with the status quo of the field. To name a few:

The authors claim that “As a symbolic feature of arterial thrombosis, severe stenosis in the blood vessel creates a high-shear, high-gradient flow However, no approach is currently available to comprehensively characterize thrombogenesis under this biorheological condition.” It has been well documented by many groups such as Brass, Diamond, Ku, Neelamegham, Neeves, and Lam, etc., with many microfluidic designs, including ones with multicolor staining that stenosis creates a high-shear condition and subsequently leads to platelet aggregation that can eventually block the lumen. A comprehensive understanding of the molecular participants in causing platelet aggregation has been largely understood.

Indeed, the stenosis assay has been used in multiple previous works, including one of our own³². However, firstly, some of these works only involved a single readout (platelets) and merely measured the thrombus size^{2,35-37}. Secondly, Kim et al. used histology and scanning electron microscopy to check the structure of shear-induced platelet thrombi, wherein no molecular biomarker was tracked³⁸. Lastly, the rest of the works did observe other biomarkers besides platelets^{32,39}. Yet, 1) as far as we know, none of these works used more than two colors (one for platelets and one for another biomarker, e.g., PS or VWF), which is unable to achieve multi-parametric thrombus characterization; 2) the second biomarker was only qualitatively assessed regarding its presence or absence, never demonstrating quantitative differences under different treatment/unhealthy conditions. By expanding our literature search to include other flow chamber assays that also emphasize shear force, e.g., the micro-post array⁴⁰, the “pressure relief” model⁴¹ and straight channel assays⁴²⁻⁴⁵, we still did not find any work that overcame the above limitations. We only found two works that used microfluidics to recapitulate hemostasis (wound closure), in which quantitative analysis was performed to inspect fibrin(ogen) deposition³⁴ or P-selectin expression⁴⁶ in addition to the size of the hemostatic clot, respectively. However, the microfluidic chips in these two works were specifically designed to mimic bleeding and hemostasis, which are essentially different from the scenario of arterial thrombosis; also, the number of readouts was still limited to two. In contrast to all previous studies, the thrombus profiling assay developed in the current work for the first time quantitatively characterizes the levels of multiple biomarkers in a thrombus, which sets the basis for its capability to comprehensively characterize biomechanical thrombogenesis and identify the effects of different anti-

thrombotic agents and pathological conditions in changing the size, composition, and activation status of the thrombus.

We would like to clarify that, our statement of "...comprehensively characterize thrombogenesis..." in the manuscript refers to the fact that currently no assay is available to concurrently track multiple key molecules indicative of thrombus size, composition and activation status within the thrombus and their expression levels, but was not meant to infer that "the molecular participants in causing platelet aggregation" are unknown. As we have stated in the manuscript, the "molecular participants in causing platelet aggregation" in biomechanical thrombogenesis are already known to mainly include GPIIb/IIIa and integrin $\alpha_{IIb}\beta_3$ and their ligands VWF and Fg. Notably, as an extra point that also reflects the novelty of our work, Figure 2 of our manuscript for the first time unraveled how these molecular participants respectively contribute to VWF and Fg deposition and platelet activation in the thrombus. For instance, shear-induced platelet activation is more sensitive to GPIIb/IIIa-VWF interaction but not integrin $\alpha_{IIb}\beta_3$ -VWF or integrin $\alpha_{IIb}\beta_3$ -Fg interaction; integrin $\alpha_{IIb}\beta_3$ -VWF and integrin $\alpha_{IIb}\beta_3$ -Fg interaction facilitate VWF and Fg deposition, respectively, but have no impact to the deposition of the other ligand.

Action taken: to clarify our points and eliminate possible misunderstanding, we made the following revisions:

1) In Abstract, "However, no approach is currently available to comprehensively characterize thrombogenesis under this biorheological condition."

was revised to:

"However, no approach is currently available to comprehensively characterize the size, composition and platelet activation status of thrombi forming under this biorheological condition."

2) In Introduction, "existing methods for observing biomechanical platelet aggregation cannot systematically characterize the thrombus or delineate different contributing factors"

was revised to:

"existing methods for observing biomechanical platelet aggregation cannot provide all-around information and quantitative analysis regarding the composition of the thrombus and the activation status of platelets within"

3) In Discussion, “Using a panel of mAbs with highly specific targets, we showed that platelet activation during biomechanical thrombogenesis—manifested by integrin $\alpha_{IIb}\beta_3$ activation and P-selectin expression—is primarily driven by GPIIb α mechanosignaling.”

was revised to:

“Using a panel of mAbs with highly specific targets, we showed that platelet activation during biomechanical thrombogenesis—manifested by integrin $\alpha_{IIb}\beta_3$ activation and P-selectin expression—is more dependent on GPIIb α mechanosignaling and less so on integrin $\alpha_{IIb}\beta_3$ mechanosignaling.”

Also, the authors claim, “However, whether abnormal biomechanical platelet aggregation is intercorrelated with higher risks of arterial thrombosis in humans was never studied.” which is completely incorrect, as this has been known for decades. Moreover, hypertension and aging have been known for decades to involve higher risks for arterial thrombosis with exacerbated platelet aggregation.

We would like to clarify that here our emphasis is “biomechanical platelet aggregation” instead of “platelet aggregation in general”. While abnormal platelet aggregation contributing to higher risks of arterial thrombosis is well acknowledged, whether it involves abnormal biomechanical platelet aggregation was never investigated. This is because when studying thrombosis in the context of hypertension, aging and other risk factors, most previous works used soluble agonists instead of shear force to trigger platelet aggregation⁴⁷. For instance, Mehta *et al.* used ADP and epinephrine⁴⁸, Zhao *et al.* used thrombin, collagen, ADP and U46619⁷, and Touyz *et al.* used thrombin⁴⁹. A small fraction of works inspected the correlation of thrombotic risks with spontaneous platelet aggregation (SPA), which refers to the formation of tiny platelet aggregates when blood or platelet rich plasma is constantly stirred over minutes^{50,51}. However, SPA still depends on the release of soluble agonists (e.g., ADP, ATP, thromboxane) from red blood cells and platelets over time to stimulate platelet activation⁵²⁻⁵⁴, while the contribution of mechanical force to platelet aggregation is likely negligible considering the very low stirring rate (20 rpm).

On the other hand, although high shear flow is known to trigger platelet aggregation (termed by us as biomechanical platelet aggregation) for decades, all previous works only used healthy human’s blood for testing and never inspected whether certain pathological conditions can intensify biomechanical platelet aggregation. Therefore, from these studies we can only conclude that biomechanical platelet aggregation contributes to arterial thrombosis, but cannot reach the conclusion that intensified biomechanical platelet aggregation contributes to higher risks of arterial thrombosis in certain populations.

To our knowledge, our work is the first to test whether human subjects with high thrombotic risk factors (e.g., hypertension, diabetes, cancer, aging, etc.) are associated with intensified biomechanical platelet aggregation.

Action taken: to clarify our points and eliminate possible misunderstanding, we revised the above sentence to:

“Unfortunately, platelet mechanobiology was barely investigated in pathological contexts^{7,8}. It remains unclear whether biomechanical platelet aggregation is intensified by any thrombotic risk factor, thereby contributing to higher incidence of CVD in certain human populations.”

Another one: “For example, platelets in biomechanical platelet aggregation are crosslinked via three molecular interactions: glycoprotein (GP) Iba–von Willebrand factor (VWF), integrin $\alpha_{IIb}\beta_3$ –VWF and integrin $\alpha_{IIb}\beta_3$ –fibrinogen (Fg)^{10,12,14}. Their respective contribution to biomechanical thrombogenesis and how they are affected by pathological conditions remains unclear. Jackson (2007) documented this almost 20 years ago, not to mention the recent advances in this area.

What we intended to say was that how these molecular interactions respectively mediate the composition (i.e., VWF and Fg levels) and the platelet activation status of the thrombus during biomechanical thrombogenesis, and how they are affected in the pathological contexts of hypertension, aging, etc., were never studied. We thank you for pointing this out and apologize for the confusion.

Action taken: to eliminate misunderstanding, we revised the above sentence to:

“For example, it remain elusive how the three molecular interactions mediating platelet crosslinking in biomechanical platelet aggregation, i.e., glycoprotein (GP) Iba–von Willebrand factor (VWF), integrin $\alpha_{IIb}\beta_3$ –VWF and integrin $\alpha_{IIb}\beta_3$ –fibrinogen (Fg)^{31,32,55}, respectively mediate the VWF and Fg levels and platelet activation in the biomechanical thrombus, and whether they are dysregulated by thrombotic risk factors to cause abnormalities in biomechanical thrombogenesis.”

We have also thoroughly checked the whole manuscript and revised all other spotted overstatements. If you find any more statement that you find inappropriate, we will be pleased to make further changes.

6. Broader impact: the study focuses heavily on the biomolecular aspects of platelet aggregation and thrombosis. Although it is good to know how each molecular axis can

be separately inhibited and alter platelet behavior, there is a lack of novel and exciting findings that can be translated to other areas, lacking excitement to the general audience.

We agree that this work is centered on thrombosis, which was why we stated during initial submission that the manuscript should be of interest to *Nature Communications'* collection of "*Advances in Heart Failure*". However, our mechanistic discoveries (Figure 5) do provide information useful to more generalized fields. Here we combined a total of 4 techniques to unravel a hyperactive state of integrin $\alpha_{IIb}\beta_3$ on hypertension patients' platelets, which directly contributes to intensified biomechanical thrombogenesis. This provides a new linkage of the dysfunction, particularly mechanobiological dysfunction, of integrins with disease pathology, which should be of interest to the broad fields of integrin biology and mechanobiology in addition to vascular biology. Of note, how integrin conformations are associated with integrin function and various pathological conditions such as cancer, fibrosis, inflammation and thrombosis, and how to regulate integrin conformations for therapeutical purposes, have been hot topics in the integrin field for over 20 years^{25,56-60}. For instance, Fan *et al.* unraveled that bent β_2 integrins can adopt high affinity, which limits neutrophil arrest (*Nature Communications* 2016; *Cell Reports* 2019)^{61,62}. Lin *et al.* discovered a chemical principle to design inhibitors that locks integrin $\alpha_{IIb}\beta_3$ in the bent conformation for safer treatment of thrombosis (*Cell* 2022)²⁵. Li *et al.* discovered that ligand binding greatly accelerates integrin extension conformational changes and provided a new perspective regarding the sequence of conformational states an integrin adopts during activation (*Cell* 2024)⁶³.

The extended-close conformation of integrins was conventionally believed to be inactive and thus was never a center of investigation in (patho)physiology. In 2019, we first discovered that extended-close integrin $\alpha_{IIb}\beta_3$ adopts an elevated affinity and elevated binding strength for ligands, which is central to biomechanical platelet aggregation³². In the current work, we further demonstrated that the extended, mostly extended-close, $\alpha_{IIb}\beta_3$ integrins are a key factor in increasing the thrombotic risks in hypertension patients. Does any of the other 23 integrin species also manifest similar high-activity behavior in the extended-close conformation? If so, do they also contribute to the pathology of diseases? These will be very important questions to address in future investigations.

Action taken: The importance of extended-close integrin $\alpha_{IIb}\beta_3$ in arterial thrombosis has already been discussed in detail in the original manuscript. To emphasize the interest of our work to general readership, in the revised manuscript, we added the following statements in Discussion:

“The results also underscore the pathophysiological relevance of E⁺-closed integrin $\alpha_{IIb}\beta_3$, which should inspire future investigations on the importance of the E⁺-closed conformation in other integrins and in the context of other diseases.”

Additional comments:

- This device is similar to many microfluidic devices that feature (10) individual channels, where each channel needs to be perfused separately. How would such a typical design make this device a high-throughput one? Can you explain what high throughput is vs. the opposite?

For the rigor of science, all experiments in this work were done with every channel perfused separately, so that we could closely monitor the thrombus formation over time. However, for future usage in drug screening and patient evaluation, where temporal resolution can be lower, our device has the potential to reach much higher throughput without needing any change in the design. Specifically, the 10 channels on the device can be run simultaneously by using a syringe pump mounted with a multi-channel syringe rack, and a microscope with a programmable motorized stage can be used to automatically alternate the image acquisition position back-and-forth between the 10 different channels. In this way, thrombus formation in the 10 channels can be recorded in parallel in real time. The feasibility of this setup has been validated in our lab (Fig. R7). The total time of one parallel run, including the time required for preparation, blood perfusion, and clean-up, is ~30 min. Therefore, we stated in our discussion that the assay can reach a relatively high throughput (10 runs per 30 min), which is equal to a maximum capacity of testing 50 drugs or 50 subjects per day. This throughput is not “high” *per se*, but already considerably good. We apologize if our language caused confusion.

Figure R7. Setup demonstrating a parallel run of the thrombus profiling assay. A multi-channel syringe rack is mounted with 10 syringes, which are connected to the 10 channels in one microfluidic chip, respectively. A programmable motorized stage on the microscope is pre-set to automatically move back and forth between the 10 channels over time, which allows alternate capturing of images of the growing thrombi within all 10 channels.

Furthermore, with the use of a multi-camera array microscope (e.g., Ref⁶⁴), thrombus formation in multiple channels can be monitored concurrently in real time. Notably, the multi-camera array microscope in Ref⁶⁴ is equipped with 96 cameras, with which the throughput of our assay can be even higher, reaching a theoretical value of 96 runs per 30 min, although the practical throughput will be slightly lowered by the time consumed in tubing connection/disconnection. To make this setup work also requires re-designing of the microfluidic chip to accommodate 96 channels. Considering that this high throughput is beyond the scope of our current work and lacks experimental validation, we decide not to discuss it in the manuscript.

Action taken:

To avoid confusion, we revised the previous statement to following:

“Hardware upgrades, e.g., using a multi-channel syringe pump and a motorized stage or a multi-camera array to reach relatively high throughput, and/or system automation, will enable the current setup to become more suitable for clinical practice.”

- Are all the reagents commercially available? How expensive and how much time does it cost to obtain one useful barcode? Would this be clinically acceptable compared to other platelet functional assays available in the market, such as PFA-100, T-TAS, etc.?

All reagents and materials are commercially available and have been listed in the Methods section. Several molecular sensors need to be fluorescently labeled, but the labelling kits and reagents are also commercially available. To acquire one useful barcode, the cost is estimated to be ~\$12. As a reference, the QStat Quantra Cartridges from Hemosonics, which are used on the Quantra Hemostasis System⁶⁵, provide multi-parametric results regarding coagulation and soluble agonist-induced platelet aggregation, are sold at a price of ~\$125 each (based on our recent purchases for a collaborator), and each cartridge can be used for one blood sample. The AR Chips for T-TAS01, which studies the interactions between primary hemostasis and the coagulation cascade under flow conditions, cost ~\$25 each (according to a quote we acquired from DiaPharma). PFA-100, which targets soluble agonist-induced platelet aggregation and provides ‘closure time’ as the only readout⁶⁶, has each test costing ~\$12⁶⁷. Therefore, our assay is competitive with regard to cost-effectiveness.

As stated in our answer to your previous question, when the thrombus profiling assay is eventually upgraded for clinical uses, it should have acquired the capacity of performing parallel runs, which will allow the test of one blood sample in 6 minutes in average. This is much faster than the T-TAS system (30-60 minutes) and the Quantra Hemostasis System (15 minutes), and is comparable to PFA-100 (2-10 minutes). Therefore, our assay is also competitive in the test speed.

Action taken:

We tuned down the statement by revising “high cost-effectiveness” to “**cost-effectiveness**” in the Discussion.

- Physiologically, the thrombotic surface is rich in collagen. Why was the VWF-A1 monomer used for the channel coating instead? How is this justifiable pathophysiologically? Why not use VWF multimer or collagen? How would the results change if VWF multimers or collagen were used?

We would like to clarify that VWFA1 monomer coating was never used in the stenosis assay, but was only used in the conventional flow chamber assay. As stated in the “*Microfluidic stenosis assay*” section of Methods, for all stenosis assays, “Microfluidic channels were coated with VWF monomer ($2 \mu\text{g mL}^{-1}$) for 1 h”. This coating condition is pathophysiologically relevant because although the thrombotic surface is rich in collagen, the collagen still needs to bind to and be covered by plasma VWF to attract and retain platelets for thrombogenesis under high shear flow. We chose to use VWF monomer instead of VWF multimer because the former retains all the key functions of native VWF when coated on a surface, allowing platelet attachment via both GPIIb/IIIa–VWF and integrin $\alpha_{IIb}\beta_3$ –VWF interactions. Yet, VWF monomer (~\$225 per 100 μg (Sinobiological)) is much more cost-effective than VWF multimer (\$626 (Sigma-Aldrich) or \$638 (Thermofisher) per 100 μg). As you pointed out above, since the assay is developed to aim for clinical use, its cost should be carefully controlled.

To validate our points, we performed additional experiments to check how changing the coating condition affects thrombus formation. As you suggested, we replaced VWF monomer with collagen for channel coating, which was found to have no impact on thrombus formation (Fig. R8). However, with collagen coating, thrombus formation was basically eliminated by RU5 which blocks plasma VWF binding to collagen (Fig. R8). This result confirms our above argument that even if we coat the surface with collagen, it still needs to recruit VWF from the blood plasma for platelet attachment. It also suggests that a surface coated with VWF multimer (by coating the surface with collagen to attract plasma VWF coverage) or with VWF monomer has comparable efficacy in supporting thrombus formation, validating our use of VWF monomer as a cheaper substitute for surface coating.

Figure R8. Scatter plots with mean±s.e.m. of the normalized size of thrombi formed in blood anti-coagulated with heparin, citrate or EDTA, and in stenotic channels coated with VWF monomer or collagen. In some runs with collagen coating, blood was added with 10 $\mu\text{g/ml}$ RU5 to block plasma VWF

binding to collagen. Data in all groups are normalized by the average of the heparin/VWF group. *P*-values: results of one-way ANOVA (F-value =8.391, degrees of freedom =46) and multiple comparison.

Action taken:

The above results were added to Supplemental Figure 3 as the new panel (a). The following description was also added into Results:

“Replacing VWF with collagen for channel coating did not significantly affect thrombus formation. However, with collagen coating, thrombus formation was basically eliminated by RU5 which blocks plasma VWF binding to collagen, reflecting an indispensable role of VWF on the hump for platelet attachment (Supp. Fig. 3a).”

- The blood was anticoagulated by heparin, while many studies use citrate or EDTA. Can you justify why heparin was used? Can you rule out the anticoagulant dependences?

Heparin was used because it best preserves platelet function. The mechanism of action of heparin as an anticoagulant is by increasing the activity of antithrombin and thereby inhibiting thrombin generation and fibrin formation. It does not act on platelets or any plasma protein involved in the biomechanical platelet aggregation. In contrast, citrate and EDTA both have direct influence on platelet function. Firstly, platelet activation during biomechanical platelet aggregation requires the influx of calcium ions from the blood³². However, both citrate and EDTA chelate calcium ions from the blood, which therefore strongly inhibit platelet activation. Secondly, as shown in this current work and in previous literature, integrin $\alpha_{IIb}\beta_3$ is a central player in biomechanical platelet aggregation^{31,32}. However, EDTA is a well-known agent that irreversibly eliminates integrin activity⁶⁸, which therefore would greatly impact the results of our assay.

To validate our points, we performed additional experiments to check how changing the anticoagulant affects thrombus formation. Agreeing with our prediction, using citrate instead of heparin reduced the thrombus size by >50%, while the use of EDTA basically eliminated thrombus formation (Fig. R8). These results validated our use of heparin.

Action taken:

The above results were added to Supplemental Figure 3 as the new panel (a). The following description was also added into Results:

“Replacing heparin with citrate or ethylenediaminetetraacetic acid (EDTA) for anticoagulation attenuated thrombus formation (Supp. Fig. 3a), because the latter two

chelate calcium from the blood and inhibit platelet activation, while EDTA also eliminates integrin $\alpha_{IIb}\beta_3$ activity. These results validate the use of VWF and heparin for channel coating and blood anticoagulation, respectively.”

- It seems that the barcode was obtained based on the changing trend of the fluorescence signal intensity (FSI). It is, however, known that the FSI can be highly noisy and is more of a qualitative metric. Also, some of the FSI features very small changes in magnitude, although being statistically significant (e.g., fig 3E-Act, 3D-fg).

The quantitation of fluorescence signal intensity, which is also referred to as “fluorometry”, is being widely used in many techniques. For instance, fluorescence quantitation standard beads from Bangs Laboratories and from BD Biosciences are used for the quantitation of fluorescence signal intensity in flow cytometry. Using this beads, one can precisely calculate how many copies of fluorescently tagged antibodies are attached to each cell in their sample. This protocol has been used by our lab and many other labs for over 15 years. As another example, fluorescence signal acquisition is one of the common signal detection methods in ELISA, which according to the webpage of “*Overview of ELISA*” from ThermoFisher, is both sensitive and highly reproducible. It is well acknowledged that ELISA detects and quantifies the amount of soluble substances in samples, which is not just a qualitative assay but a highly quantitative assay. Here are also some other references that used fluorometry for the quantitative analysis of: DNA⁶⁹, dopamine⁷⁰ and chlorophyll levels⁷¹.

In this work, the robustness of our quantitative analysis on the fluorescence signal intensity can be reflected in the following aspects:

1) As shown in the original Supp. Fig. 3c (new Supp. Fig. 3d), using Sensor Set 1 and 2 to respectively test the same subjects’ blood samples rendered comparable total platelet signal intensity. This shows high repeatability of our assay in quantifying the fluorescence signal intensity of platelets.

2) As stated in our original manuscript, “we repeated our test on 14 randomly picked subjects after different time intervals (from 2 weeks to 9 months). Among a total of 21 re-tests, only 2 showed changes in the personal thrombus barcodes, which were associated with the longest time intervals (7 and 9 months, respectively) (Supp. Fig. 10).” This shows high repeatability of our assay in quantifying the fluorescence signal intensity of all biomarkers in the thrombus profile.

Therefore, the scattering of data points in Fig. 4c,f and original Supp. Fig. 5 (new Supp. Fig. 6) should be mainly due to inter-individual variability instead of measurement noise. By acknowledging the scattering pattern of the data, we did use multiple analyses in parallel to check whether the different biomarkers can distinguish healthy young group

from hypertensive and/or older groups. These include: 1) using two-way ANOVA to check whether the levels of these biomarkers were significantly different among different cohorts (Fig. 4c); 2) using linear regression to check whether a positive correlation exists between the thrombus size and Fg, VWF, P-selectin, PS, E+ $\alpha_{IIb}\beta_3$, Act. $\alpha_{IIb}\beta_3$ levels (Fig. 4f; new Supp. Fig. 6; Supp. Table 2); 3) checking the sensitivity-specificity performance of different biomarkers where a threshold value is used for separation (Supp. Table 2). To address your concern, we added another two statistical analyses (Spearman rank correlation coefficient⁷² and Kendall's tau correlation coefficient⁷³), which check whether a positive correlation exists between the thrombus size and Fg, VWF, P-selectin, PS, E+ $\alpha_{IIb}\beta_3$, Act. $\alpha_{IIb}\beta_3$ levels without assuming any fitting model. The analysis results of these two methods are consistent with the result of the linear regression, identifying Fg, E+ $\alpha_{IIb}\beta_3$, and Act. $\alpha_{IIb}\beta_3$ as positive correlating factors of thrombus size (Table R1).

Readout	Fg/Plt	VWF/Plt	P-sel/Plt	PS/Plt	E+ $\alpha_{IIb}\beta_3$ /Plt	Act. $\alpha_{IIb}\beta_3$ /Plt
Slope non-zero p-value	0.040	0.880	0.954	0.154	<0.0001	<0.0001
Spearman rank correlation p-value	0.007	0.20	0.77	0.18	<0.0001	<0.0001
Kendall's tau correlation p-value	0.007	0.18	0.82	0.20	<0.0001	<0.0001

Table R1. Comparing the results of different statistical analysis methods on assessing the correlation between different thrombus profiling readouts.

Action taken:

The results from the two new statistical analyses were added to Supplemental Table 2. The corresponding descriptions were also added in Results and the *Statistical Analysis* section of Methods.

How did the authors decide on the barcode specifically? Manually? Automatically based on some threshold?

The barcodes were decided as follows:

1) To acquire the effect barcode of an anti-thrombotic agent: the agent was tested on blood samples from multiple healthy young subjects (n =5 in this study), and then the

readout from each biomarker was compared between the runs with and without the agent. A statistically significant difference in the readout would be recorded as '+' or '-'; otherwise, '0'.

2) To acquire the effect barcode of a pathological condition: blood samples from multiple healthy young subjects (n =33 in this study) and from subjects carrying the pathological condition were tested, and then the readout from each biomarker was compared between different cohorts. A statistically significant difference in the readout would be recorded as '+' or '-'; otherwise, '0'.

3) To acquire personal thrombus barcodes, as stated in the original manuscript: "From the thrombus profiles of healthy young subjects, values of each dimension were fitted to a Gaussian distribution, of which the mean \pm 2S.D. (~95% confidence interval) was defined as the reference range ('0') (original Supp. Fig. 9 or new Supp. Fig. 10), and values lower or higher were defined as abnormally low ('-') and high ('+'), respectively." In other words, the barcodes were decided based on standardized thresholds.

Does a small change make a difference in deciding therapy while statistically being a significant difference?

It is certainly possible that some individuals' personal thrombus profiles have values around the threshold or borderline, so that small changes in the value would result in the corresponding barcode changing between "normal" and "abnormal". Also, as we have stated in the manuscript, the reference ranges for deciding the personal thrombus barcode are decided based on the principle that they should include ~95% of healthy young subjects, meaning that the remaining 5% will manifest as "abnormal". Therefore, the barcode being normal or abnormal may not always provide definite black-and-white diagnostic decisions. However, in most cases, the barcode is still informative.

In fact, the inclusion of 95% of the healthy population as the reference range is being used in numerous clinical tests, e.g., blood tests (platelet count, RBC count, hematocrit, WBC count, etc.), where a value that is borderline high or low would also make the judgement relatively difficult, and an abnormal value in these tests also does not necessarily mean that the tested individual has pathological issues. Clinical practice has two approaches to address this problem and make the diagnosis more informative. Firstly, the use of a more detailed segmentation standard. For example, the judgement of high blood pressure generally uses the following standard: healthy blood pressure is less than 120/80; prehypertension is a systolic pressure of 120 to 139 or a diastolic pressure of 80 to 89; stage-1 high blood pressure ranges from a systolic pressure of 140 to 159 or a diastolic pressure of 90 to 99; and lastly, stage-2 high blood pressure is over 160/100. With more detailed segmentation, the most suitable treatment regimen

can be decided more specifically and personally. Secondly, complementary tests, and sometimes the clinical judgement of the physician, ensure more robust diagnosis. For instance, with the identification of white blood count being too high or too low, additional tests such as a complete blood count, blood differential, blood smear, and/or bone marrow test will be conducted to further check the health condition of the individual and whether a disease exists. Although no other bioassay is currently available for clinically evaluating risks of thrombosis, as we mentioned in the manuscript, risk score assessment is being widely used in preventive cardiology^{74,75}. Therefore, the application of our thrombus profiling assay in clinical practice will certainly benefit from being combined with the risk score assessment, so that if borderline values are acquired in the thrombus profiling assay, the physician can review the patient's risk score and make a more comprehensive judgement.

Importantly, the focus of the current work is to showcase the potential functions and reliability of the thrombus profiling assay. How to optimize its application in clinical practice, for instance, by designing more detailed segmentation and its combination with the risk score assessment as mentioned above, is beyond the scope of this manuscript, which will be a promising direction for future investigations.

Action taken:

1) The following sentence was added at the beginning of the paragraph that introduces the different statistical analyses.

“We then evaluated the inter-correlation of the different biomarkers and their performance in distinguishing different cohorts. To address the scattering pattern of their signal intensities (Fig. 4c), which is likely due to inter-individual variability, multiple statistical analyses were performed for cross-check.”

2) We added the results of the newly performed statistical analyses to Supp. Table 2, and added the following statement in Results:

“Firstly, by using linear regression model, Spearman rank correlation coefficient⁷² and Kendall's tau correlation coefficient⁷³, a positive correlation was consistently identified between thrombus size and Fg, E⁺ $\alpha_{11b}\beta_3$ and Act. $\alpha_{11b}\beta_3$ levels (Fig. 4e, Supp. Fig. 6), with E⁺ $\alpha_{11b}\beta_3$ being the strongest correlating factor (Supp. Table 2).”

In Materials and Methods, the following statement was added:

“Spearman rank correlation coefficient⁷² and Kendall's tau correlation coefficient⁷³ were also used to test whether a positive correlation exists between different readouts of the thrombus profile.”

3) To increase the robustness of our results, for subjects who have been repeatedly tested on different days, we used the average values of all test results instead of their first time's test results in data presentation and statistical analysis in Fig. 4, Supp. 5b,c, Supp. Fig. 7, 8 and Supp. Table 2. A note was added in Methods.

4) We have added the following statement in Discussion:

“To provide more accurate diagnosis and treatment suggestions, the assay can benefit from more detailed segmentation (e.g., borderline, stage-I, and stage-II abnormal) in judging normal *versus* abnormal thrombus barcodes, and can be combined with other existing diagnostic approaches, e.g., risk score assessment^{74,75}.”

Reviewer #2 (Remarks to the Author):

The authors present interesting and significant findings on the interactions between anti-thrombotic inhibitors and hypertension, as well as the inter-individual variability in personal thrombus profiles. However, the research could benefit from a more defined focus and clearer conclusions. The manuscript can be strengthened by improving clarity and organization.

We sincerely thank you for appreciating our work's novelty and importance and the constructive suggestions.

1. Consider redefining the title to articulate the novelty and focus of the experimental data. If microfluidics are central to your methods, explicitly include "via microfluidics" in the title. The use of terms like “uncovers” and “mechanobiology” might be too broad and non-specific for this investigation. In addition, please find a better way to define your cohort of hypertension and age-related thrombosis.

Thank you for these helpful suggestions.

Action taken: We have changed our title to the following:

“Multi-parametric thrombus profiling microfluidics detects intensified biomechanical thrombogenesis associated with hypertension and aging”

2. Could you succinctly describe how the study quantifies the impact of biomechanical forces on thrombus formation? Additionally, please provide a brief overview of the specific interactions studied and their distinct influences on the thrombus.

The thrombus formation observed in this work is termed “biomechanical platelet aggregation”, because it uses shear force as the only external stimulation to induce thrombus formation because no external agonist is added to the blood. Briefly, when the platelets experience high-shear and high-gradient flow, they will be able to undergo spontaneous aggregation like material self-assembly. Therefore, we investigate not just how biomechanical forces impact, but in fact how biomechanical forces drive thrombus formation.

The mechanisms of how platelets use biomechanical forces to achieve aggregation are as follows. Firstly, shear-induced VWF activation⁷⁶ and a catch bond mechanism in GPIb α –VWF interaction⁷⁷ together enable the GPIb α –VWF binding to become stronger under force, which initiates platelet crosslinking and the aggregate build-up under high shear force. This process is independent of platelet activation. Secondly, binding of GPIb α to VWF under force initiates GPIb α mechanosignaling that leads to integrin α _{IIb} β ₃ activation to an intermediate affinity and an extended-close (E⁺Act.) conformation³², which allows more stable thrombus development by binding to Fg and VWF. These mechanisms have been carefully investigated in previous studies by us and our colleagues^{31,32}. Notably, the relative contribution of the above molecular interactions to platelet aggregation changes as the shear rate increases, shifting from GPIb α and integrin α _{IIb} β ₃ contributing equally to GPIb α eventually outweighing integrin α _{IIb} β ₃⁷⁸.

On the other hand, the so-named spontaneous platelet aggregation (SPA), which refers to the formation of small platelet aggregates when blood or platelet rich plasma is constantly stirred over minutes^{50,51}, should be a secondary and even trivial mechanism that contributes to the thrombus formation in our study. This is because SPA requires the localized accumulation of agonists released from attached platelets and red blood cells to activate platelets, which is prevented by the high-speed perfusion.

Action taken:

As suggested, we added descriptions and overviews at several places of the manuscript.

1) We added the following content in Introduction:

“The biomechanical platelet aggregation process is composed of two steps mainly involving three molecular interactions. Firstly, shear-induced VWF activation⁷⁶ and GPIb α –VWF catch bond⁷⁷ together enable the GPIb α –VWF binding to become stronger under force, which initiates the aggregation of platelets in an activation-independent manner. Then, the GPIb α –VWF binding under force initiates GPIb α mechanosignaling that leads to integrin α _{IIb} β ₃ activation to an intermediate affinity and an extended-close

(E⁺Act.⁻) conformation³², which subsequently binds to Fg and VWF to allow more stable thrombus development.”

2) We added the follow content in the first paragraph of Results:

“With the above settings, platelet thrombi can be consistently observed within the channel, which is primarily driven by shear force because no external agonist is added to the blood and the high-speed perfusion prevents the localized accumulation of agonists released from attached platelets and red blood cells.”

3) We added the follow content in Discussion:

“On the other hand, as shown in previous works, when the shear rate increases, the dependency of shear-induced platelet aggregation on GPIIb/IIIa activity becomes progressively stronger and on integrin α IIb β 3 activity progressively weaker⁷⁸. It will be interesting to test whether changing the shear rate in our assay affects how the three receptor–ligand interactions contribute to the thrombus profile.”

3. Please include citations to substantiate the claim that the biomechanical aspect of platelet aggregation (driven by mechanical forces) has not been thoroughly studied in its correlation with increased risks of arterial thrombosis. Additionally, please elaborate on how interactions involving GPIIb/IIIa–von Willebrand factor (VWF), integrin α IIb β 3–VWF, and integrin α IIb β 3–fibrinogen distinctly contribute to thrombogenesis.

1) Regarding your first point, to our knowledge, only two previous papers reported platelet hyperreactivity associated with mechanobiology. The first work reported abnormal activity of Piezo1, a mechanosensitive ion channel, associated with hypertension⁷. However, its exclusive use of mouse blood and mice in experiments compromises the human relevance of its conclusion. Moreover, as in many other papers, mechanistic investigations in this work only stimulated platelets with soluble agonists and observed platelet signalling and aggregation under a normal pressure environment. Therefore, although this work claims to associate platelet hyperreactivity with mechanobiology, it cannot exclude the possibility that the observed intensified contribution of Piezo1 is associated with biochemical but not biomechanical mechanisms. The second work, which was done by some of the authors of the current manuscript, discovered intensified shear-driven thrombus formation in diabetic mice associated with a higher sensitivity of platelets to compressive forces⁸. However, the paper stated that the compressive forces are likely introduced by frequent collisions of red blood cells in the blood flow to the adhered platelets, which is in the phase of platelet recruitment when platelet aggregation has not started. Also, the work did not identify the receptor of the compressive forces on platelet surface and thus did not provide the molecular mechanism underlying the mechanosensing.

2) Regarding your second point, based on our findings in Fig. 2, all the three molecular interactions: GPIb α -VWF, integrin $\alpha_{IIb}\beta_3$ -VWF and integrin $\alpha_{IIb}\beta_3$ -Fg, contribute to the thrombus growth. However, GPIb α -VWF interaction is more critical than the other two because: 1) blocking GPIb α -VWF interaction eliminates thrombus formation, which is not achievable by blocking integrin $\alpha_{IIb}\beta_3$ -VWF or integrin $\alpha_{IIb}\beta_3$ -Fg interaction; 2) GPIb α -VWF interaction is more important in inducing platelet activation, because its partial blockage inhibits P-selectin expression and integrin activation in the growing thrombus, which is not achievable by partially blocking integrin $\alpha_{IIb}\beta_3$ -VWF or integrin $\alpha_{IIb}\beta_3$ -Fg interaction. This agrees with our previous observation that without GPIb α mechanosignaling to prime integrin $\alpha_{IIb}\beta_3$ into the intermediate state, inactive integrin $\alpha_{IIb}\beta_3$ itself cannot be activated by binding to its own ligands³². On the other hand, our data indicate that GPIb α -VWF and integrin $\alpha_{IIb}\beta_3$ -VWF interactions modulate the deposition of VWF into the thrombus, while integrin $\alpha_{IIb}\beta_3$ -Fg modulates that of Fg. These observations are intuitive because the recruitment of VWF and Fg into the thrombus is achieved by their attachment to their respective receptors on the platelet surface.

Action taken:

1) We added the citation of the two works mentioned above in Introduction (References 15 and 16 in the revised manuscript):

“However, while the mechanobiology of platelets is already barely investigated in pathological contexts^{7,8}, whether biomechanical platelet aggregation can be intensified by any thrombotic risk factor, thereby directly contributing to higher incidence of CVD in certain human populations was never studied.”

2) We extensively expanded the discussion on the contribution of different molecular interactions in Discussion:

“Using a panel of mAbs with highly specific targets, we showed that GPIb α -VWF, integrin $\alpha_{IIb}\beta_3$ -VWF and integrin $\alpha_{IIb}\beta_3$ -Fg interactions all contribute to the size growth of biomechanical thrombi. However, suggesting a central role of GPIb α -VWF interaction, only its blockage but not the blockage of integrin $\alpha_{IIb}\beta_3$ -VWF or integrin $\alpha_{IIb}\beta_3$ -Fg interaction can eliminate thrombus formation. Also, GPIb α mechanosignaling appears to be more critical than integrin $\alpha_{IIb}\beta_3$ mechanosignaling to platelet activation during biomechanical thrombogenesis, as manifested by integrin $\alpha_{IIb}\beta_3$ activation and P-selectin expression. This agrees with our previous observation that GPIb α mechanosignaling serves as the initiator in the GPIb α -integrin $\alpha_{IIb}\beta_3$ mechanosensing axis, without which integrin $\alpha_{IIb}\beta_3$ cannot activate itself by binding to its own ligands³². We showed that GPIb α -VWF and integrin $\alpha_{IIb}\beta_3$ -VWF interactions both modulate the

deposition of VWF into the thrombus, while integrin $\alpha_{IIb}\beta_3$ -Fg only modulates that of Fg, which seem intuitive because both VWF and Fg need to be bound to their respective platelet receptors to maintain their presence in the thrombus. However, the fact that inhibiting either VWF or Fg binding to integrin $\alpha_{IIb}\beta_3$ does not enrich the other ligand, but both reduce the thrombus size, suggests that VWF and Fg cooperate, rather than mutually compensate, in integrin $\alpha_{IIb}\beta_3$ crosslinking for biomechanical platelet aggregation.”

4. In Figure 1, the results indicate initial rapid thrombogenesis followed by slower progression, with signal quantification at 450 seconds. Please explain the rationale behind selecting this specific time point for signal quantification. What do the observed changes in the rate of thrombogenesis imply about the underlying mechanisms of thrombus formation under biomechanical conditions?

As described in previous publications, the dynamics of biomechanical platelet aggregation, which can be fitted by a sigmoid function, is as follows: 1) a monolayer of platelets first covers the site of stenosis; 2) a linear phase of platelet aggregation; and lastly 3) a quasi-steady phase that undergoes cyclical aggregation–disaggregation^{2,31}. Due to the low activation status, platelets in the biomechanical thrombus can still be detached by shear force and leave the thrombus. Meanwhile, new platelets join the thrombus. Eventually, the thrombogenesis reaches a relative dynamic equilibrium between platelet detachment and attachment, realizing the slower progression. In our setting, the linear phase of platelet aggregation occurs within the first 300-400 s, and 450 s corresponds to a time that the thrombi has already reached the quasi-steady phase (Fig. 1g and 4b of the manuscript). Selecting different timepoints for signal quantification would certainly affect the readings. However, we believe that the timepoint of 450 s should be representative because it gives the thrombus enough time to develop into the quasi-steady state. At the same time, it avoids long observation time which would require a larger volume of blood, increase the test cost, and lower the experiment throughput.

Action taken:

1) We expanded our explanation for selecting 450 s as the timepoint for signal quantification:

“Agreeing with previous observations, real-time tracking showed rapid thrombogenesis in the first 300-400 s followed by a quasi-steady phase, in which the thrombus reaches a relative equilibrium between platelet aggregation and disaggregation^{2,31} (Fig. 1g). Thus, we selected 450 s after the onset as the time point for quantitating fluorescent

signals so as to assess the thrombus in the fully developed status while avoiding unnecessary waiting (Fig. 1h).”

2) We used the sigmoidal model to fit the data in Fig. 4b (shown below in Fig. R9) and identified a difference in the dynamics of biomechanical thrombogenesis between healthy young and hypertensive groups.

Figure R9. Comparing the time course of thrombus growth (mean±s.e.m. with fitting lines of sigmoidal model) between healthy young and hypertensive groups (n =8)

We updated the original Fig. 4b accordingly, and described the identified difference in the dynamics of thrombogenesis as follows:

“By fitting the “thrombus size versus time” curves with the sigmoidal model, it was observed that unlike the growth of healthy young subjects’ thrombi which approached a plateau at ~400 s, hypertension patients’ thrombi remained in the rapid development phase until ~500 s, also indicating a clear prothrombotic tendency (Fig. 4b).”

5. In Figure 2, the findings indicate that only AK2 and NMC4, which inhibit the GPIIb/IIIa–VWF interaction, completely halted thrombogenesis. Could you discuss the underlying mechanisms that render this interaction critical for initiating biomechanical platelet aggregation?

The related mechanisms have been discussed in our previous publication³². Briefly, on unstimulated healthy platelets, GPIIb/IIIa is the only receptor that can mediate shear-induced platelet aggregation, while integrin $\alpha_{IIb}\beta_3$ stays in its inactive state and is incapable of effectively binding to ligands. In other words, GPIIb/IIIa–VWF interaction has to serve as the initiator of biomechanical platelet aggregation. While binding to VWF

under force, GPIIb/IIIa triggers mechanosignaling that induces integrin $\alpha_{IIb}\beta_3$ activation to the intermediate (E⁺Act⁻) state. Only after this will integrin $\alpha_{IIb}\beta_3$ be able to effectively bind to VWF and Fg and further contribute to biomechanical platelet aggregation.

Action taken:

In response to your second comment, we have already added a description for the process of biomechanical platelet aggregation in Introduction. In addition, we also revised our discussion on the related results to the following:

“Only AK2 and NMC4, but not the other mAbs, eliminated thrombogenesis (Fig. 2c-e), which agrees with previous findings that GPIIb/IIIa–VWF interaction serves as the initiator of biomechanical platelet aggregation, without which integrin $\alpha_{IIb}\beta_3$ cannot be activated for platelet crosslinking^{31,32}.”

6. The inhibition of integrin $\alpha_{IIb}\beta_3$ interactions with VWF and Fg significantly, albeit variably, affected thrombus size. Particularly with the potent effects of 7E9, could you discuss how these interactions contribute to thrombus structure and growth? What does the variable impact on thrombus size and composition suggest about their roles in thrombogenesis?

As shown in Fig. 2 and Supp. Table 1, the potency of LJ-P5 is comparable to that of 7E9 and the cocktail of 7E9/LJ-155B39/LJ-134B29, which all can reduce the size of the thrombus by ~95%. This indicates that the importance of integrin $\alpha_{IIb}\beta_3$ –VWF and integrin $\alpha_{IIb}\beta_3$ –Fg interactions to biomechanical platelet aggregation should be comparable. Although targeting integrin $\alpha_{IIb}\beta_3$ –VWF interaction like LJ-P5, 152B6 appears to be a bit weaker, rendering 85% inhibition. To this point, it is unclear to us why, although we cannot exclude the possibility that 152B6 cannot completely block VWF binding to integrin $\alpha_{IIb}\beta_3$. Due to the lack of evidence, we decided not to make any speculation in the manuscript.

On the other hand, indeed as you pointed out, although 7E9, LJ-155B39 and LJ-134B29 all target Fg, 7E9 manifests much more potent inhibition than the other two. As explained in the original manuscript, this is because these three antibodies target three integrin-binding epitopes in Fg (Fig. R10): “7E9, LJ-155B39 and LJ-134B29 inhibit integrin $\alpha_{IIb}\beta_3$ –Fg interaction by respectively blocking one of the three integrin-binding sites in Fg: γ 408-411 (AGDV), A α 95-98 (RGDF) and A α 572-575 (RGDS)^{79,80}.” It is well acknowledged that γ 408-411 (AGDV) serves as the primary motif for integrin $\alpha_{IIb}\beta_3$ binding, which is supported by the observation that disrupting the AGDV-containing motif, but not mutating either of the two RGD-containing motifs, effectively impair the

ability of fibrinogen to support ADP-induced platelet aggregation⁸¹. However, here we show that LJ-155B39 and LJ-134B29 can also inhibit biomechanical platelet aggregation, albeit their potency is still weaker than 7E9. This suggests that the two RGD-containing motifs may only play a negligible role in agonist-induced platelet aggregation, but becomes more important in biomechanical platelet aggregation. It makes sense because agonists like ADP would directly activate integrin $\alpha_{IIb}\beta_3$ to the fully active state, for which the AGDV-containing motifs alone would be sufficient to mediate strong platelet crosslinking. In contrast, the initial phase of biomechanical platelet aggregation is mainly driven by intermediate state integrin $\alpha_{IIb}\beta_3$ whose affinity is much lower than the fully active state³². As a result, platelet crosslinking by the AGDV-containing motifs alone may become deficient, and the RGD-containing motifs can make extra contributions.

Figure R10. Integrin binding sites within fibrinogen and their respective inhibition by 7E9, LJ-155B39 and LJ-134B29. Adapted from Ref⁸².

Action taken:

1) We added the above discussion in the Discussion section:

“Our observation that inhibiting the two RGD sequences in Fg effectively reduces the thrombus size contrasts with the previous report that mutating either of these two sequences does not impair Fg function in mediating ADP-induced platelet aggregation⁸¹. This is likely because ADP activates integrin $\alpha_{IIb}\beta_3$ to the fully active state, while biomechanical platelet aggregation is mainly driven by intermediate state integrin $\alpha_{IIb}\beta_3$ ³², so that the RGD sequences in Fg are redundant in the former scenario for platelet crosslinking but become a useful supplement to the Fg AGDV sequences in the latter. This suggests that Fg activity is mechanistically distinct during biomechanical

versus biochemical platelet aggregation and unravels an underestimated contribution of the Fg RGD sequences to arterial thrombosis.”

2) In results, we added the statement that the roles of integrin $\alpha_{IIb}\beta_3$ -VWF and $\alpha_{IIb}\beta_3$ -Fg interactions in biomechanical platelet aggregation are “comparably important”.

7. Your study introduces an “effect barcode” system to depict the influence of different factors on the thrombus profile. How does this system aid in interpreting data and informing clinical or therapeutic decisions?

We have addressed a similar concern from Reviewer 1 in their 3rd comment. We respectfully ask you to read our original response in Page 9. To avoid redundancy, here we will just briefly go over the key points.

The barcode system is useful in the following aspects:

1) Anti-thrombotic drug evaluation As we pointed out in our response to the 3rd comment of Reviewer 1, size is not the only important parameter of a thrombus. Abnormal levels of platelet activation, and presumably abnormal levels of Fg and VWF in the thrombus would also indicate critical prothrombotic risks and induce life-threatening issues. For example, drugs can be effective in reducing the thrombus size but in the meantime elevate platelet activation, which would bring extra risks to patients' health^{20,21}. By acquiring the effect barcodes of anti-thrombotic drug candidates, we will have a clearer understanding of their desirable and undesirable effects.

2) Patient prothrombotic status characterization People carrying risk factors of arterial thrombosis can have abnormalities in their thrombus barcode. Detecting these abnormalities would provide valuable information regarding the patient's prothrombotic status and define the starting point for mechanistic investigations. Taking the study of hypertension in our work as an example, it was the identification of hypertension patients' thrombus barcode, [+ + 0 0 0 + +], with a high E⁺ $\alpha_{IIb}\beta_3$ expression, that inspired us to eventually uncover the hyperactivity of GPIIb/IIIa-integrin $\alpha_{IIb}\beta_3$ mechanosensing axis as a new mechanism contributing to the high risk of CVD in hypertension patients. Only measuring the thrombus size would not provide us with any specific mechanistic cues.

3) Personalized drug selection Acquiring the thrombus barcode of a patient may also help clinicians decide what drug or combination of drugs can reach higher efficacy and less side-effect in this patient. For example, as demonstrated in Fig. 6a,b and Supplementary Figure 10 of the original manuscript (current Supplementary Figure 12), NMC4 and 7E3 both inhibited the intensified biomechanical thrombogenesis of hypertension patients. However, only NMC4 but not 7E3 can correct the over-activation

of integrin $\alpha_{IIb}\beta_3$, indicating that abciximab (a derivative of 7E3 approved for clinical use) may not be the optimal drug for reducing the thrombotic risks in hypertension patients. We suspect that the best-matching drug/regimen likely varies from patient to patient, and we need to develop a library of anti-thrombotic agents to accommodate the need of different patients.

4) Drug screening and drug target identification. As we stated in the original manuscript, an interesting attribute of the barcode system is that, “the effect barcode of each anti-thrombotic agent is dictated by its target rather than its pharmacological design”. This point is supported by many lines of experimental evidence in the paper, which are summarized in Figure 3f. We further explained in the manuscript that, this principle enables our assay to screen anti-thrombotic agents against biomechanical platelet aggregation and use their effect barcodes to reason backward their targets.

Action taken:

All the above points have been discussed in the original manuscript. In response to Reviewer 1’s concern, we have made some revisions to make these points clearer while avoiding over-statement. For your convenience, here we list the revisions again:

1) The following sentence in Discussion:

“In this context, our work further showcases the ability of the assay to identify the sources/mechanisms of prothrombotic tendency and suggest counteractive prevention/treatment strategies.”

was revised to:

“In this context, the detection of integrin $\alpha_{IIb}\beta_3$ over-activation and the identification of “treatment mismatch” using the barcode system further showcase the assay’s ability in identifying the mechanisms of prothrombotic tendency and evaluate counteractive prevention/treatment strategies.”

2) The following sentence in Discussion:

“This principle enables our assay to act as a one-step platform to screen anti-thrombotic agents against biomechanical platelet aggregation and use their effect barcodes to reason backward their targets. By the same token, effect barcodes can be used as identifiers of different contributing factors of biomechanical platelet aggregation, which allows the assay to pinpoint the functional context of new mediators and mechanisms, tackling the limitation of existing similar assays in providing mechanistic implications of arterial thrombosis.”

was simplified to:

“These principles may enable us to use the effect barcode to quickly narrow down the possible target(s) of uncharacterized anti-thrombotic agents.”

3) The following sentence in Discussion:

“Developing anti-thrombotics with diversified effect barcodes and determining their interactions with thrombosis-exacerbating factors can help avoid treatment mismatch, in which the ‘addition rule’ could be greatly helpful for prediction.”

was revised to:

“Developing diversified anti-thrombotics and determining their effect barcodes and their interactions with thrombosis-exacerbating factors can help avoid treatment mismatch, in which the ‘addition rule’ could be greatly helpful for prediction.”

8. In Figure 3, the results support an “addition rule” where the combined effects of inhibitors can be predicted by summing the effect barcodes of individual inhibitors. Could you elaborate on the mechanistic or biochemical foundations supporting this rule?

The “addition rule” is originally applied in mathematics, e.g., in the calculation of matrices and vectors, where the basic premise is that different elements in the array of a matrix or a vector do not interfere with each other, so that the addition calculations can be independently carried out within each element. Similarly, in our work, the “addition rule” seems to indicate that the contributions of different molecular interactions to thrombus development and the involved intraplatelet signaling pathways are relatively independent and parallel, instead of having synergy or discord.

Action taken:

We added the following statement in Discussion:

“Moreover, the observed “addition rule” suggests a lack of synergy or discord when concurrently inhibiting multiple targets, indicating that different molecular interactions and signaling pathways function in relatively independent and parallel ways.”

9. In Figure 4, the study notes that both aging and hypertension are associated with increased thrombus size, fibrinogen (Fg) levels, and activation of integrin $\alpha\text{IIb}\beta\text{3}$, without corresponding changes in hematocrit or platelet count. Could you elaborate on the potential physiological or cellular mechanisms that account for these specific increases

in older and hypertensive individuals? What implications might these findings have for our understanding of the underlying processes of thrombogenesis in these populations?

The slightly increased level of Fg in the thrombi is likely due to the elevated Fg level in hypertensive and older subjects' plasma^{83,84} as well as higher integrin $\alpha_{IIb}\beta_3$ activity that more efficiently recruits Fg.

Regarding what causes the intensified biomechanical platelet aggregation associated with hypertension and aging, we have several hypotheses:

1) Chronic low-grade inflammation and high oxidative stress are prevalent in individuals with hypertension and aging^{85,86}. Results from several studies even showed a likely causal relationship between inflammation and the development of hypertension^{87,88}. On the other hand, inflammation can cause platelet activation by exposing the platelets to lipid mediators, cytokines, chemokines and ADP released by activated leucocytes, endothelial cells and perivascular cells⁸⁹. This may attribute to the baseline activation of integrin $\alpha_{IIb}\beta_3$ observed in hypertension patients.

2) Essential hypertension is frequently associated with metabolic abnormalities, which suggests that hypertension may increase the risk of metabolic disorders⁹⁰. Also, aging causes metabolic changes⁹¹. On the other hand, metabolic disorders can change protein glycosylation, e.g., reduction of galactose and sialic acids as generally seen in aged people⁹². Therefore, it is possible that hypertension and aging induce modified glycosylation of GPIb α , resulting in its hyperactivity and the over-activation of integrin $\alpha_{IIb}\beta_3$ in hypertensive and elder people's biomechanical thrombi.

3) The mechanosensitive ion channel Piezo1 was found to be more active in hypertensive than normotensive mice⁷. This may allow the platelets to be more sensitive to mechanical force in general, triggering stronger activation under shear flow.

4) Inflammation, oxidative stress and/or high blood pressure also cause endothelial dysfunction and VWF expression on endothelium surface⁹³, which can lead to platelet adhesion and mechano-activation.

Action taken: the following statement in Discussion:

“The cause of GPIb α and integrin $\alpha_{IIb}\beta_3$ hyperactivity in hypertension patients warrants further investigation, which is possibly relevant to hypertension-associated oxidative stress and inflammation^{94,95}, dysregulated glycosylation of the receptors by metabolic complications⁹⁶, and/or the activation of mechanosensitive ion channel Piezo1⁷.”

was revised to:

“The causes of GPIb α and integrin $\alpha_{IIb}\beta_3$ hyperactivity in hypertension patients as well as the similar trend of platelet hyperreactivity in older people warrant further

investigation, which are possibly relevant to hypertension/aging-associated oxidative stress and inflammation that cause platelet pre-activation^{86,94,95}, dysregulated glycosylation of GPIIb/IIIa and integrin $\alpha_{IIb}\beta_3$ by metabolic disorders^{91,96}, and/or the activation of mechanosensitive ion channel Piezo1 that causes platelet hyper-sensitivity to shear force⁷. On the other hand, the slightly higher Fg level in hypertensive and older subjects' thrombi is likely due to the elevated Fg plasma concentration in these populations^{83,84} as well as integrin $\alpha_{IIb}\beta_3$ hyperactivity that more efficiently recruits Fg.”

10. In Figure 5, despite observing significant impacts of hypertension on thrombus profiles, the study found no significant correlations between thrombus size or E+ $\alpha_{IIb}\beta_3$ levels and other cardiovascular risk factors such as HbA1C, BMI, or lipid levels. How do you interpret these findings within the broader context of cardiovascular health?

Due to our intention of focusing on hypertension, most of the patients enrolled in this study only had hypertension but no other disease that is related to CVD, and their HbA1C, BMI, and lipid levels were all in the healthy range (Supp. Table 3). The intention of our correlation analysis was only to rule out the alternative possibility that another health factor is required to cooperate with hypertension to achieve the observed thrombus profile abnormality, thus confirming that hypertension can independently cause intensified biomechanical thrombogenesis and thrombus profile abnormality. Because of the small number of patients with abnormal HbA1C (n =1), BMI (n =4), or lipid (n =2) levels, our data cannot draw any conclusion regarding whether a co-morbidity of diabetes, obesity or dyslipidemia can have extra contributions to the thrombus profile abnormality, which requires the enrollment of different subject cohorts.

Action taken:

1. To further validate the above point, we did extra analysis to confirm that compared with healthy young subjects, the thrombi of hypertensive subjects who have systolic and diastolic blood pressures and HbA1C, BMI, total cholesterol, LDL-C, HDL-C and triglyceride levels all in the normal range still have larger sizes and higher E+ $\alpha_{IIb}\beta_3$ levels regardless of aging (Fig. R11). These results were added to the manuscript as Supplementary Figure 7c.

Figure R11. Comparing the total platelet signal intensity and normalized E⁺ α_{IIb}β₃ signal intensity of thrombi from the blood of healthy young subjects (n =33) and from hypertensive young (n =5) or older (n =5) subjects who have normal systolic and diastolic blood pressures and HbA1C, BMI, total cholesterol, LDL-C, HDL-C and triglyceride levels. P-values: results of one-way ANOVA (left: F-value =18.4; degrees of freedom =42; right: F-value =18.4; degrees of freedom =42) and multiple comparison.

2. To avoid misunderstanding, we revised the corresponding description in Results to following:

“Most hypertension patients enrolled in this study had their blood pressure well controlled by medication (systolic/diastolic <140/90 mmHg, respectively) and had their hemoglobin A1C (HbA1C), body mass index (BMI) and cholesterol levels within the healthy range (Fig. 4f; Supp. Table 3). Furthermore, neither the size nor the E⁺ α_{IIb}β₃ level of these patients’ thrombi was correlated with the disease duration, systolic or diastolic blood pressure, or the sum of the two, or the patients’ HbA1C level, BMI, total cholesterol, low-density lipoprotein cholesterol (LDL-C), high-density lipoprotein cholesterol (HDL-C) or triglyceride levels (Fig. 4f, Supp. Fig. 7a,b). Also, the thrombi of hypertensive subjects who have systolic and diastolic blood pressures and HbA1C, BMI, and cholesterol levels all in the normal ranges still have larger sizes and higher E⁺ α_{IIb}β₃ levels than healthy young subjects, regardless of aging (Supp. Fig. 7c). These results indicate that hypertension can independently cause intensified biomechanical thrombogenesis and thrombus profile abnormality even with relatively short disease duration and effective antihypertensive medication. However, we cannot exclude the likelihood that poorly controlled blood pressure, diabetes (high HbA1C level), obesity (high BMI) or dyslipidemia (abnormal cholesterol levels) can have extra contributions to

the thrombus profile abnormality, especially considering that the latter three diseases are known risk factors of CVD.”

11. The results suggest that hyperactivity of GPIIb/IIIa and integrin $\alpha_{IIb}\beta_3$ leads to increased platelet activation in hypertension due to enhanced mechanosignaling. How might these changes impact the overall hemostatic balance and risk of thrombus formation in hypertensive patients?

We believe that this leads to a prothrombotic status in hypertensive patients, which directly contributes to their higher risks of CVD than healthy individuals. This point has been included in the third paragraph of Discussion:

“Considering the central roles of GPIIb/IIIa and E⁺ integrin $\alpha_{IIb}\beta_3$ in biomechanical platelet aggregation³², these results explain the intensified biomechanical thrombogenesis observed in hypertension patients’ blood, and suggest that GPIIb/IIIa-integrin $\alpha_{IIb}\beta_3$ mechanosensing axis hyperactivity directly contributes to the high incidence rate of CVD in hypertension patients.”

12. The discussion emphasizes the unique capabilities of the thrombus profiling assay in evaluating biomechanical platelet aggregation. Could you discuss how this assay compares with conventional methods regarding sensitivity, specificity, and predictive accuracy for clinical outcomes?

Currently, no bioassay or biomarker is available in standard clinical settings for evaluating risks of thrombosis. Below we list all relatively common laboratory and point-of-care hematological function assays:

- 1) Targeting coagulation: conventional coagulation assays;
- 2) Targeting soluble agonist-induced platelet aggregation: aggregometry assays, PFA-100/200⁶⁶;
- 3) Targeting both coagulation and soluble agonist-induced platelet aggregation: global coagulation assays (e.g., viscoelastic tests including thromboelastography (TEG) and thromboelastometry (TEM))⁹⁷⁻⁹⁹, seer sonorheometry (e.g., the Quantra Hemostasis System)⁶⁵;
- 4) Targeting both coagulation and biomechanical platelet aggregation: global (or Gorog) thrombosis test¹⁰⁰;
- 5) Targeting biomechanical platelet aggregation: microfluidic stenosis assay (the prototype of our thrombus profiling assay)^{31,32}, microfluidic post assay¹⁰¹.

Among these assays, conventional coagulation assays and aggregometry are unreliable in predicting thrombosis or major adverse cardiovascular events as indicated by previous literature¹⁰²⁻¹⁰⁴. Global thrombosis test, which claims to recapitulate all elements of arterial thrombosis, rendered disappointing discrepancy in evaluating risks

of thrombosis: its comparison between Western and Japanese subjects showed an obvious difference in occlusion time; however, its comparison between healthy donors and cancer patients showed no statistically significant difference in occlusion time¹⁰⁵. Finally, PFA-100/200, global coagulation assays and seer sonorheometry have only limited data supporting their predictive performance. Viscoelastic tests were approved by FDA in 2021 for use only on COVID-19 patients, where the primary goal was the assessment of bleeding and thrombosis following a traumatic injury or event. Only 0.0023% of T2DM patients have had a TEG panel performed as a trial, according to TriNetx Diamond Network data¹⁰⁶. The Quantra Hemostasis System was also only approved on COVID-19, perioperative, and trauma patients. Some of the main reasons why these assays cannot be used in standard clinical settings for evaluating thrombosis include: large interlaboratory variation, high cost, low sensitivity, lack of standardization, and/or are sensitive to pre-analytical variables^{104,107}. Lastly, no assay targeting biomechanical platelet aggregation, including the microfluidic stenosis assay, was used to assess the prothrombotic status of patients. In summary, so far, no assay has been proven to reliably evaluate thrombotic risks, while conventional coagulation assays, aggregometry, and global thrombosis test in fact showed convincing evidence against it. As an advancement, our thrombus profiling assay was demonstrated to accurately detect the prothrombotic status of hypertensive and elder subjects.

Action taken:

The following statement was added in Introduction:

“while conventional coagulation assays and aggregometry assay were already indicated to be unreliable in predicting thrombosis or major adverse cardiovascular events, the new generation of hematological function assays (e.g., global coagulation assays, seer sonorheometry) also have limited evidence supporting their performance, and contain major drawbacks such as high cost, low sensitivity, and lack of standardization^{104,107}.”

What are some potential limitations or challenges that could be encountered when integrating this assay into routine clinical practice?

Potential limitations or challenges when integrating this assay into routine clinical practice include:

- 1) The need of a multi-channel fluorescence microscope and other accessories;
- 2) Making microfluidic chips and conjugating antibodies are labor-intensive;
- 3) Substantial training is required before one can correctly perform the test and analyze the data.

All the above limitations and challenges can be addressed by automation during the translation of the assay for commercialization. For instance, with sufficient funding, we could establish a robotic production line to make the microfluidic chips and fluorophore-conjugated sensors in large scale, and package the experimental platform (including the microscope, the syringe pump, the tubing, etc.) into a compatible automated machine that can load, test and analyze blood samples with a one click operation.

Action taken:

The following statement was added in Discussion:

“Hardware upgrades, e.g., using a multi-channel syringe rack and a motorized stage or a multi-camera array to reach much higher throughput, and/or system automation, will enable the current setup to become more suitable for clinical practice.”

13. The discussion notes significant contributions from the hyperactivity of GPIIb α and integrin α IIb β 3 to thrombosis in hypertension patients. How does this hyperactivity compare to other recognized mechanisms of thrombosis in hypertension?

The following pathological changes were reported to be associated with essential hypertension and claimed to contribute to the increased risk of thrombosis¹⁰⁸:

- 1) Endothelial damage due to elevated blood pressure and chronic inflammation.
- 2) Increased blood viscosity which increases the shear stress.
- 3) Endogenous platelet activation. Evidence collected by previous studies include morphological and biochemical (e.g., increased urinary 11-dehydrothromboxane B2, serum β -thromboglobulin, and basal Ca^{2+}) changes, P-selectin expression (only inspected in malignant and untreated hypertension patients¹⁰⁹), increased agonist-induced platelet aggregation and monocyte–platelet aggregate formation, and diminished sensitivity to exogenous nitric oxide. However, integrin activation was never reported. One work in fact reported that they did not observe native integrin α IIb β 3 activation in hypertension patients, likely because they used PAC-1 instead of MBC370.2 as the indicator which lacks the sensitivity¹¹⁰.
- 4) Abnormalities in the coagulation and fibrinolysis pathways.

In contrast to the previously identified potential mechanisms, our work for the first time detected native α IIb β 3 activation in hypertension patients. We also for the first time report GPIIb α abnormality in hypertension patients.

It is important to note that, the fact that multiple pathogenic mechanisms can potentially contribute to thrombosis does not compromise the usefulness of our thrombus profiling assay, because of two reasons. Firstly, our assay actually covers most of the above

mechanisms to different extents: 1) Endothelial damage and inflammation directly facilitate low-level platelet activation, which contributes to biomechanical platelet aggregation by activating integrin $\alpha_{IIb}\beta_3$; 2) increased blood viscosity is a contributing factor in our assay because it uses whole blood for testing; 3) Endogenous platelet activation contributes to biomechanical platelet aggregation also by activating integrin $\alpha_{IIb}\beta_3$. Secondly, as we argued above, the relative importance of different mechanisms to thrombosis vary in the scenarios of arterial vs. venous thrombosis. The only mechanism not covered by our assay, 4) abnormal coagulation and fibrinolysis should be more important to venous thrombosis but less so to arterial thrombosis, because the fast shear flow prevents the local accumulation of coagulation factors.

Action taken:

The following statement was added in Discussion:

“Among multiple postulated mechanisms, abnormal platelet activation has been identified as a central contributor to the prothrombotic status of hypertension patients, where changes in platelet morphology and biochemical activities (e.g., elevated sensitivity to soluble agonists, reduced sensitivity to exogenous nitric oxide) were reported¹⁰⁸. In comparison, here we...”

14. E+ $\alpha_{IIb}\beta_3$ has been identified as a potential biomarker for arterial thrombosis due to its strong correlation with thrombus size and prevalence in hypertension patients. What steps are necessary to validate E+ $\alpha_{IIb}\beta_3$ for clinical application? Additionally, how does this marker compare with others such as P-selectin and Act. $\alpha_{IIb}\beta_3$ in terms of diagnostic accuracy and prognostic value?

Both P-selectin and Act. $\alpha_{IIb}\beta_3$ have limited potential as biomarkers of arterial thrombosis due to low sensitivity. PAC-1 (detecting Act. $\alpha_{IIb}\beta_3$) was used to successfully evaluate platelet response to anti-platelet treatments (e.g., abciximab and warfarin) in patients that have already developed thrombotic complications and cardiovascular diseases^{111,112} or platelet status in patients undergoing treatments that elevate the risk of thrombosis¹¹³. However, in the attempt to distinguish patients with acute clinical pulmonary embolism from healthy individuals, it showed low sensitivity¹¹⁴. Also, as mentioned above, PAC-1 is not able to detect native integrin $\alpha_{IIb}\beta_3$ activation in hypertension patients, also indicative of its low sensitivity¹¹⁰. On the other hand, platelet P-selectin expression is correlated to several diseases associated with a high risk of thrombosis (e.g., COVID-19 and malignant and untreated hypertension)^{109,115-117}, and has been tested for the potential of predicting and evaluating thrombosis^{118,119}. However, its sensitivity was also found to be less satisfactory in the clinical

settings^{114,120}. As demonstrated in our work, only E⁺ $\alpha_{IIb}\beta_3$ but not P-selectin or Act. $\alpha_{IIb}\beta_3$ can be detected on the platelets of hypertension patients well controlled by medication, which strongly indicates the higher sensitivity of E⁺ $\alpha_{IIb}\beta_3$ than P-selectin and Act. $\alpha_{IIb}\beta_3$. Due to word limit, the above points were made succinctly in Discussion (4th paragraph) of the original manuscript.

To validate E⁺ $\alpha_{IIb}\beta_3$ for clinical application, we need to run flow cytometry and thrombus profiling assay in parallel on patients' (hypertension, diabetes, cancer, etc.) and healthy subjects' blood samples and check: 1) whether the patients natively express E⁺ $\alpha_{IIb}\beta_3$ on their platelets; and 2) whether a positive correlation can be found between E⁺ $\alpha_{IIb}\beta_3$ level (assessed by flow cytometry) and thrombus size (assessed by thrombus profiling assay). Notably, it is possible that certain CVD risk factors do not induce native E⁺ $\alpha_{IIb}\beta_3$ expression. For instance, our study identified no elevated activity of integrin $\alpha_{IIb}\beta_3$ in older people, suggesting that their $\alpha_{IIb}\beta_3$ integrins are not activated to the E⁺ state. Thus, patient cohorts need to be tested one by one to determine the application range of E⁺ $\alpha_{IIb}\beta_3$ as a biomarker for arterial thrombosis.

Action taken:

The following statement was added in Discussion:

“To validate this application requires an investigation on the correlation between native E⁺ $\alpha_{IIb}\beta_3$ expression (assessed by flow cytometry) and thrombus size (assessed by thrombus profiling assay) in different patient cohorts.”

Reviewer #3 (Remarks to the Author):

The manuscript by Din et al. proposes a new methodology to analyze a growing thrombus in a stenotic microfluidic model, specifically by looking at several of the key players in a growing clot, such as platelets, fibrinogen, integrin $\alpha_{IIb}\beta_3$ conformation, and VWF. In addition, this work further investigates an innovative concept related to biomechanical platelet aggregation, a unique phenomenon in which shear and GPIIb signaling leads to thrombus growth primarily through an intermediate affinity of $\alpha_{IIb}\beta_3$. Uniquely, thrombus growth is not affected by conventional antiplatelets or amplification loop blockers. Here, the phenomenon is further explored with a series of inhibitor studies, and then further shown to have potential clinical relevance via a series of studies on clinical samples from hypertensive patients, both with their novel device showing differences in stenotic clot formation, then with follow-up mechanistic studies using a biomembrane force probe, which is a technically challenging approach.

This is an important contribution for several reasons. First, the assay itself represents a more wholistic approach to examining clot formation, showing interesting differences in the relative ratio of key components and platelet integrin states. Second, this assay provides new insights into the platelet reactivity of hypertensive patients, and highlights a potential need to explore new antiplatelet strategies. Finally, the tool developed here may find use in helping to screen new anti-platelet approaches, as well as serve as a tool to tailor individual anticoagulant strategies to patients, although a significant development effort would be needed for clinical use.

Owing to the rigor and well-designed studies, I have minor comments aimed at improving the accessibility of the manuscript:

We sincerely thank you for recognizing the significance and importance of the methodology developed in our work as well as our scientific discoveries.

1. If there is space in Figure 2, I wonder if panel B could be moved below panel A and a small table with more specific details on what is targeted by the inhibitors could be added. My only reason for asking is that I was initially confused as to why NMC4 targets VWF on the GP1ba axis, and 152B6 targets VWF on the α IIb β 3 axis. This was subsequently cleared up when reading lines 154-160, but I found myself creating a table to keep track of the various inhibitors.

Thank you for this suggestion.

Action taken:

The following table was added to the right side of Fig. 2a together with the corresponding figure legend: “(Right) Table layout of tested antibodies and their respective antigens and targeting receptor–ligand interactions.”

Antibody	Antigen	Target
AK2	GPIb α	GPIb α - VWF
NMC4	VWF A1 domain	
LJ-P5	Integrin $\alpha_{IIb}\beta_3$	$\alpha_{IIb}\beta_3$ - VWF
152B6	VWF C4 domain	
7E9	Fg γ 408-411 (AGDV)	$\alpha_{IIb}\beta_3$ -Fg
LJ-155B39	Fg A α 95-98 (RGDF)	
LJ-134B29	Fg A α 572-575 (RGDS)	

2. Lines 205-207: I also agree that the barcode summation idea is interesting. In this second example of addition, one of the addends (GPIba-VWF) equals the polystyrene barcode, so it feels akin to saying $A + 0 = A$ (although that is a simplification). Noting that this concept is revisited in Figure 6, I recommend providing a hint to readers that this concept will be further tested later in the paper. (If a reader linearly examines the manuscript, this concept may come across as unconvincing at this point)

Thank you for this suggestion.

Action taken:

We added the following sentence at the end of the section “*Identifying an ‘addition rule’ in the effect barcode system*”:

“This addition rule will be further validated below in drug-disease interactions.”

3. In Figure 4, I found the labeling related to “Slope non-zero?” a bit confusing at first. I now understand that the null hypothesis is that the slope was zero, but perhaps other wording may help with this.

Thank you for this suggestion.

Action taken:

We changed “Slope non-zero?” to “Positive/Negative correlation?” in all related panels of Fig. 4 and new Supp. Fig. 7.

4. In Figure 4H, I expected the number of healthy young donors to match 4A, were these separate studies?

We recruited a total of 33 healthy young donors, as reflected in Fig. 4h. This matches the number of healthy young donors (≤ 49 years old) in Fig. 4a (15, 13 and 5 donors in the age range of ≤ 29 , 30-39, and 40-49, respectively). Please note that Fig. 4a also includes healthy older donors (50 years old and above) to inspect how aging affects the thrombus size. You probably also added the number of these donors when calculating the group size.

5. In Figure 5, I understand the logic that in the flow chambers, there is not a large difference between hypertensive young/old platelet behaviors. However, Figure 4 shows that there are likely differences between these groups. I would suggest qualifying that hypertensive patients were not tested separately, but that an effect may exist that could be identified with later studies.

Thank you for this suggestion.

Action taken:

We added the following statement:

“However, this does not exclude the possibility that aging can influence hypertensive patients’ GPIIb/IIIa and integrin $\alpha IIb\beta 3$ as a secondary factor, which shall be inspected in later studies.”

6. In the BFP and fBFP experiments, I had a hard time telling if each dot represented a single platelet from a single donor, or a single platelet from the same donor. Could you clarify?

We thank you for pointing this out, which is indeed confusing. The dots represent a mixture of multiple platelets from multiple donors.

Action taken:

We replotted the related panels (Fig. 5f,i,n) so that different symbol colors are used to indicate different donors. Accordingly, we added the following statement in the figure legend:

“f,i,n: different symbol colors indicate data collected from different subjects.”

7. Figure 6C – the idea of personal thrombus barcodes is very interesting, and I appreciate the rigor in showing the proportion of patients that fit the proposed schema of “normal” values. With that said, panel c was a bit difficult to understand at first glance. My suggestions would be to consider a prior panel visually illustrating how high, normal, low are defined, and also including the number of patients represented by these tests (I believe it’s 32, but don’t see it in the figure or caption). Other possibilities include a title for panel c to orient the reader (akin to those in Figure 5), and using the labels healthy, healthy older, instead of HY, HO etc.

We thank you for the helpful suggestions.

Action taken:

1) We added a panel (Fig. R12) as the new Fig. 6c to illustrate how values being high, normal, low are defined.

Figure R12. Illustration of how values in the thrombus profiles are defined as high, normal or low. Illustration of how values in the personal thrombus profiles being low, normal or high are defined. Thrombus profiles of healthy young subjects were used as the reference, the values of which are fitted to a Gaussian distribution. Mean \pm 2s.d. range is defined as normal, and values lower or higher are defined as abnormally low and high, respectively.

2) We changed ‘HY’, ‘HO’, ‘TY’ and ‘TO’ to ‘Healthy young’, ‘Healthy older’, ‘HTN young’ and ‘HTN older’, respectively.

8. Methods: I may have missed this, but what are the dimensions of the microfluidic structure, I was specifically looking for the height of the channels, although also

providing more details on the geometry tested would help others seeking to replicate these results.

Each channel has a width of 200 μm and a height of 50 μm . This information was provided in the first sentence of Results. However, we noticed that it was not specified what dimensions the '200 μm ' and '50 μm ' correspond to.

Action taken:

To make the information clearer, we made the following revision:

“Our microfluidic chip is composed of 10 rectangular (**width** \times **height**: 200 μm \times 50 μm) channels...”

References:

- 1 Jain, A. *et al.* A shear gradient-activated microfluidic device for automated monitoring of whole blood haemostasis and platelet function. *Nature communications* **7**, 10176, doi:10.1038/ncomms10176 (2016).
- 2 Brazilek, R. J. *et al.* Application of a strain rate gradient microfluidic device to von Willebrand's disease screening. *Lab on a chip* **17**, 2595-2608, doi:10.1039/c7lc00498b (2017).
- 3 Das, D. *et al.* Endothelial dysfunction, platelet hyperactivity, hypertension, and the metabolic syndrome: molecular insights and combating strategies. *Front Nutr* **10**, 1221438, doi:10.3389/fnut.2023.1221438 (2023).
- 4 Wagner, D. D. & Burger, P. C. Platelets in inflammation and thrombosis. *Arteriosclerosis, thrombosis, and vascular biology* **23**, 2131-2137, doi:10.1161/01.ATV.0000095974.95122.EC (2003).
- 5 Zhang, S. *et al.* SARS-CoV-2 binds platelet ACE2 to enhance thrombosis in COVID-19. *Journal of hematology & oncology* **13**, 120, doi:10.1186/s13045-020-00954-7 (2020).
- 6 Natarajan, A., Zaman, A. G. & Marshall, S. M. Platelet hyperactivity in type 2 diabetes: role of antiplatelet agents. *Diab Vasc Dis Res* **5**, 138-144, doi:10.3132/dvdr.2008.023 (2008).
- 7 Zhao, W. *et al.* Piezo1 initiates platelet hyperreactivity and accelerates thrombosis in hypertension. *Journal of thrombosis and haemostasis : JTH*, doi:10.1111/jth.15504 (2021).
- 8 Ju, L. *et al.* Compression force sensing regulates integrin α IIb β 3 adhesive function on diabetic platelets. *Nature communications* **9**, 1087, doi:10.1038/s41467-018-03430-6 (2018).
- 9 Koupenova, M., Kehrel, B. E., Corkrey, H. A. & Freedman, J. E. Thrombosis and platelets: an update. *European heart journal* **38**, 785-791, doi:10.1093/eurheartj/ehw550 (2017).
- 10 Jackson, S. P. Arterial thrombosis--insidious, unpredictable and deadly. *Nature medicine* **17**, 1423-1436, doi:10.1038/nm.2515 (2011).
- 11 Zhou, Y. *et al.* The Emerging Role of Neutrophil Extracellular Traps in Arterial, Venous and Cancer-Associated Thrombosis. *Front Cardiovasc Med* **8**, 786387, doi:10.3389/fcvm.2021.786387 (2021).
- 12 Mahalingam, A. *et al.* Numerical analysis of the effect of turbulence transition on the hemodynamic parameters in human coronary arteries. *Cardiovasc Diagn Ther* **6**, 208-220, doi:10.21037/cdt.2016.03.08 (2016).
- 13 Stein, P. D. & Sabbah, H. N. Measured turbulence and its effect on thrombus formation. *Circulation research* **35**, 608-614, doi:10.1161/01.res.35.4.608 (1974).
- 14 Ouriel, K. *et al.* The hemodynamics of thrombus formation in arteries. *Journal of vascular surgery* **14**, 757-762; discussion 762-753, doi:10.1067/mva.1991.33157 (1991).
- 15 Panteleev, M. A. *et al.* Wall shear rates in human and mouse arteries: Standardization of hemodynamics for in vitro blood flow assays: Communication from the ISTH SSC subcommittee on biorheology. *Journal of thrombosis and haemostasis : JTH* **19**, 588-595, doi:10.1111/jth.15174 (2021).
- 16 Ku, D. N. Blood flow in arteries. *Annu. Rev. Fluid Mech.* **29**, 399-434 (1997).

- 17 Yakusheva, A. A. *et al.* Traumatic vessel injuries initiating hemostasis generate high shear conditions. *Blood advances* **6**, 4834-4846, doi:10.1182/bloodadvances.2022007550 (2022).
- 18 Wang, G. R., Yang, F. & Zhao, W. There can be turbulence in microfluidics at low Reynolds number. *Lab on a chip* **14**, 1452-1458, doi:10.1039/c3lc51403j (2014).
- 19 Topol, E. J., Byzova, T. V. & Plow, E. F. Platelet GPIIb-IIIa blockers. *Lancet* **353**, 227-231, doi:10.1016/S0140-6736(98)11086-3 (1999).
- 20 Topol, E. J. *et al.* Randomized, double-blind, placebo-controlled, international trial of the oral IIb/IIIa antagonist lotrafiban in coronary and cerebrovascular disease. *Circulation* **108**, 399-406, doi:10.1161/01.CIR.0000084501.48570.F6 (2003).
- 21 Chew, D. P., Bhatt, D. L., Sapp, S. & Topol, E. J. Increased mortality with oral platelet glycoprotein IIb/IIIa antagonists: a meta-analysis of phase III multicenter randomized trials. *Circulation* **103**, 201-206, doi:10.1161/01.cir.103.2.201 (2001).
- 22 Bougie, D. W., Rasmussen, M., Zhu, J. & Aster, R. H. Antibodies causing thrombocytopenia in patients treated with RGD-mimetic platelet inhibitors recognize ligand-specific conformers of alphaIIb/beta3 integrin. *Blood* **119**, 6317-6325, doi:10.1182/blood-2012-01-406322 (2012).
- 23 Cox, D. *et al.* Evidence of platelet activation during treatment with a GPIIb/IIIa antagonist in patients presenting with acute coronary syndromes. *Journal of the American College of Cardiology* **36**, 1514-1519, doi:10.1016/s0735-1097(00)00919-0 (2000).
- 24 Bosco, A., Kidson-Gerber, G. & Dunkley, S. Delayed tirofiban-induced thrombocytopenia: two case reports. *Journal of thrombosis and haemostasis : JTH* **3**, 1109-1110, doi:10.1111/j.1538-7836.2005.01296.x (2005).
- 25 Lin, F. Y. *et al.* A general chemical principle for creating closure-stabilizing integrin inhibitors. *Cell* **185**, 3533-3550 e3527, doi:10.1016/j.cell.2022.08.008 (2022).
- 26 Comanici, M., Bhudia, S. K., Marczin, N. & Raja, S. G. Antiplatelet Resistance in Patients Who Underwent Coronary Artery Bypass Grafting: A Systematic Review and Meta-Analysis. *Am J Cardiol* **206**, 191-199, doi:10.1016/j.amjcard.2023.08.063 (2023).
- 27 Akturk, I. F. *et al.* Hypertension as a risk factor for aspirin and clopidogrel resistance in patients with stable coronary artery disease. *Clin Appl Thromb Hemost* **20**, 749-754, doi:10.1177/1076029613481102 (2014).
- 28 Liu, X. F. *et al.* Prevalence of and risk factors for aspirin resistance in elderly patients with coronary artery disease. *J Geriatr Cardiol* **10**, 21-27, doi:10.3969/j.issn.1671-5411.2013.01.005 (2013).
- 29 Previtali, E., Bucciarelli, P., Passamonti, S. M. & Martinelli, I. Risk factors for venous and arterial thrombosis. *Blood Transfus* **9**, 120-138, doi:10.2450/2010.0066-10 (2011).
- 30 Cattaneo, M. Resistance to anti-platelet agents. *Thrombosis research* **127 Suppl 3**, S61-63, doi:10.1016/S0049-3848(11)70017-2 (2011).
- 31 Nesbitt, W. S. *et al.* A shear gradient-dependent platelet aggregation mechanism drives thrombus formation. *Nature medicine* **15**, 665-673, doi:10.1038/nm.1955 (2009).

- 32 Chen, Y. *et al.* An integrin α IIb β 3 intermediate affinity state mediates biomechanical platelet aggregation. *Nature materials* **18**, 760-769, doi:10.1038/s41563-019-0323-6 (2019).
- 33 Li, M., Hotaling, N. A., Ku, D. N. & Forest, C. R. Microfluidic thrombosis under multiple shear rates and antiplatelet therapy doses. *PloS one* **9**, e82493, doi:10.1371/journal.pone.0082493 (2014).
- 34 Sakurai, Y. *et al.* A microengineered vascularized bleeding model that integrates the principal components of hemostasis. *Nature communications* **9**, 509, doi:10.1038/s41467-018-02990-x (2018).
- 35 Colace, T. V. & Diamond, S. L. Direct observation of von Willebrand factor elongation and fiber formation on collagen during acute whole blood exposure to pathological flow. *Arteriosclerosis, thrombosis, and vascular biology* **33**, 105-113, doi:10.1161/ATVBAHA.112.300522 (2013).
- 36 Westein, E. *et al.* Atherosclerotic geometries exacerbate pathological thrombus formation poststenosis in a von Willebrand factor-dependent manner. *Proc Natl Acad Sci USA* **110**, 1357-1362, doi:10.1073/pnas.1209905110 (2013).
- 37 Tovar-Lopez, F. J. *et al.* A microfluidics device to monitor platelet aggregation dynamics in response to strain rate micro-gradients in flowing blood. *Lab on a chip* **10**, 291-302, doi:10.1039/b916757a (2010).
- 38 Kim, D. A. & Ku, D. N. Structure of shear-induced platelet aggregated clot formed in an in vitro arterial thrombosis model. *Blood advances* **6**, 2872-2883, doi:10.1182/bloodadvances.2021006248 (2022).
- 39 Receveur, N. *et al.* Shear rate gradients promote a bi-phasic thrombus formation on weak adhesive proteins, such as fibrinogen in a VWF-dependent manner. *Haematologica* **105**, 2471-2483, doi:10.3324/haematol.2019.235754 (2020).
- 40 Ting, L. H. *et al.* Contractile forces in platelet aggregates under microfluidic shear gradients reflect platelet inhibition and bleeding risk. *Nature communications* **10**, 1204, doi:10.1038/s41467-019-09150-9 (2019).
- 41 Berry, J. *et al.* An "occlusive thrombosis-on-a-chip" microfluidic device for investigating the effect of anti-thrombotic drugs. *Lab on a chip* **21**, 4104-4117, doi:10.1039/d1lc00347j (2021).
- 42 Maloney, S. F., Brass, L. F. & Diamond, S. L. P2Y₁₂ or P2Y₁ inhibitors reduce platelet deposition in a microfluidic model of thrombosis while apyrase lacks efficacy under flow conditions. *Integr Biol (Camb)* **2**, 183-192, doi:10.1039/b919728a (2010).
- 43 Neeves, K. B. *et al.* Microfluidic focal thrombosis model for measuring murine platelet deposition and stability: PAR4 signaling enhances shear-resistance of platelet aggregates. *Journal of thrombosis and haemostasis : JTH* **6**, 2193-2201, doi:10.1111/j.1538-7836.2008.03188.x (2008).
- 44 Jain, A. *et al.* Assessment of whole blood thrombosis in a microfluidic device lined by fixed human endothelium. *Biomed Microdevices* **18**, 73, doi:10.1007/s10544-016-0095-6 (2016).
- 45 Flamm, M. H. *et al.* Multiscale prediction of patient-specific platelet function under flow. *Blood* **120**, 190-198, doi:10.1182/blood-2011-10-388140 (2012).
- 46 Muthard, R. W., Welsh, J. D., Brass, L. F. & Diamond, S. L. Fibrin, gamma'-fibrinogen, and transclot pressure gradient control hemostatic clot growth during

- human blood flow over a collagen/tissue factor wound. *Arteriosclerosis, thrombosis, and vascular biology* **35**, 645-654, doi:10.1161/ATVBAHA.114.305054 (2015).
- 47 Blann, A. D., Nadar, S. & Lip, G. Y. Pharmacological modulation of platelet function in hypertension. *Hypertension* **42**, 1-7, doi:10.1161/01.HYP.0000077901.84467.E1 (2003).
- 48 Mehta, J. L., Lopez, L. M., Chen, L. & Cox, O. E. Alterations in nitric oxide synthase activity, superoxide anion generation, and platelet aggregation in systemic hypertension, and effects of celiprolol. *Am J Cardiol* **74**, 901-905, doi:10.1016/0002-9149(94)90583-5 (1994).
- 49 Touyz, R. M. & Schiffrin, E. L. Effects of angiotensin II and endothelin-1 on platelet aggregation and cytosolic pH and free Ca²⁺ concentrations in essential hypertension. *Hypertension* **22**, 853-862, doi:10.1161/01.hyp.22.6.853 (1993).
- 50 Scrobobaci, M. L., Cunescu, V. & Orha, I. Recurrent thromboembolism with spontaneous platelet aggregation. *Thromb Haemost* **36**, 645-646 (1976).
- 51 Pravenec, M., Kunes, J., Zicha, J., Kren, V. & Klir, P. Platelet aggregation in spontaneous hypertension: genetic determination and correlation analysis. *J Hypertens* **10**, 1453-1456, doi:10.1097/00004872-199210120-00003 (1992).
- 52 Bampalis, V. G., Brantl, S. A. & Siess, W. Why and how to eliminate spontaneous platelet aggregation in blood measured by multiple electrode aggregometry. *Journal of thrombosis and haemostasis : JTH* **10**, 1710-1714, doi:10.1111/j.1538-7836.2012.04819.x (2012).
- 53 Nenci, G. G., Berrettini, M., Iadevaia, V., Parise, P. & Ballatori, E. Inhibition of spontaneous platelet aggregation and adhesion by indobufen (K 3920). A randomized, double-blind crossover study on platelet, coagulation and fibrinolysis function tests. *Pharmatherapeutica* **3**, 188-194 (1982).
- 54 Saniabadi, A. R., Lowe, G. D., Barbenel, J. C. & Forbes, C. D. A comparison of spontaneous platelet aggregation in whole blood with platelet rich plasma: additional evidence for the role of ADP. *Thromb Haemost* **51**, 115-118 (1984).
- 55 Zhang, X. F. & Cheng, X. Platelet mechanosensing axis revealed. *Nature materials* **18**, 661-662, doi:10.1038/s41563-019-0393-5 (2019).
- 56 Chen, Y., Li, Z., Kong, F., Ju, L. A. & Zhu, C. Force-Regulated Spontaneous Conformational Changes of Integrins $\alpha_5\beta_1$ and $\alpha_v\beta_3$. *Acs Nano* **18**, 299–313, doi:10.1021/acsnano.3c06253 (2024).
- 57 Rosetti, F. *et al.* A Lupus-Associated Mac-1 Variant Has Defects in Integrin Allostery and Interaction with Ligands under Force. *Cell Reports* **10**, 1655–1664, doi:10.1016/j.celrep.2015.02.037 (2015).
- 58 Banerjee, S., Nara, R., Chakraborty, S., Chowdhury, D. & Haldar, S. Integrin Regulated Autoimmune Disorders: Understanding the Role of Mechanical Force in Autoimmunity. *Frontiers in cell and developmental biology* **10**, 852878, doi:10.3389/fcell.2022.852878 (2022).
- 59 Nandagopal, S. *et al.* C3aR Signaling Inhibits NK-cell Infiltration into the Tumor Microenvironment in Mouse Models. *Cancer immunology research* **10**, 245-258, doi:10.1158/2326-6066.CIR-21-0435 (2022).

- 60 Shimaoka, M. & Springer, T. A. Therapeutic antagonists and conformational regulation of integrin function. *Nature reviews. Drug discovery* **2**, 703-716, doi:10.1038/nrd1174 (2003).
- 61 Fan, Z. *et al.* Neutrophil recruitment limited by high-affinity bent beta2 integrin binding ligand in cis. *Nature communications* **7**, 12658, doi:10.1038/ncomms12658 (2016).
- 62 Fan, Z. *et al.* High-Affinity Bent beta2-Integrin Molecules in Arresting Neutrophils Face Each Other through Binding to ICAMs In cis. *Cell Reports* **26**, 119-130 e115, doi:10.1016/j.celrep.2018.12.038 (2019).
- 63 Li, J. *et al.* Ligand binding initiates single-molecule integrin conformational activation. *Cell*, doi:10.1016/j.cell.2024.04.049 (2024).
- 64 Thomson, E. E. *et al.* Gigapixel imaging with a novel multi-camera array microscope. *eLife* **11**, doi:10.7554/eLife.74988 (2022).
- 65 Volod, O. & Viola, F. The Quantra System: System Description and Protocols for Measurements. *Methods in molecular biology* **2663**, 743-761, doi:10.1007/978-1-0716-3175-1_50 (2023).
- 66 Gorog, D. A. & Becker, R. C. Point-of-care platelet function tests: relevance to arterial thrombosis and opportunities for improvement. *Journal of thrombosis and thrombolysis* **51**, 1-11, doi:10.1007/s11239-020-02170-z (2021).
- 67 Akin, M. & Polat, Y. Platelet function analyser (PFA)-100 closure time in the evaluation of non-steroidal anti-inflammatory drug-induced platelet dysfunction in children with bleeding symptoms. *Blood Transfus* **10**, 545-546, doi:10.2450/2012.0125-11 (2012).
- 68 Litvinov, R. I., Shuman, H., Bennett, J. S. & Weisel, J. W. Binding strength and activation state of single fibrinogen-integrin pairs on living cells. *Proceedings of the National Academy of Sciences of the United States of America* **99**, 7426-7431, doi:10.1073/pnas.112194999 (2002).
- 69 Rengarajan, K., Cristol, S. M., Mehta, M. & Nickerson, J. M. Quantifying DNA concentrations using fluorometry: a comparison of fluorophores. *Mol Vis* **8**, 416-421 (2002).
- 70 Alqarni, A. O., Alkahtani, S. A., Mahmoud, A. M. & El-Wekil, M. M. Design of "Turn On" fluorometric nanoprobe based on nitrogen doped graphene quantum dots modified with beta-cyclodextrin and vitamin B(6) cofactor for selective sensing of dopamine in human serum. *Spectrochim Acta A Mol Biomol Spectrosc* **248**, 119180, doi:10.1016/j.saa.2020.119180 (2021).
- 71 Xing, X. *et al.* Correction of profiles of in-situ chlorophyll fluorometry for the contribution of fluorescence originating from non-algal matter. *Limnology and Oceanography: Methods* **15**, 80-93 (2016).
- 72 Charles, S. The proof and measurement of association between two things. *The American Journal of Psychology* **15**, 72–101 (1904).
- 73 Kendall, M. G. A New Measure of Rank Correlation. *Biometrika* **30**, 81-93 (1938).
- 74 Wong, N. D. *et al.* Atherosclerotic cardiovascular disease risk assessment: An American Society for Preventive Cardiology clinical practice statement. *Am J Prev Cardiol* **10**, 100335, doi:10.1016/j.ajpc.2022.100335 (2022).

- 75 Anjum, M. *et al.* Stroke and bleeding risk in atrial fibrillation with CHA2DS2-VASC risk score of one: the Norwegian AFNOR study. *European heart journal* **45**, 57-66, doi:10.1093/eurheartj/ehad659 (2024).
- 76 Fu, H. *et al.* Flow-induced elongation of von Willebrand factor precedes tension-dependent activation. *Nature communications* **8**, 324, doi:10.1038/s41467-017-00230-2 (2017).
- 77 Yago, T. *et al.* Platelet glycoprotein Ibalpha forms catch bonds with human WT vWF but not with type 2B von Willebrand disease vWF. *The Journal of clinical investigation* **118**, 3195-3207, doi:10.1172/JCI35754 (2008).
- 78 Jackson, S. P. The growing complexity of platelet aggregation. *Blood* **109**, 5087-5095, doi:10.1182/blood-2006-12-027698 (2007).
- 79 Lengweiler, S. *et al.* Preparation of monoclonal antibodies to murine platelet glycoprotein IIb/IIIa (alphallbbeta3) and other proteins from hamster-mouse interspecies hybridomas. *Biochemical and biophysical research communications* **262**, 167-173, doi:10.1006/bbrc.1999.1172 (1999).
- 80 Felding-Habermann, B., Ruggeri, Z. M. & Cheresh, D. A. Distinct biological consequences of integrin alpha v beta 3-mediated melanoma cell adhesion to fibrinogen and its plasmic fragments. *The Journal of biological chemistry* **267**, 5070-5077 (1992).
- 81 Farrell, D. H., Thiagarajan, P., Chung, D. W. & Davie, E. W. Role of fibrinogen alpha and gamma chain sites in platelet aggregation. *Proceedings of the National Academy of Sciences of the United States of America* **89**, 10729-10732 (1992).
- 82 Litvinov, R. I., Farrell, D. H., Weisel, J. W. & Bennett, J. S. The Platelet Integrin alphallbbeta3 Differentially Interacts with Fibrin Versus Fibrinogen. *The Journal of biological chemistry*, doi:10.1074/jbc.M115.706861 (2016).
- 83 Shankar, A., Wang, J. J., Rochtchina, E. & Mitchell, P. Positive association between plasma fibrinogen level and incident hypertension among men: population-based cohort study. *Hypertension* **48**, 1043-1049, doi:10.1161/01.HYP.0000245700.13817.3c (2006).
- 84 Hager, K., Felicetti, M., Seefried, G. & Platt, D. Fibrinogen and aging. *Aging (Milano)* **6**, 133-138, doi:10.1007/BF03324226 (1994).
- 85 Patrick, D. M., Van Beusecum, J. P. & Kirabo, A. The role of inflammation in hypertension: novel concepts. *Curr Opin Physiol* **19**, 92-98, doi:10.1016/j.cophys.2020.09.016 (2021).
- 86 Li, X. *et al.* Inflammation and aging: signaling pathways and intervention therapies. *Signal transduction and targeted therapy* **8**, 239, doi:10.1038/s41392-023-01502-8 (2023).
- 87 Solak, Y. *et al.* Hypertension as an autoimmune and inflammatory disease. *Hypertens Res* **39**, 567-573, doi:10.1038/hr.2016.35 (2016).
- 88 Dinh, Q. N., Drummond, G. R., Sobey, C. G. & Chrissobolis, S. Roles of inflammation, oxidative stress, and vascular dysfunction in hypertension. *Biomed Res Int* **2014**, 406960, doi:10.1155/2014/406960 (2014).
- 89 Stokes, K. Y. & Granger, D. N. Platelets: a critical link between inflammation and microvascular dysfunction. *The Journal of physiology* **590**, 1023-1034, doi:10.1113/jphysiol.2011.225417 (2012).

- 90 Ferrannini, E. & Natali, A. Essential hypertension, metabolic disorders, and insulin resistance. *Am Heart J* **121**, 1274-1282, doi:10.1016/0002-8703(91)90433-i (1991).
- 91 Palmer, A. K. & Jensen, M. D. Metabolic changes in aging humans: current evidence and therapeutic strategies. *The Journal of clinical investigation* **132**, doi:10.1172/JCI158451 (2022).
- 92 Paton, B., Suarez, M., Herrero, P. & Canela, N. Glycosylation Biomarkers Associated with Age-Related Diseases and Current Methods for Glycan Analysis. *International journal of molecular sciences* **22**, doi:10.3390/ijms22115788 (2021).
- 93 Steffes, L. C., Cheng, P., Quertermous, T. & Kumar, M. E. von Willebrand Factor Is Produced Exclusively by Endothelium, Not Neointima, in Occlusive Vascular Lesions in Both Pulmonary Hypertension and Atherosclerosis. *Circulation* **146**, 429-431, doi:10.1161/CIRCULATIONAHA.121.058427 (2022).
- 94 Griendling, K. K. *et al.* Oxidative Stress and Hypertension. *Circulation research* **128**, 993-1020, doi:10.1161/CIRCRESAHA.121.318063 (2021).
- 95 Scherlinger, M., Richez, C., Tsokos, G. C., Boilard, E. & Blanco, P. The role of platelets in immune-mediated inflammatory diseases. *Nature reviews. Immunology* **23**, 495-510, doi:10.1038/s41577-023-00834-4 (2023).
- 96 Schjoldager, K. T., Narimatsu, Y., Joshi, H. J. & Clausen, H. Global view of human protein glycosylation pathways and functions. *Nature reviews. Molecular cell biology* **21**, 729-749, doi:10.1038/s41580-020-00294-x (2020).
- 97 Lopez-Jaime, F. J. *et al.* Clot Stiffness Measured By Seer Sonorheometry As a Marker Of Poor Prognosis In Hospitalized COVID-19 Patients. *Clin Appl Thromb Hemost* **28**, 10760296221112085, doi:10.1177/10760296221112085 (2022).
- 98 Bochen, L., Wiinberg, B., Kjelgaard-Hansen, M., Steinbruchel, D. A. & Johansson, P. I. Evaluation of the TEG platelet mapping assay in blood donors. *Thrombosis journal* **5**, 3, doi:10.1186/1477-9560-5-3 (2007).
- 99 Lipets, E. N. & Ataulakhanov, F. I. Global assays of hemostasis in the diagnostics of hypercoagulation and evaluation of thrombosis risk. *Thrombosis journal* **13**, 4, doi:10.1186/s12959-015-0038-0 (2015).
- 100 Yamamoto, J. *et al.* Gorog Thrombosis Test: a global in-vitro test of platelet function and thrombolysis. *Blood Coagul Fibrinolysis* **14**, 31-39, doi:10.1097/01.mbc.0000046170.06450.9b (2003).
- 101 Ju, L. A. *et al.* Microfluidic post method for 3-dimensional modeling of platelet-leukocyte interactions. *Analyst* **147**, 1222-1235, doi:10.1039/d2an00270a (2022).
- 102 Lim, H. Y., O'Malley, C., Donnan, G., Nandurkar, H. & Ho, P. A review of global coagulation assays - Is there a role in thrombosis risk prediction? *Thrombosis research* **179**, 45-55, doi:10.1016/j.thromres.2019.04.033 (2019).
- 103 Paniccia, R., Priora, R., Liotta, A. A. & Abbate, R. Platelet function tests: a comparative review. *Vasc Health Risk Manag* **11**, 133-148, doi:10.2147/VHRM.S44469 (2015).
- 104 Zhang, Y., Jiang, F., Chen, Y. & Ju, L. A. Platelet Mechanobiology Inspired Microdevices: From Hematological Function Tests to Disease and Drug Screening. *Frontiers in pharmacology* **12**, 779753, doi:10.3389/fphar.2021.779753 (2021).

- 105 Gorog, D. A. *et al.* First direct comparison of platelet reactivity and thrombolytic status between Japanese and Western volunteers: possible relationship to the "Japanese paradox". *Int J Cardiol* **152**, 43-48, doi:10.1016/j.ijcard.2010.07.002 (2011).
- 106 Committee, L. O. I. N. a. C. Thromboelastography LOINC Code 67790-6. (2022).
- 107 Zhang, Y. *et al.* Emerging Microfluidic Approaches for Platelet Mechanobiology and Interplay With Circulatory Systems. *Front Cardiovasc Med* **8**, 766513, doi:10.3389/fcvm.2021.766513 (2021).
- 108 Gkaliagkousi, E., Passacquale, G., Douma, S., Zamboulis, C. & Ferro, A. Platelet activation in essential hypertension: implications for antiplatelet treatment. *Am J Hypertens* **23**, 229-236, doi:10.1038/ajh.2009.247 (2010).
- 109 Lip, G. Y., Edmunds, E., Hee, F. L., Blann, A. D. & Beevers, D. G. A cross-sectional, diurnal, and follow-up study of platelet activation and endothelial dysfunction in malignant phase hypertension. *Am J Hypertens* **14**, 823-828, doi:10.1016/s0895-7061(01)02045-3 (2001).
- 110 Labios, M. *et al.* Flow cytometric analysis of platelet activation in hypertensive patients. Effect of doxazosin. *Thrombosis research* **110**, 203-208, doi:10.1016/s0049-3848(03)00377-3 (2003).
- 111 Bihour, C. *et al.* Expression of markers of platelet activation and the interpatient variation in response to abciximab. *Arteriosclerosis, thrombosis, and vascular biology* **19**, 212-219, doi:10.1161/01.atv.19.2.212 (1999).
- 112 Dropinski, J. *et al.* Anti-thrombotic action of clopidogrel and P1(A1/A2) polymorphism of beta3 integrin in patients with coronary artery disease not being treated with aspirin. *Thromb Haemost* **94**, 1300-1305 (2005).
- 113 Abdullah, W. Z., Roshan, T. M., Hussin, A., Zain, W. S. & Abdullah, D. Increased PAC-1 expression among patients with multiple myeloma on concurrent thalidomide and warfarin. *Blood Coagul Fibrinolysis* **24**, 893-895, doi:10.1097/MBC.0b013e3283642ee2 (2013).
- 114 Chung, T. *et al.* Platelet activation in acute pulmonary embolism. *Journal of thrombosis and haemostasis : JTH* **5**, 918-924, doi:10.1111/j.1538-7836.2007.02461.x (2007).
- 115 Manne, B. K. *et al.* Platelet gene expression and function in patients with COVID-19. *Blood* **136**, 1317-1329, doi:10.1182/blood.2020007214 (2020).
- 116 Stumpf, C. *et al.* Enhanced levels of platelet P-selectin and circulating cytokines in young patients with mild arterial hypertension. *J Hypertens* **23**, 995-1000, doi:10.1097/01.hjh.0000166840.63312.12 (2005).
- 117 Preston, R. A., Coffey, J. O., Materson, B. J., Ledford, M. & Alonso, A. B. Elevated platelet P-selectin expression and platelet activation in high risk patients with uncontrolled severe hypertension. *Atherosclerosis* **192**, 148-154, doi:10.1016/j.atherosclerosis.2006.04.028 (2007).
- 118 Amalia, L. The Role of Platelet-Selectin as a Marker of Thrombocyte Aggregation on Cerebral Sinus Venous Thrombosis. *J Blood Med* **13**, 267-274, doi:10.2147/JBM.S356028 (2022).
- 119 Minamino, T. *et al.* Increased expression of P-selectin on platelets is a risk factor for silent cerebral infarction in patients with atrial fibrillation: role of nitric oxide. *Circulation* **98**, 1721-1727, doi:10.1161/01.cir.98.17.1721 (1998).

120 Michelson, A. D. Platelet function testing in cardiovascular diseases. *Circulation* **110**, e489-493, doi:10.1161/01.CIR.0000147228.29325.F9 (2004).

REVIEWERS' COMMENTS

Reviewer #1 (Remarks to the Author):

no further comments. recommend for publication as it is

Reviewer #2 (Remarks to the Author):

The authors have addressed all the questions satisfactorily, and I have no further inquiries. However, I would like to offer a few comments and suggestions for this significant research.

This study effectively identifies “biomechanical platelet aggregation” driven by shear force, where GPIIb α -VWF interactions initiate platelet crosslinking and integrin α IIb β 3 activation. The barcode system for anti-thrombotic drug evaluation and prothrombotic status characterization is commendable, offering valuable insights. Future studies are needed to validate E+ α IIb β 3 as a biomarker and to investigate the impact of comorbidities, such as diabetes and obesity, on thrombus profiles. Additionally, expanding clinical validation will strengthen the findings.